



# CSIRO Environmental Modelling Suite (EMS): Scientific description of the optical and biogeochemical models (vB3p0).

Mark E. Baird[1], Karen A. Wild-Allen[1], John Parslow[1], Mathieu Mongin[1], Barbara Robson[2], Jennifer Skerratt[1], Farhan Rizwi[1], Monika Soja-Wozniak[1], Emlyn Jones[1], Mike Herzfeld[1], Nugzar Margvelashvili[1], John Andrewartha[1], Clothilde Langlais[1], Matthew P. Adams[3], Nagur Cherukuru[4], Malin Gustafsson[5], Scott Hadley[1], Peter J. Ralph[5], Uwe Rosebrock[1], Thomas Schroeder[1], Leonardo Laiolo[1], Daniel Harrison[6], and Andrew D. L. Steven[1]

[1]CSIRO, Oceans and Atmosphere, Hobart, Australia
[2]Australian Institute of Marine Science, Townsville, Australia
[3]School of Chemical Engineering, The University of Queensland, Brisbane, Australia
[4]CSIRO, Land and Water, Canberra, Australia
[5]Plant Functional Biology and Climate Change Cluster, Faculty of Science, University of Technology Sydney, Sydney, Australia
[6]Southern Cross University, Coffs Harbour, Australia

**Correspondence:** Mark Baird (mark.baird@csiro.au)

**Abstract.**

Since the mid 1990s, Australia's Commonwealth Science Industry and Research Organisation (CSIRO) has developed a biogeochemical (BGC) model for coupling with a hydrodynamic and sediment model for application in estuaries, coastal waters and shelf seas. The suite of coupled models is referred to as the CSIRO Environmental Modelling Suite (EMS) and has

been applied at tens of locations around the Australian continent. At a mature point in the BGC model's development, this paper presents a full mathematical description, as well as links to the freely available code and User Guide. The mathematical description is structured into processes so that the details of new parameterisations can be easily identified, along with their derivation. The EMS BGC model cycles carbon, nitrogen, phosphorous and oxygen through multiple phytoplankton, zooplankton, detritus and dissolved organic and inorganic forms in multiple water column and sediment layers. The underwater light

field is simulated by a spectrally-resolved optical model that includes the calculation of water-leaving reflectance for validation with remote sensing. The water column is dynamically coupled to the sediment to resolve deposition, resuspension and benthic-pelagic biogeochemical fluxes. With a focus on shallow waters, the model also includes particularly-detailed representations of benthic plants such as seagrass, macroalgae and coral polyps. A second focus has been on, where possible, the use of geometric derivations of physical limits to constrain ecological rates, which generally requires population-based rates to

be derived from initially considering the size and shape of individuals. For example, zooplankton grazing considers encounter rates of one predator on a prey field based on summing relative motion of the predator with the prey individuals and the search area, chlorophyll synthesis includes a geometrically-derived self-shading term, and the bottom coverage of benthic plants is generically-related to their biomass using an exponential form derived from geometric arguments. This geometric approach has led to a more algebraically-complicated set of equations when compared to more empirical biogeochemical model formu-



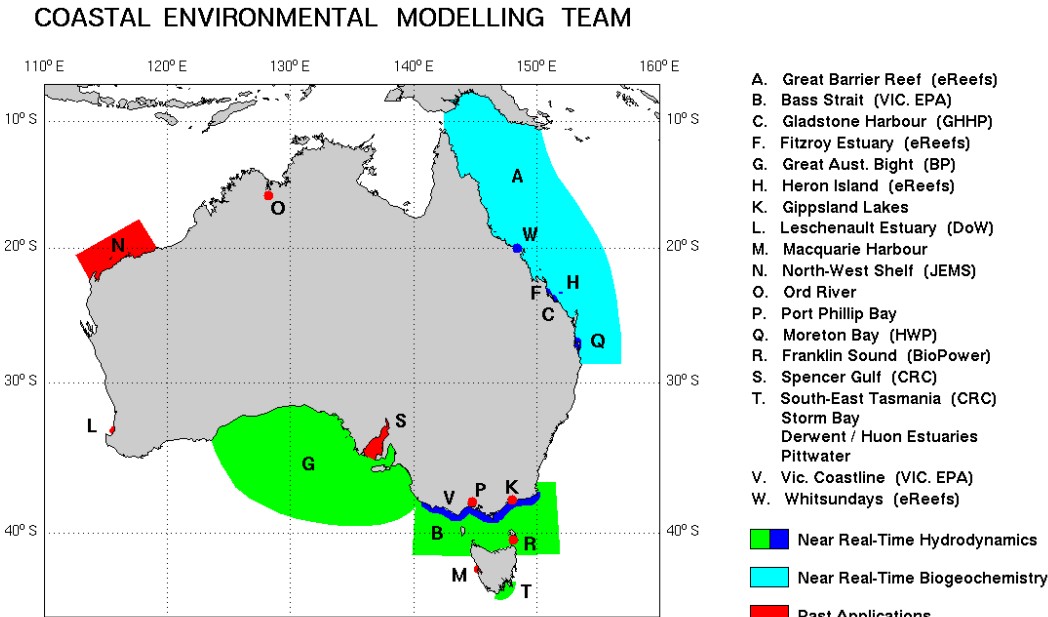

**Figure 1.** Model domains of the CSIRO EMS hydrodynamic and biogeochemical applications from 1996 onwards. Additionally EMS was used for the nation-wide Simple Estuarine Response Model (SERM), that was applied generically around Australia's 1000+ estuaries (Baird et al., 2003). Brackets refer to specific funding bodies. EMS has also been applied in the Los Lagos region of Chile. A full list of past and current applications and funding bodies is available at: `https://research.csiro.au/cem/projects/`.

lations. But while being algebraically-complicated, the model has fewer unconstrained parameters and is therefore simpler to move between applications than it would otherwise be. The version of the biogeochemistry described here is implemented in the eReefs project that is delivering a near real time coupled hydrodynamic, sediment and biogeochemical simulation of the Great Barrier Reef, northeast Australia, and its formulation provides an example of the application of geometric reasoning in

5    the formulation of aquatic ecological processes.

**Keywords.** Great Barrier Reef, mechanistic model, geometric derivation

# 1   Introduction

The first model of marine biogeochemistry was a developed more than 70 years ago to explain phytoplankton blooms

10   (Riley, 1947). Today the modelling of estuarine, coastal and global biogeochemical systems has been used for a wide variety of



applications including coastal eutrophication (Madden and Kemp, 1996; Baird et al., 2003), shelf carbon and nutrient dynamics (Yool and Fasham, 2001; Dietze et al., 2009), plankton ecosystem diversity (Follows et al., 2007), ocean acidification (Orr et al., 2005), impact of local developments such as fish farms and sewerage treatment plants (Wild-Allen et al., 2010), fishery production (Stock et al., 2008) and operational forecasting (Fennel et al., 2019), to name a few. As a result of these varied

applications, a diverse range of biogeochemical models have emerged, with some models developed over decades and being capable of investigating a suite of biogeochemical phenomena (Butenschön et al., 2016). With model capabilities typically dependent on the history of applications for which a particular model has been funded, and perhaps even the backgrounds and interests of the developers themselves, significant differences exist between models. Thus it is vital that biogeochemical models are accurately described in full (e.g. Butenschön et al. (2016); Aumont et al. (2015) and Dutkiewicz et al. (2015)), so

that model differences can be understood, and, where useful, innovations shared between modelling teams.

Estuarine, coastal and shelf modelling projects undertaken over the past 20+ years by Australia's national science agency, the Commonwealth Science Industry and Research Organisation (CSIRO), have led to the development of the CSIRO Environmental Modelling Suite (EMS). The EMS contains a suite of hydrodynamic, transport, sediment, optical and biogeochemical models that can be run coupled or sequentially. The EMS biogeochemical model, the subject of this paper, has been applied

around the Australian coastline (Fig. 1) leading to characteristics of the model which have been tailored to the Australian environment and its challenges.

Australian shelf waters range from tropical to temperate, micro- to macro-tidal, with shallow waters containing coral, sea-grass or algae-dominated benthic communities. With generally narrow continental shelves, and being surrounded by two poleward-flowing boundary currents (Thompson et al., 2009), primary production in Australian coastal environments is gen-

20 erally limited by dissolved nitrogen in marine environments, phosphorus in freshwaters, and unlimited by silica and iron. The episodic nature of rainfall on the Australian continent, especially in the tropics, and a lack of snow cover, delivers intermittent but occasionally extreme river flows to coastal waters. With a low population density, continent-wide levels of human impacts are small relative to other continents, but can be significant locally, often due to large isolated developments such as dams, irrigation schemes, mines and ports. Global changes such as ocean warming and acidification affect all regions. The EMS

BGC model has many structural features similar to other models (e.g. multiple plankton functional types, nutrient and detrital pools, an increasing emphasis on optical and carbon chemistry components). Nonetheless, the geographical characteristics of, and anthropogenic influences on, the Australian continent have shaped the development of EMS, and led to a BGC model with many unique features.

As the national science body, CSIRO needed to develop a numerical modelling system that could be deployed across the

30 broad range of Australian coastal environments and capable of resolving multiple anthropogenic impacts. With a long coastline (60,000+ km by one measure), containing over 1000 estuaries, an Australian-wide configuration has insufficient resolution to be used for many applied environmental challenges. Thus, in 1999, the EMS biogeochemical model development was targeted to increase its applicability across a range of ecosystems. In particular, given limited resources to model a large number of environments / ecosystems, developments aimed to minimise the need for re-parameterisation of biogeochemical processes

for each application. Two innovations arose from this imperative: 1. the software development of a process-based modelling





architecture, such that processes could be included, or excluded, while using the same executable file; and 2. the use, where possible, of geometric descriptions of physical limits to ecological processes as a means of reducing parameter uncertainty (Baird et al., 2003). It is the use of these geometric descriptions that has led to the greatest differences between EMS and other aquatic biogeochemical models.

In the aquatic sciences there has been a long history of experimental and process studies that use geometric arguments to quantify ecological processes, but these derivations have rarely been applied in biogeochemical models, with notable exceptions (microalgal light absorption and plankton sinking rates generally, surface area to volume considerations (Reynolds, 1984), among others). By prioritising geometric arguments, EMS has included a number of previously-published geometric forms including diffusion limitation of microalgae nutrient uptake (Hill and Whittingham, 1955), absorption cross-sections of

microalgae (Fig. 2C, Duysens (1956); Kirk (1975); Morel and Bricaud (1981), diffusion limits to macroalgae and coral nutrient uptake (Munk and Riley, 1952; Atkinson and Bilger, 1992; Zhang et al., 2011), and encounter-rate limitation of grazing rates (Fig. 2B, Jackson (1995)).

Perhaps the most important consequence of using geometric constraints in the BGC model is the representation of benthic flora as two dimensional surfaces, while plankton are represented as three dimensional suspended objects (Baird et al., 2003).

Thus leafy benthic plants such as macroalgae take up nutrients and absorb light on a 2D surface. In contrast, nutrient uptake to microalgae occurs through a 3D field while light uptake of the 3D cell is limited by the 2D projected area (Fig. 2A). These geometric properties, from which the model equations are derived, generates greater potential light absorption relative to nutrient uptake of benthic communities relative to the same potential light absorption relative to nutrient uptake in unicellular algae (Baird et al., 2004). In the most simple terms, this can be related to the surface area to projected area of a leaf being 1/4

20    times that of a microalgae cell (Fig. 2A). Thus the competition for nutrients, ultimately being driven by light absorption and its rate compared to nutrient uptake, is explicitly determined by the contrasting geometries of cells and leaves.

In addition geometric constraints derived by others, a number of novel geometric descriptions have been introduced into the EMS BGC model, including:

1. Geometric derivation of the relationship between biomass, $B$, and fraction of the bottom covered, $A_{eff} = 1 - \exp(-\Omega B)$,
where $\Omega$ is the nitrogen-specific leaf area (Sec. 4).

2. Impact of self-shading on chlorophyll synthesis quantified by the incremental increase in absorption with the increase in pigment content (Sec. 3.3.3).

3. Mass-specific absorption coefficients of photosynthetic pigments have been better utilised to determine phytoplankton absorption cross-sections (Duysens, 1956; Kirk, 1975; Morel and Bricaud, 1981) through the availability of a library of
mass-specific absorption coefficients (Clementson and Wojtasiewicz, 2019), and their wavelength correction using the refractive index of the solvent used in the laboratory determinations (Fig. 6).

4. The space-limitation of zooxanthellae within coral polyps using zooxanthellae projected areas in a two layer gastroder-mal cell anatomy (Sec. 4.4.1).

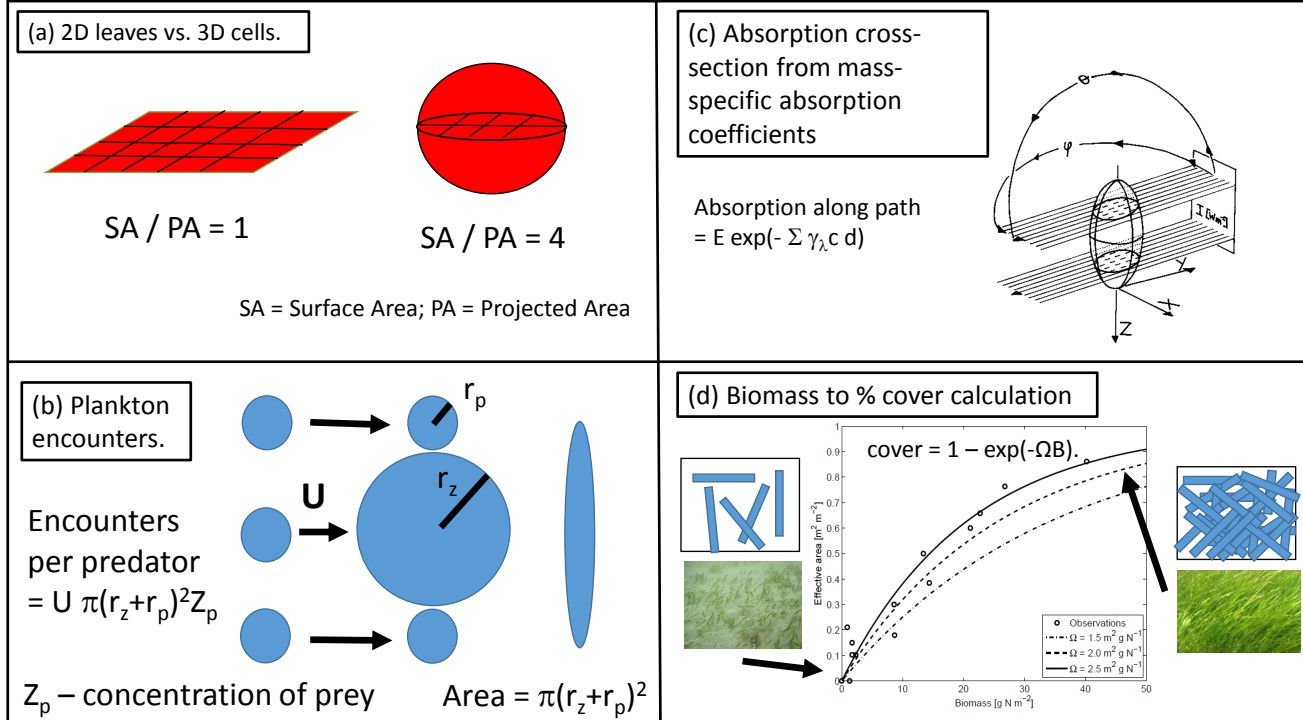

**Figure 2.** Examples of geometric descriptions of ecological processes. (a) The relative difference in the 2D experience to nutrient and light fields of leaves compared to the 3D experience of cells, as typified by the ratio of surface area (coloured) to projected area (hashed area); (b) The encounter rate of prey per individual predator as a function of the radius of encounter (the sum the predator and prey radii) and the relative motion and prey concentration following Jackson (1995); (c) The use of ray tracing and the mass-specific absorption coefficient to calculate an absorption cross section for a randomly oriented spheroid following (Kirk, 1975); (d) Fraction of the bottom covered as seen from above as a result of increasing the number of randomly placed leaves (Baird et al., 2016a). Based on the assumption that leaves are randomly placed, the cover reaches $1 - \exp(-1) = 0.63$ when the sum of the shaded areas induced by all individual leaves equals the ground area (i.e. a Leaf Area Index of 1).

5. Preferential ammonia uptake, which is often calculated using different half-saturation coefficients of nitrate and ammonia uptake (Lee et al., 2002a), is determined in the EMS BGC model by allowing ammonia uptake to proceed up to the diffusion limit. Should this diffusion limit not meet the required demand, nitrate uptake supplements the ammonia uptake. This representation has the benefit that no additional parameters are required to assign preference, with the same approach can be applied for both microalgae and benthic plants (Sec. 6.1).

To be clear, these geometric definitions have their own set of assumptions (e.g. a single cell size for a population), and simplifications (e.g. spherical shape). Nonetheless, the effort to apply geometric descriptions of physical limits across the BGC





model appears to have been beneficial, as measured by the minimal amount of re-parameterisation that has been required to apply the model to contrasting environments. Of the above mentioned new formulations, the most useful and easily applied is the bottom cover calculation (Fig. 2D). In fact it is so simple, and such a clear improvement on empirical forms as demonstrated in Baird et al. (2016a), that it is likely to have been applied in other ecological / biogeochemical models, although we are
unaware of any other implementation.

In addition to using geometric descriptions, there are a few other features unique to the EMS BGC model including:

1. Calculation of remote-sensing reflectance from an optical-depth weighted ratio of backscatter to absorption plus backscatter (Sec. 3.2.2).

2. Calculation of scalar irradiance from downwelling irradiance, vertical attenuation and a photon balance within a layer
(Sec. 3.2.2).

3. An oxygen balance achieved through use of biological and chemical oxygen demand tracers (Sec. 7.5.2).

The calculation of remote-sensing reflectance, which is also undertaken in other biogeochemical modelling studies (Dutkiewicz et al., 2015), is one example of an approach that we are pursuing to bring the model outputs closer to the observations (Baird et al., 2016b). The most dramatical example of this is the development of simulated true colour, which renders model calcula-
tions of spectrally-resolved remote-sensing reflectance as would be expected by the human visual experience (Fig. 3).

## 1.1   Outline

This document provides a summary of the biogeochemical processes included in the model (Sec. 2), a full description of the model equations (Sec. 3 - Sec. 7), as well as links to model evaluation (Sec. 8), code availability (Sec. 9) and test case
generation (Sec. 10). The description of the optical and biogeochemical models is divided into the primary environmental zones: pelagic (Sec. 3), epibenthic (Sec. 4) and sediment (Sec. 5), as well as processes that are common to all zones (Sec. 6) and numerical integration details (Sec. 7). Within these zones, descriptions are sorted by processes, such as microalgae growth, coral growth, food web interactions etc. This organisation allows the model to be explained, with notation, in self-contained chunks. For each process the complete set of model equations, parameter values and state variables are given in tables. As the
code itself allows the inclusion / exclusion of processes at runtime, the process-based structuring of this document aligns with the architecture of the code. To investigate the complete equation for a single state variable, such as nitrate concentration, the reader will need to combine the individual terms affecting the variable from all processes (such as nitrate uptake by each of the autotrophs, remineralisation etc.). Section 7 gives some of the details of the numerical methods that solve the model equations. The Discussion (Sec. 11) details how past and present applications have influenced the development of the EMS BGC model,
and anticipates some future developments. Finally, the Appendices gives tables of state variables and parameters, with both mathematical and numerical code details.

**Figure 3.** Observed (top) and simulated (bottom) true colour from simulated remote-sensing reflectance at 670, 555 and 470 nm in the GBR4 model configuration in the region of the Burdekin River. A brightening of 10 (left) and 20 (right) was applied for comparison. See Baird et al. (2016b).



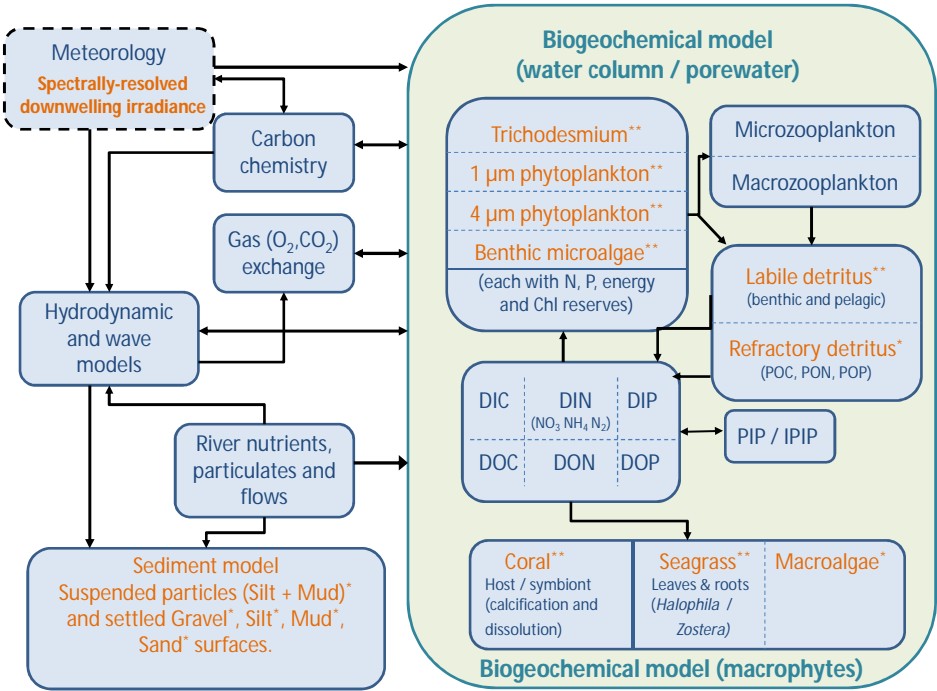

**Figure 4.** Schematic of CSIRO Environmental Modelling Suite, illustrating the biogeochemical processes in the water column, epipelagic and sediment zones, as well as the carbon chemistry and gas exchange used in vB3p0 for the Great Barrier Reef application.

## 2 Overview of the EMS biogeochemical and optical model

### 2.1 Spectrally-resolved optical model

The optical model undertakes calculations at distinct wavelengths of light (say 395, 405, 415, ... 705 nm) representative of individual wavebands (say 400-410, 410-420 nm etc.), and can be modified for the particular application. First the spectrally-resolved Inherent Optical Properties (IOPs) of the water column are calculated from the optically-active model biogeochemical state (phytoplankton biomass, particulate concentrations etc.). These include the absorption and scattering properties of clear water, coloured dissolved organic matter, suspended solids and each microalgal population.

Using the calculated IOPs, as a well as sun angle, surface albedo and refraction, the spectrally-resolved light field (downwelling and scalar irradiance) is calculated for each grid cell in the model. From this light field phytoplankton absorption is calculated. The light that reaches the bottom is absorbed by epibenthic flora as a function of wavelength, depending on the absorbance of each individual flora. From the calculation of the light field other apparent optical properties (AOPs), such as remote-sensing reflectance, can be determined and compared to either in situ measurements or remotely-sensed products.





As AOPs can be recalculated from IOPs post-simulation, the model can be run for one set of wavelengths to optimise the integration speed and accuracy, and the AOPs re-calculated at another set of wavelengths for comparison with hyperspectral observations such as those used to calculate chlorophyll from the ocean color sensors.

The use of remote-sensing reflectance introduces a novel means of model assessment - simulated true colour. The model

output can be processed to produce simulated true colour images of the water surface, with features such as bottom reflectance, river plumes, submerged coral reefs and microalgal blooms easily characterised by their colour (Baird et al., 2016b).

## 2.2    Biogeochemical model

The ecological model is organised into 3 zones: pelagic, epibenthic and sediment. Depending on the grid formulation the pelagic zone may have one or several layers of similar or varying thickness. The epibenthic zone overlaps with the lowest

pelagic layer and the top sediment layer and shares the same dissolved and suspended particulate material fields. The sediment is modelled in multiple layers with a thin layer of easily resuspendable material overlying thicker layers of more consolidated sediment.

Dissolved and particulate biogeochemical tracers are advected and diffused throughout the model domain in an identical fashion to temperature and salinity. Additionally, biogeochemical particulate substances sink and are resuspended in the same

way as sediment particles. Biogeochemical processes are organized into pelagic processes of phytoplankton and zooplankton growth and mortality, detritus remineralisation and fluxes of dissolved oxygen, nitrogen and phosphorus; epibenthic processes of growth and mortality of macroalgae, seagrass and corals, and sediment based processes of plankton mortality, microphytobenthos growth, detrital remineralisation and fluxes of dissolved substances (Fig. 4).

The biogeochemical model considers four groups of microalgae (small and large phytoplankton, microphytobenthos and

*Trichodesmium*), four macrophytes types (seagrass types corresponding to *Zostera*, *Halophila*, deep *Halophila* and macroalgae) and coral communities. For temperate system applications of the EMS, dinoflagellates, *Nodularia* and multiple macroalgal species have also been characterised (Wild-Allen et al., 2013; Hadley et al., 2015a)

Photosynthetic growth is determined by concentrations of dissolved nutrients (nitrogen and phosphate) and photosynthetically active radiation. Autotrophs take up dissolved ammonium, nitrate, phosphate and inorganic carbon. Microalgae incor-

porate carbon (C), nitrogen (N) and phosphorus (P) at the Redfield ratio (106C:16N:1P) while macrophytes do so at the Atkinson ratio (550C:30N:1P). Microalgae contain two pigments (chlorophyll *a* and an accessory pigment), and have variable carbon:pigment ratios determined using a photoadaptation model.

Micro- and meso-zooplankton graze on small and large phytoplankton respectively, at rates determined by particle encounter rates and maximum ingestion rates. Additionally large zooplankton consume small zooplankton. Of the grazed material that is

not incorporated into zooplankton biomass, half is released as dissolved and particulate carbon, nitrogen and phosphate, with the remainder forming detritus. Additional detritus accumulates by mortality. Detritus and dissolved organic substances are remineralised into inorganic carbon, nitrogen and phosphate with labile detritus transformed most rapidly (days), refractory detritus slower (months) and dissolved organic material transformed over the longest timescales (years). The production (by photosynthesis) and consumption (by respiration and remineralisation) of dissolved oxygen is also included in the model





and depending on prevailing concentrations, facilitates or inhibits the oxidation of ammonia to nitrate and its subsequent denitrification to di-nitrogen gas which is then lost from the system.

Additional water column chemistry calculations are undertaken to solve for the equilibrium carbon chemistry ion concentrations necessary to undertake ocean acidification (OA) studies, and to consider sea-air fluxes of oxygen and carbon dioxide. The

5 adsorption and desorption of phosphorus onto inorganic particles as a function of the oxic state of the water is also considered.

In the sediment porewaters, similar remineralisation processes occur as in the water column (Fig. 5). Additionally, nitrogen is denitrified and lost as $N_2$ gas while phosphorus can become adsorbed onto inorganic particles, and become permanently immobilised in sediments.

## 3 Pelagic processes

### 3.1 Transport

The local rate of change of concentration of each dissolved and particulate constituent, $C$, contains sink/source terms, $S_C$, which are described in length in this document, and the advection, diffusion and sinking terms:

$$\frac{\partial C}{\partial t} + \mathbf{v} \cdot \nabla^2 C = \nabla \cdot (K \nabla C) + w_{sink} \frac{\partial C}{\partial z} + S_C \tag{1}$$

where the symbol $\nabla = \left( \frac{\partial}{\partial x}, \frac{\partial}{\partial y}, \frac{\partial}{\partial z} \right)$, $\mathbf{v}$ is the velocity field, $K$ is the eddy diffusion coefficient which varies in space and time, and $w_C$ is the local sinking rate (positive downwards) and the $z$ co-ordinate is positive upwards. The calculation of $\mathbf{v}$ and $K$ is described in the hydrodynamic model (Herzfeld, 2006; Gillibrand and Herzfeld, 2016).

The microalgae are particulates that contain internal concentrations of dissolved nutrients (C, N, P) and pigments that are specified on a per cell basis. To conserve mass, the local rate of change of the concentration of microalgae, $B$, multiplied by

20 the content of the cell, $R$, is given by:

$$\frac{\partial (BR)}{\partial t} + \mathbf{v} \cdot \nabla^2 (BR) = \nabla \cdot (K \nabla (BR)) + w_C \frac{\partial (BR)}{\partial z} + S_{BR} \tag{2}$$

For more information see Sec. 3.3.5 and Sec. 3.1 of Baird et al. (2004).

### 3.2 Optical model

The optical model considers the processes of absorption and scattering by clear water, coloured dissolved organic matter

(CDOM), non-algal particulates (NAP) and phytoplankton cells. First the inherent optical properties (IOPs), such as spectrally-resolved total phytoplankton absorption, are calculated from the model state variables (e.g. phytoplankton chlorophyll biomass) and model parameters (e.g. cell radius). The optical model then solves for the apparent optical properties (AOPs), such as the spectrally-resolved scalar irradiance, from the surface downwelling light field and the IOPs. Finally, the AOPs can be directly compared to remotely-sensed products such as remote-sensing reflectance and simulated true colour images.





**Figure 5.** Schematic of sediment nitrogen and phosphorus pools and fluxes. Processes represented include phytoplankton mortality, detrital decomposition, denitrification (nitrogen only), phosphorus adsorption (phosphorus only) and microphytobenthic growth.





### 3.2.1 Inherent optical properties (IOPs)

*Phytoplankton absorption.* The model contains 4 phytoplankton types (small and large phytoplankton, benthic mircoalgae and *Trichodesmium*), each with a unique ratio of internal concentration of accessory photosynthetic pigments to chlorophyll-a. To calculate the absorption due to each pigment, we use a database of spectrally-resolved mass-specific absorption coefficients

(Clementson and Wojtasiewicz, 2019). As it can be assumed that accessory pigments stay in a constant ratio to chlorophyll-$a$, the model needs only a state variable for chlorophyll-$a$ for each phytoplankton type. The model then calculates the chlorophyll-$a$ specific absorption coefficient due to all pigments by using the Chl-a state variable, the ratio of concentration of the accessory pigment to chlorophyll-$a$, and the mass-specific absorption coefficient of each of the accessory pigments. Thus the chlorophyll-$a$ specific absorption coefficient due to all photosynthetic pigments for small phytoplankton at wavelength $\lambda$, $\gamma_{small,\lambda}$, is given

by:

$$\gamma_{small,\lambda} = 1.0\gamma_{Chla,\lambda} + 0.35\gamma_{Zea,\lambda} + 0.05\gamma_{Echi,\lambda} + 0.1\gamma_{\beta-car,\lambda} + 2\gamma_{PE,\lambda} + 1.72\gamma_{PC,\lambda} \tag{3}$$

where Chla is the pigment chlorophyll-$a$, Zea is zeaxanthin, Echi is echinenone, $\beta$-car is beta-carotene, PE is phycoerithin, and PC is phycocyanin, and the ratios of chlorophyll-$a$ to accessory pigment concentration are determined from Wojtasiewicz and Stoń-Egiert (2016). Note that the coefficient in Eq. 3 for Chla is 1.0 because the ratio of chlorophyll-$a$ to chlorophyll-$a$ is 1.

The resulting chlorophyll-$a$ specific absorption coefficient is shown in Fig. 6.

Similarly for large phytoplankton and microphytobenthos (Wright et al., 1996):

$$\gamma_{large,\lambda} = 1.0\gamma_{Chla,\lambda} + 0.6\gamma_{Fuco,\lambda} \tag{4}$$

where Fuco is fucoxanthin. And for *Trichodesmium* (Carpenter et al., 1993) :

$$\gamma_{Tricho,\lambda} = 1.0\gamma_{Chla,\lambda} + 0.1\gamma_{Zea,\lambda} + 0.02\gamma_{Myxo,\lambda} + 0.09\gamma_{\beta-car,\lambda} + 2.5\gamma_{PE,\lambda} \tag{5}$$

where Myxo is myxoxanthophyll.

The absorption cross-section at wavelength $\lambda$ ($\alpha_\lambda$) of a spherical cell of radius ($r$), chla-specific absorption coefficient ($\gamma_\lambda$), and homogeneous intracellular chlorophyll-$a$ concentration ($c_i$) can be calculated using geometric optics (i.e., ray tracing without considering internal scattering) and is given by (Duysens, 1956; Kirk, 1975):

$$\alpha_\lambda = \pi r^2 \left(1 - \frac{2(1 - (1 + 2\gamma_\lambda c_i r)e^{-2\gamma_\lambda c_i r})}{(2\gamma_\lambda c_i r)^2}\right) \tag{6}$$

where $\pi r^2$ is the projected area of a sphere, and the bracketed term is 0 for no absorption ($\gamma c_i r = 0$) and approaches 1 as the cell becomes fully opaque ($\gamma c_i r \rightarrow \infty$). Note that the bracketed term in Eq. 6 is mathematically equivalent to the dimensionless efficiency factor for absorption, $Q_a$ (used in Morel and Bricaud (1981), Finkel (2001) and Bohren and Huffman (1983)), of homogeneous spherical cells with an index of refraction close to that of the surrounding water.

The use of an absorption cross-section of an individual cell has two significant advantages. Firstly, the same model parame-

30 ters used here to calculated absorption in the water column are used to determine photosynthesis by individual cells, including





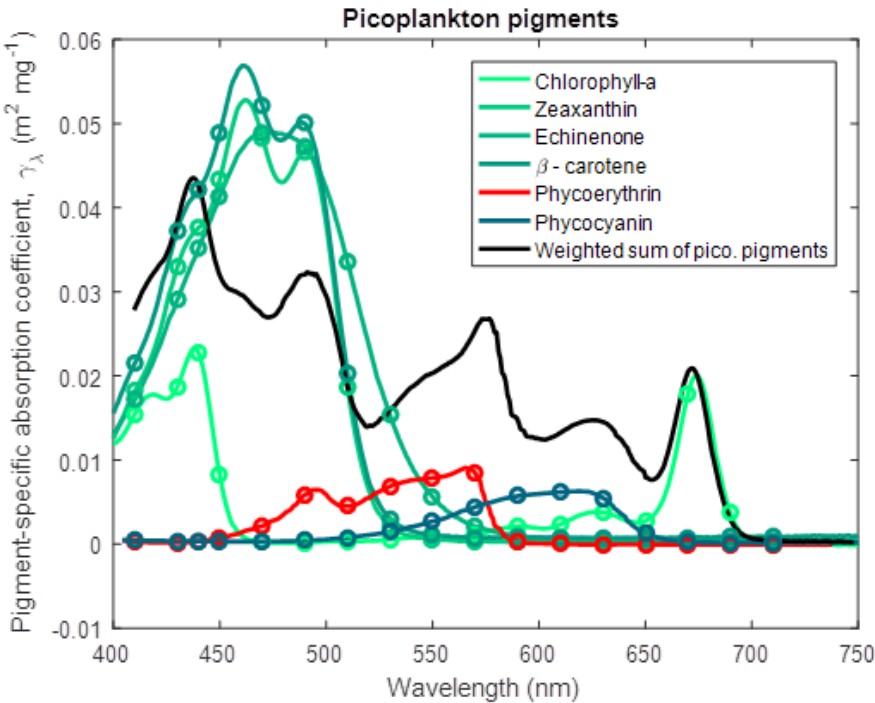

**Figure 6.** Pigment-specific absorption coefficients for the dominant pigments found in small phytoplankton determined using laboratory standards in solvent in a 1 cm vial. Green and red lines are photosynthetic pigments constructed from 563 measured wavelengths. Circles represent the wavelengths at which the optical properties are calculated in the simulations. The black line represents the weighted sum of the photosynthetic pigments (Eq. 3), with the weighting calculated from the ratio of each pigment concentration to chlorophyll *a*. The spectra are wavelength-shifted from their raw measurement by the ratio of the refractive index of the solvent to the refractive index of water (1.352 for acetone used with chlorophyll *a* and $\beta$-carotene; 1.361 for ethanol used with zeaxanthin, echinenone; 1.330 for water used with phycoerythrin, phycocyanin).

the effect of packaging of pigments within cells. Secondly, the dynamic chlorophyll concentration determined later can be explicitly included in the calculation of phytoplankton absorption. Thus the absorption of a population of $n$ cell m$^{-3}$ is given by $n\alpha$ m$^{-1}$, while an individual cell absorbs $\alpha E_o$ light, where $E_o$ is the scalar irradiance.

*Coloured Dissolved Organic Matter (CDOM) absorption.* Two equations for CDOM absorption are presently being trialled. 5 The two schemes are:

*Scheme 1.* The absorption of CDOM, $a_{CDOM,\lambda}$, is determined from a relationship with salinity in the region (Schroeder et al., 2012):

$$a_{CDOM,443} = -0.0332S + 1.2336 \tag{7}$$



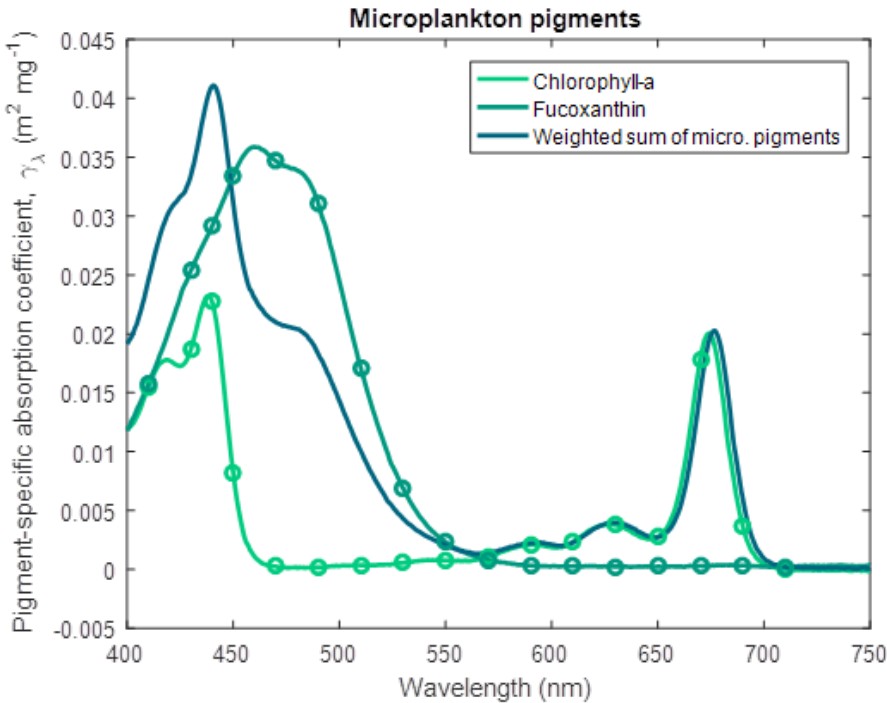

**Figure 7.** Pigment-specific absorption coefficients for the dominant pigments found in large phytoplankton and microphytobenthos determined using laboratory standards in solvent in a 1 cm vial. The aqua line represents the weighted sum of the photosynthetic pigments (Eq. 4), with the weighting calculated from the ratio of each pigment concentration to chlorophyll *a*. See Fig. 6 for more details. Fucoxanthin was dissolved in ethanol.

where $S$ is the salinity. In order to avoid unrealistic extrapolation, the salinity used in this relationship is the minimum of the model salinity and 36. In some cases coastal salinities exceed 36 due to evaporation. The absorption due to CDOM at other wavelengths is calculated using a CDOM spectral slope for the region (Blondeau-Patissier et al., 2009):

$$a_{CDOM,\lambda} = a_{CDOM,443} \exp\left(-S_{CDOM}\left(\lambda - 443.0\right)\right) \tag{8}$$

5  where $S_{CDOM}$ is an approximate spectral slope for CDOM, with observations ranging from 0.01 to 0.02 nm$^{-1}$ for significant concentrations of CDOM. Lower magnitudes of the spectral slope generally occur at lower concentrations of CDOM (Blondeau-Patissier et al., 2009).

*Scheme 2.* The absorption of CDOM, $a_{CDOM,\lambda}$, is directly related to the concentration of dissolved organic carbon, $D_C$.

$$a_{CDOM,\lambda} = k^*_{CDOM,443} D_C \exp\left(-S_{CDOM}\left(\lambda - 443.0\right)\right) \tag{9}$$

10  where $k^*_{CDOM,443}$ is the dissolved organic carbon-specific CDOM absorption coefficient at 443 nm.

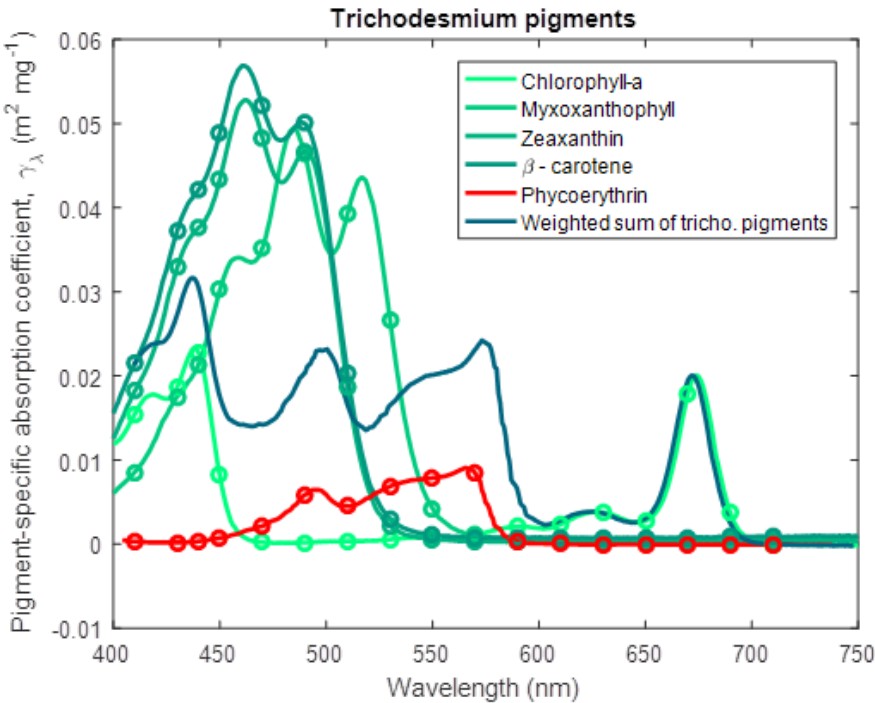

**Figure 8.** Pigment-specific absorption coefficients for the dominant pigments found in *Trichodesmium* determined using laboratory standards in solvent in a 1 cm vial. The aqua line represents the weighted sum of the photosynthetic pigments (Eq. 5), with the weighting calculated from the ratio of each pigment concentration to chlorophyll *a*. See Fig. 6 for more details. Myxoxanthophyll was dissolved in acetone.

Both schemes have drawbacks. Scheme 2, using the concentration of dissolved organic carbon, is closer to reality, but is likely to be sensitive to poorly-known parameters such as remineralisation rates and initial detritial concentrations. Scheme 1, a function of salinity, will be more stable, but perhaps less accurate, especially in estuaries where hypersaline waters may have large estuarine loads of coloured dissolved organic matter.

5    *Absorption due to non-algal particulate material*. The waters of the Great Barrier Reef contain suspended sediments originating from various marine sources, such as the white calcium carbonate fragments generated by coral erosion, and sediments derived from terrestrial sources such as granite (Soja-Woźniak et al., submitted). The model uses spectrally-resolved mass-specific absorption coefficients (and also total scattering measurements) from a database of laboratory measurements conducted on either pure mineral suspensions, or mineral mixtures, at two ranges of size distributions (Fig. 9, Stramski et al.

10   (2007)). In this model version we use the calcium carbonate sample CAL1 for $CaCO_3$-based particles

For the terrestrially-sourced particles we used observations from Gladstone Harbour in the central GBR (Fig. 10). These IOPs gave a realistic surface colour for the Queensland river sediment plumes (Baird et al., 2016b). In the model, optically-active non-algal particulates (NAPs) includes the inorganic particulates (such as sand and mud, see Sec. 5.1) and detritus. We





|  | Symbol | Value |
|---|---|---|
| *Constants* | | |
| Speed of light | $c$ | $2.998 \times 10^8$ m s$^{-1}$ |
| Planck constant | $h$ | $6.626 \times 10^{-34}$ J s$^{-1}$ |
| Avogadro constant | $A_V$ | $6.02 \times 10^{23}$ mol$^{-1}$ |
| [a]Total scattering coefficient of phytoplankton | $b_{phy}$ | 0.2 (mg Chl $a$ m$^{-2}$)$^{-1}$ |
| [b]Azimuth-independent scattering coefficient | $g_i$ | 0.402 |
| [b]Azimuth-dependent scattering coefficient | $g_{ii}$ | 0.180 |
| [c]CDOM-specific absorption coefficient at 443 nm | $k^*_{CDOM,443}$ | 0.02 m$^2$ mg C$^{-1}$ |
| [c]Spectral slope of CDOM absorption | $S_{CDOM}$ | 0.012 nm$^{-1}$ |
| [d]Linear remote-sensing reflectance coefficient | $g_0$ | 0.0895 |
| [d]Quadratic remote-sensing reflectance coefficient | $g_1$ | 0.1247 |

**Table 1.** Constants and parameter values used in the optical model.[a] Kirk (1994).[b] Kirk (1991) using an average cosine of scattering of 0.924 (Mobley, 1994). [c] Blondeau-Patissier et al. (2009) see also Cherukuru et al. (2019). [d] Brando et al. (2012). [e] Vaillancourt et al. (2004).

|  | Symbol | Units |
|---|---|---|
| Downwelling irradiance at depth $z$, wavelength $\lambda$ | $E_{d,z,\lambda}$ | W m$^{-2}$ |
| Scalar irradiance at depth $z$, wavelength $\lambda$ | $E_{o,z,\lambda}$ | W m$^{-2}$ |
| In water azimuth angle | $\theta$ | rad |
| Fractional backscattering | $u_\lambda$ | - |
| Below-surface remote-sensing reflectance | $r_{rs,\lambda}$ | sr$^{-1}$ |
| Above-surface remote-sensing reflectance | $R_{rs,\lambda}$ | sr$^{-1}$ |
| Thickness of model layer | $h$ | m |
| Optical depth weighting function | $w_{z,\lambda}$ | |
| Vertical attenuation coefficient | $K_\lambda$ | m$^{-1}$ |
| Total absorption coefficient | $a_{T,\lambda}$ | m$^{-1}$ |
| Total scattering coefficient | $b_{T,\lambda}$ | m$^{-1}$ |
| Absorption cross-section | $\alpha_\lambda$ | m$^2$ cell$^{-1}$ |
| Concentration of cells | $n$ | cell m$^{-3}$ |

**Table 2.** State and derived variables in the water column optical model.

assumed the optical properties of the detritus was the same as the optical properties in Gladstone Harbour, although open ocean studies have used a detritial absorption that is more like CDOM (Dutkiewicz et al., 2015).





The absorption due to calcite-based NAP is given by:

$$a_{NAP_{\text{CaCO}_3},\lambda} = c_1 NAP_{\text{CaCO}_3} \tag{10}$$

where $c_1$ is the mass-specific, spectrally-resolved absorption coefficient determine from laboratory experiments (Fig. 9). The absorption due to non-calcite NAPs, $NAP_{\text{non}-\text{CaCO}_3}$, combined with detritus, is given by:

$$a_{NAP_{\text{non}-\text{CaCO}_3},\lambda} = c_2 NAP_{\text{non}-\text{CaCO}_3} + \left( \frac{550}{30}\frac{12}{14}D_{Atk} + \frac{106}{16}\frac{12}{14}D_{Red} + D_C \right)/10^6 \tag{11}$$

where $c_2$ is the mass-specific, spectrally-resolved absorption coefficient determine from field measurements (Fig. 10), $NAP_{\text{non}-\text{CaCO}_3}$ is quantified in kg m$^{-3}$, $D_{Atk}$ and $D_{Red}$ are quantified in mg N m$^{-3}$ and $D_C$ is quantified in mg C m$^{-3}$.

*Total absorption.* The total absorption, $a_{T,\lambda}$, is given by:

$$a_{T,\lambda} = a_{w,\lambda} + a_{NAP_{\text{non}-\text{CaCO}_3},\lambda} + a_{NAP_{\text{CaCO}_3},\lambda} + a_{CDOM,\lambda} + \sum_{x=1}^{N} n_x \alpha_{x,\lambda} \tag{12}$$

where $a_{w,\lambda}$ is clear water absorption (Fig. 11) and $N$ is the number of phytoplankton classes (see Table 4).

*Scattering.* The total scattering coefficient is given by

$$b_{T,\lambda} = b_{w,\lambda} + c_1 NAP_{\text{non}-\text{CaCO}_3} + c_2 NAP_{\text{CaCO}_3} + b_{phy,\lambda} \sum_{x=1}^{N} n_x c_{i,x} V_x \tag{13}$$

where $NAP$ is the concentration of non-algal particulates, $b_{w,\lambda}$ is the scattering coefficient due to clear water (Fig. 11), $c_1$ and $c_2$ are the spectrally-resolved, mass-specific coefficients (Figs. 9 & 10) and phytoplankton scattering is the product of the chlorophyll-specific phytoplankton scattering coefficient, $b_{phy,\lambda}$, and the water column chlorophyll concentration of all classes, $\sum n_x c_{i,x} V_x$ (where $c_i$ is the chlorophyll concentration in the cell, and $V$ is the cell volume). The value for $b_{phy,\lambda}$ is set to 0.2 (mg Chl $a$ m$^{-2}$)$^{-1}$ for all wavelengths, a typical value for marine phytoplankton (Kirk, 1994). For more details see Baird et al. (2007).

*Backscattering* In addition to the IOPs calculated above, the calculation of remote-sensing reflectance uses a backscattering coefficient, $b_b$, which has a component due to pure seawater, and a component due to algal and non-algal particulates. The backscattering ratio is a coarse resolution representation of the volume scattering function, and is the ratio of the forward and backward scattering.

The backscattering coefficient for clear water is 0.5, a result of isotropic scattering of the water molecule.

The particulate component of backscattering for phytoplankton is strongly related to cell carbon (and therefore cell size) and the number of cells (Vaillancourt et al., 2004):

$$b_{bphy,\lambda}^* = 5 \times 10^{-15} m_C^{1.002} \quad (R^2 = 0.97) \tag{14}$$

where $m_C$ is the carbon content of the cells, here in pg cell$^{-1}$.

For inorganic particles, backscattering can vary between particle mineralogies, size, shape, and at different wavelengths, resulting, with spectrally-varying absorption, in the variety of colours that we see from suspended sediments. Splitting sediment types by mineralogy only, the backscattering ratio for carbonate and non-carbonate particles is given in Table 3.



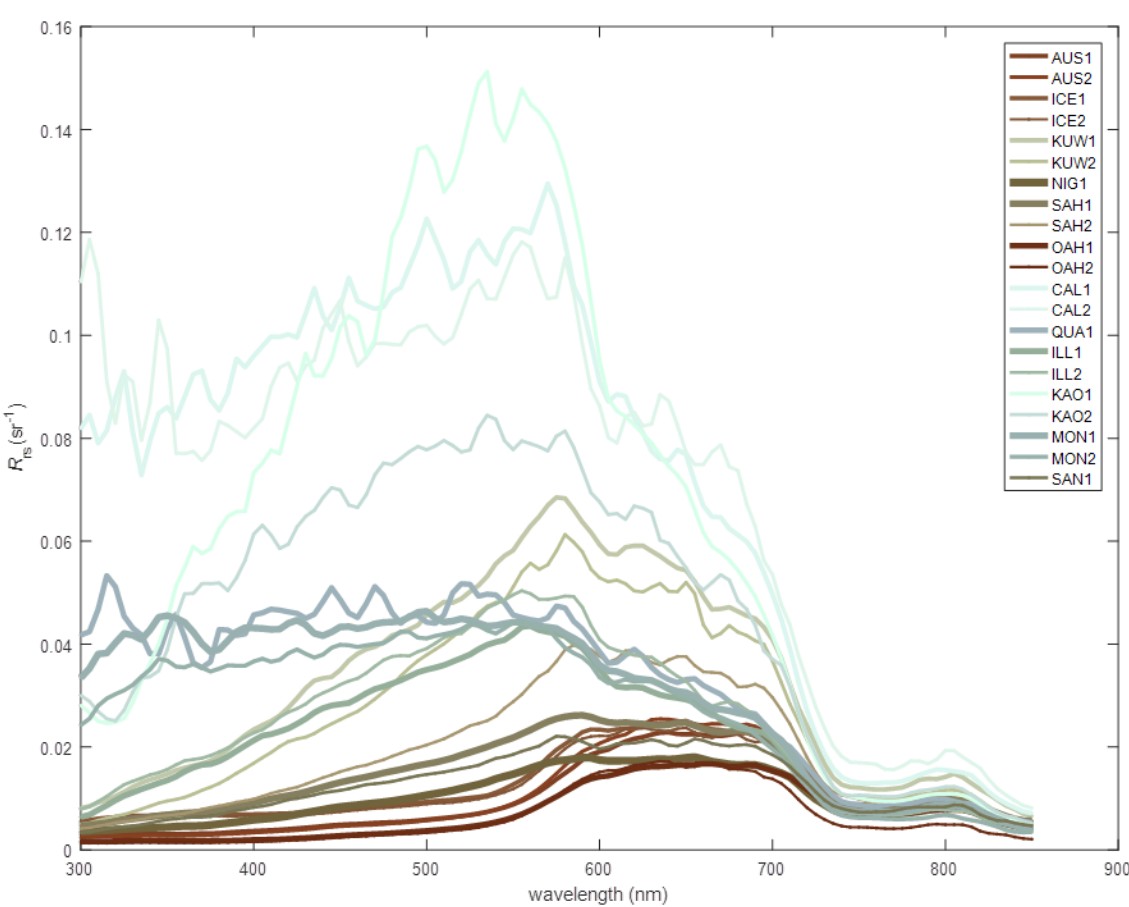

**Figure 9.** The remote-sensing reflectance of the 21 mineral mixtures suspended in water as measured by Stramski et al. (2007). Laboratory measurements of absorption and scattering properties are used to calculated $u$ (Eq. 24). The remote-sensing reflectance is then calculated using Eq. 28, with the line colouring corresponding to that produced by the mineral suspended in clear water as calculated using the MODIS true color algorithm (Gumley et al., 2010). CAL1, with a median particle diameter of 2 $\mu$m, is used for $\text{Mud}_{\text{CaCO}_3}$.





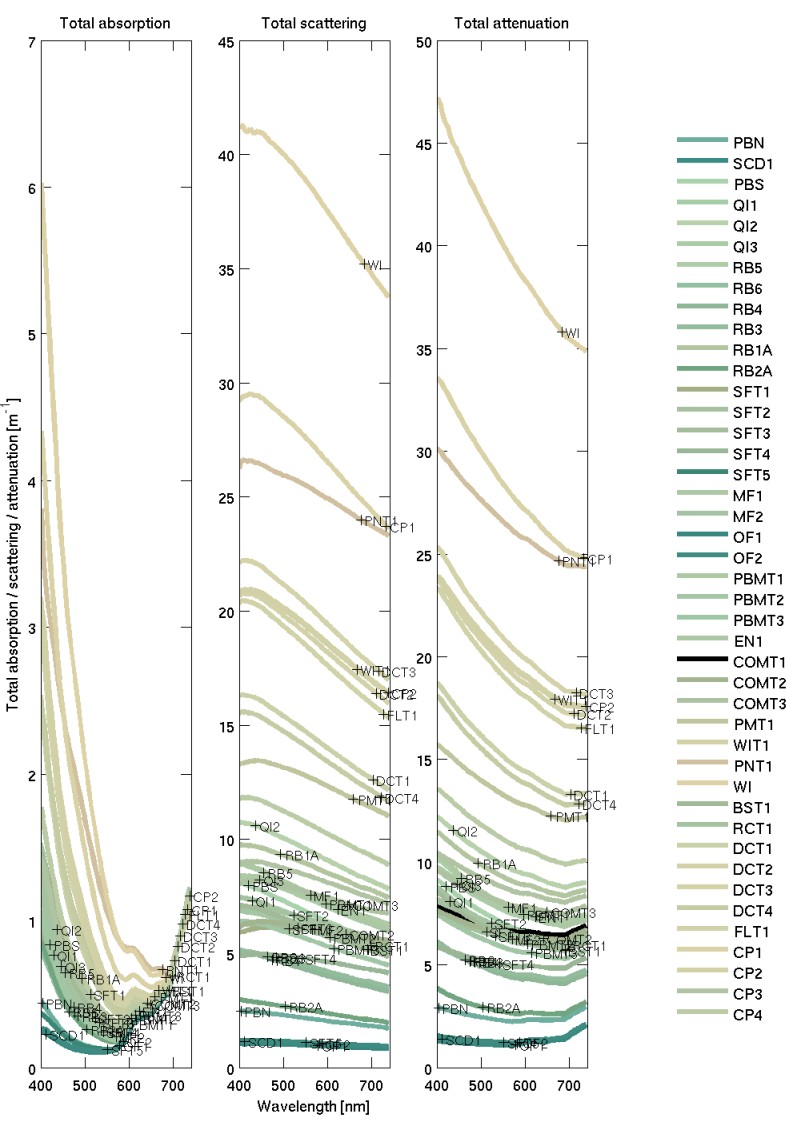

**Figure 10.** Inherent optical properties (total absorption and total scattering) at sample sites in Gladstone Harbour on 13-19 September 2013 (Babcock et al., 2015). The line colour is rendered like Fig. 9. The site labelling is ordered in time, from the first sample collected during neap tides at the top, to the last sample collected at spring tides on the bottom. The IOPs used for the Mud$_{non-CaCO_3}$ end-member is from the WIT site at the centre of the harbour, was dominated by inorganic particles. The measured concentration of NAP at the site was 33.042 mg L$^{-1}$, and is used to calculate mass-specific IOPs.





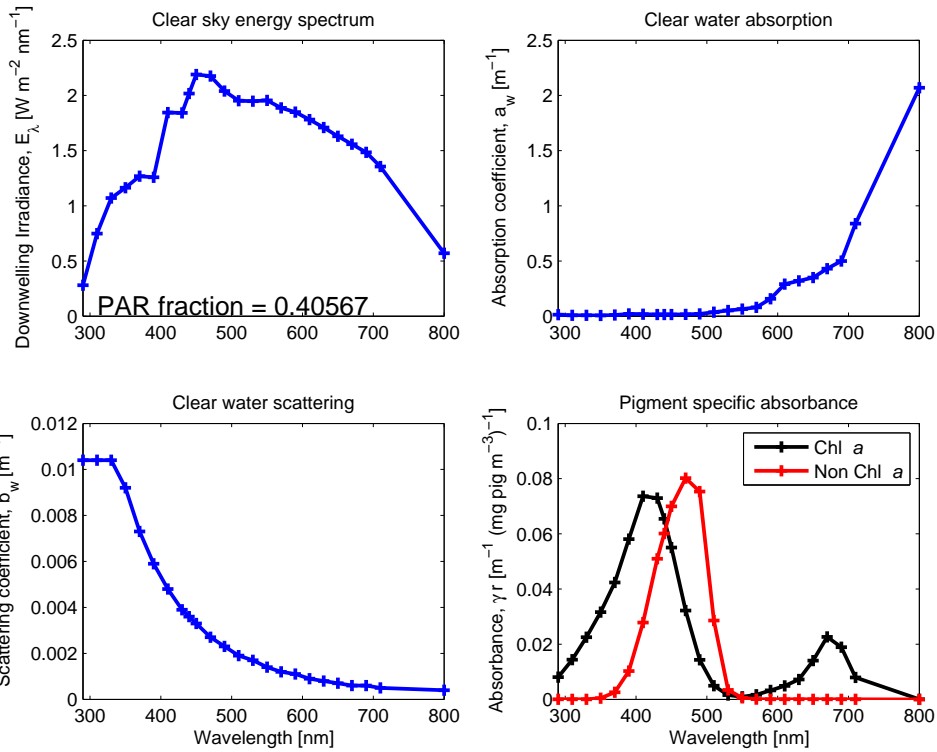

**Figure 11.** Spectrally-resolved energy distribution of sunlight, clear water absorption, and clear water scattering (Smith and Baker, 1981). The fraction of solar radiation between 400 and 700 nm for clear sky irradiance at the particular spectral resolution is given in the top left panel. The centre of each waveband used in the model simulations is identified by a cross on each curve. The bottom right panel shows the pigment-specific absorbance of Chl *a* and generic photosynthetic carotenoids (Ficek et al., 2004) that were used in earlier versions of this model (Baird et al., 2016b) before the mass-specific absorption coefficients of multiple accessory pigments was implemented (Figs. 6, 7 & 8).

|  | Wavelength [nm] | | | | | | | | |
|---|---|---|---|---|---|---|---|---|---|
|  | 412.0 | 440.0 | 488.0 | 510.0 | 532.0 | 595.0 | 650.0 | 676.0 | 715.0 |
| Carbonate | 0.0209 | 0.0214 | 0.0224 | 0.0244 | 0.0216 | 0.0201 | 0.0181 | 0.0170 | 0.0164 |
| Terrestrial | 0.0028 | 0.0119 | 0.0175 | 0.0138 | 0.0128 | 0.0134 | 0.0048 | 0.0076 | 0.0113 |

**Table 3.** Particulate backscattering ratio for carbonate and non-carbonate minerals based on samples at Lucinda Jetty Coastal Observatory, a site at the interface on carbonate and terrestrial bottom sediment (Soja-Woźniak et al., submitted).

The backscatter due to phytoplankton is approximately 0.02. To account for a greater backscattering ratio, and therefore backscatter, at low wavelengths (Fig. 4 of Vaillancourt et al. (2004)), we linearly increased the backscatter ratio from 0.02 at 555 nm to 0.04 at 470 nm. Above and below 555 nm and 470 nm respectively the backscatter ratio remained constant.





The total backscatter then becomes:

$$b_{b,\lambda} = \tilde{b}_w b_{w,\lambda} + b^*_{bphy,\lambda} n + \tilde{b}_{b,NAP_{non-CaCO_3},\lambda} c_1 NAP_{\text{non-CaCO}_3} + \tilde{b}_{b,NAP_{CaCO_3},\lambda} c_2 NAP_{\text{non-CaCO}_3} NAP_{\text{CaCO}_3} \tag{15}$$

where the backscatter ratio of pure seawater, $\tilde{b}_w$, is 0.5, $n$ is the concentration of cells, and for particulate matter (NAP and detritus), $\tilde{b}_{b,NAP,\lambda}$, is variable (Table 3) and the coefficients $c_1$ and $c_2$ come from the total scattering equations above.

### 3.2.2 Apparent optical properties (AOPs)

The optical model is forced with the downwelling short wave radiation just above the sea surface, based on remotely-sensed cloud fraction observations and calculations of top-of-the-atmosphere clear sky irradiance and solar angle. The calculation of downwelling radiation and surface albedo (a function of solar elevation and cloud cover) is detailed in the hydrodynamic scientific description (`https://research.csiro.au/cem/software/ems/ems-documentation/`, Sec 9.1.1).

The downwelling irradiance just above the water interface is split into wavebands using the weighting for clear sky irradiance (Fig. 11). Snell's law is used to calculate the azimuth angle of the mean light path through the water, $\theta_{sw}$, as calculated from the atmospheric azimuth angle, $\theta_{air}$, and the refraction of light at the air/water interface (Kirk, 1994):

$$\frac{\sin \theta_{air}}{\sin \theta_{sw}} = 1.33 \tag{16}$$

*Calculation of in-water light field.* Given the IOPs determined above, the exact solution for AOPs would require a radiative transfer model (Mobley, 1994), which is too computationally-expensive for a complex ecosystem model such as developed here. Instead, the in-water light field is solved for using empirical approximations of the relationship between IOPs and AOPs (Kirk, 1991; Mobley, 1994).

The vertical attenuation coefficient at wavelength $\lambda$ when considering absorption and scattering, $K_\lambda$, is given by:

$$K_\lambda = \frac{a_{T,\lambda}}{\cos \theta_{sw}} \sqrt{1 + (g_i \cos \theta_{sw} - g_{ii}) \frac{b_{T,\lambda}}{a_{T,\lambda}}} \tag{17}$$

The term outside the square root quantifies the effect of absorption, where $a_{T,\lambda}$ is the total absorption. The term within the square root of Eq. 17 represents scattering as an extended pathlength through the water column, where $g_i$ and $g_{ii}$ are empirical constants and take values of 0.402 and 0.180 respectively. The values of $g_i$ and $g_{ii}$ depend on the average cosine of scattering. For filtered water with scattering only due to water molecules, the values of $g_i$ and $g_{ii}$ are quite different to natural waters. But for waters ranging from coastal to open ocean, the average cosine of scattering varies by only a small amount (0.86 - 0.95, Kirk (1991)), and thus uncertainties in $g_i$ and $g_{ii}$ do not strongly affect $K_\lambda$.

The downwelling irradiance at wavelength $\lambda$ at the bottom of a layer $h$ thick, $E_{d,\lambda,bot}$, is given by:

$$E_{d,bot,\lambda} = E_{d,top,\lambda} e^{-K_\lambda h} \tag{18}$$

where $E_{d,top,\lambda}$ is the downwelling irradiance at wavelength $\lambda$ at the top of the layer and $K_\lambda$ is the vertical attenuation coefficient at wavelength $\lambda$, a result of both absorption and scattering processes.





Assuming a constant attenuation rate within the layer, the average downwelling irradiance at wavelength $\lambda$, $E_{d,\lambda}$, is given by:

$$E_{d,\lambda} = \frac{1}{h} \int_{bot}^{top} E_{d,z,\lambda} e^{-K_\lambda z} dz = \frac{E_{d,top,\lambda} - E_{d,bot,\lambda}}{K_\lambda h} \tag{19}$$

We can now calculate the scalar irradiance, $E_o$, for the calculation of absorbing components, from downwelling irradiance, $E_d$. The light absorbed within a layer must balance the difference in downwelling irradiance from the top and bottom of the layer (since scattering in this model only increases the pathlength of light), thus:

$$E_{o,\lambda} a_{T,\lambda} h = E_{d,top,\lambda} - E_{d,bot,\lambda} = E_{d,\lambda} K_\lambda h \tag{20}$$

Canceling $h$, and using Eq. 17, the scalar irradiance as a function of downwelling irradiance is given by:

$$E_{o,\lambda} = \frac{E_{d,\lambda}}{\cos\theta_{sw}} \sqrt{1 + (g_i \cos\theta_{sw} - g_{ii}) \frac{b_{T,\lambda}}{a_{T,\lambda}}} \tag{21}$$

This correction conserves photons within the layer, although it is only as a good as the original approximation of the impact of scattering and azimuth angle on vertical attenuation (Eq. 17).

*Vertical attenuation of heat*. The vertical attenuation of heat is given by:

$$K_{heat} = -\int \frac{1}{E_{d,z,\lambda}} \frac{\partial E_{d,z,\lambda}}{\partial z} d\lambda \tag{22}$$

and the local heating by:

$$\frac{\partial T}{\partial t} = -\frac{1}{\rho c_p} \int \frac{\partial E_{d,\lambda}}{\partial z} d\lambda \tag{23}$$

where $T$ is temperature, $\rho$ is the density of water, and $c_p = 4.1876$ J m$^{-3}$ K$^{-1}$ is the specific heat of water. This calculation does not feed back to the hydrodynamic model.

### 3.2.3 Remote-sensing reflectance

The ratio of the backscattering coefficient to the sum of backscattering and absorption coefficients for the water column, $u_\lambda$, is:

$$u_\lambda = \sum_{z'=1}^{N} \frac{w_{\lambda,z'} b_{b,\lambda,z'}}{a_{\lambda,z'} + b_{b,\lambda,z'}} \tag{24}$$

where $w_{\lambda,z'}$ is a weighting representing the component of the remote-sensing reflectance due to the absorption and scattering in layer number $z'$, and $N$ is the number of layers.





The weighting fraction in layer $z'$ is given by:

$$w_{\lambda,z'} = \frac{1}{z_1 - z_0}\left(\int_0^{z_1} \exp\left(-2K_{\lambda,z}\right)dz - \int_0^{z_0} \exp\left(-2K_{\lambda,z}\right)dz\right) \tag{25}$$

$$= \frac{1}{z_1 - z_0}\int_{z_0}^{z_1} \exp\left(-2K_{\lambda,z}\right)dz \tag{26}$$

where $K_\lambda$ is the vertical attenuation coefficient at wavelength $\lambda$ described above, and the factor of 2 accounts for the pathlength of both downwelling and upwelling light. The integral of $w_{\lambda,z}$ to infinite depth is 1. In areas where light reaches the bottom, the integral of $w_{\lambda,z}$ to the bottom is less than one, and bottom reflectance is important (see Sec. 5.2.2).

The below-surface remote-sensing reflectance, $r_{rs}$, is given by:

$$r_{rs,\lambda} = g_0 u_\lambda + g_1 u_\lambda^2 \tag{27}$$

where $g_0 = 0.0895$ (close to $1/4\pi$) and $g_1 = 0.1247$ are empirical constants for the nadir-view in oceanic waters (Lee et al., 2002b; Brando et al., 2012), and these constants result in a change of units from the unitless $u$ to a per unit of solid angle, $\text{sr}^{-1}$, quantity $r_{rs,\lambda}$.

The above-surface remote-sensing reflectance, through rearranging Lee et al. (2002b), is given by:

$$R_{rs,\lambda} = \frac{0.52 r_{rs,\lambda}}{1 - 1.7 r_{rs,\lambda}} \tag{28}$$

At open ocean values, $R_{rs} \sim 0.06u$ $\text{sr}^{-1}$. Thus if total backscattering and absorption are approximately equal, $u = 0.5$ and $R_{rs} \sim 0.03$ $\text{sr}^{-1}$.

### 3.3  Microalgae

The model contains four function groups of suspended microalgae: small and large phytoplankton, microphytobenthos and *Trichodesmium*. The growth model for each of the functional groups is identical and explained below. The differences in the

ecological interactions of the four functional groups are summarised in Table 4.

### 3.3.1  Growth

The model considers the diffusion-limited supply of dissolved inorganic nutrients (N and P) and the absorption of light, delivering N, P and fixed C to the internal reserves of the cell (see Fig. 3 in Baird et al. (2018)). Nitrogen and phosphorus are taken directly into the reserves, but carbon is first fixed through photosynthesis (Kirk, 1994):

$$106\text{CO}_2 + 212\text{H}_2\text{O} \xrightarrow{1060\text{ photons}} 106\text{CH}_2\text{O} + 106\text{H}_2\text{O} + 106\text{O}_2 \tag{29}$$





| | small phyto. | large phyto. | benthic phyto. | *Trichodesmium* |
|---|---|---|---|---|
| Radius ($\mu$m) | 1 | 4 | 10 | 5 |
| [a]Maximum growth rate (d$^{-1}$) | 1.6 | 1.4 | 0.839 | 0.2 |
| Sink rate (m d$^{-1}$) | | | | variable |
| Surface sediment growth | × | × | √ | × |
| Nitrogen fixation | × | × | × | √ |
| Water column mort. | √ | √ | × | √ |
| Sediment mort. | √ | √ | √ | √ |

**Table 4.** Traits of suspended microalgae cells.[a] At $T_{ref} = 20°$C.

The internal reserves of C, N, and P are consumed to form structural material at the Redfield ratio (Redfield et al., 1963):

$$106\text{CH}_2\text{O} + 16\text{NH}_4^+ + \text{PO}_4^{3-} + 16\text{H}_2\text{O} \tag{30}$$
$$\longrightarrow \quad (\text{CH}_2\text{O})_{106}(\text{NH}_3)_{16}\text{H}_3\text{PO}_4 + 13\text{H}^+$$

5  where we have represented nitrogen as ammonia (NH$_4$) in Eq. 30. When the nitrogen source to the cell is nitrate, NO$_3$, it is assumed to lose its oxygen at the cell wall (Sec. 6.1). The growth rate of microalgae is given by the maximum growth rate, $\mu^{max}$, multiplied by the normalised reserves, $R^*$, of each of N, P and C:

$$\mu = \mu^{max} R_N^* R_P^* R_C^* \tag{31}$$

The mass of the reserves (and therefore the total C:N:P:Chl *a* ratio) of the cell depends on the interaction of the supply and 10 consumption rates (Fig. 12). When consumption exceeds supply, and the supply rates are non-Redfield, the normalised internal reserves of the non-limiting nutrients approach 1 while the limiting nutrient becomes depleted. Thus the model behaves like a 'Law of the Minimum' growth model, except during fast changes in nutrient supply rates.

The molar ratio of a cell is given by:

$$\text{C} : \text{N} : \text{P} = 106(1 + R_C^*) : 16(1 + R_N^*) : 1 + R_P^* \tag{32}$$

15 ### 3.3.2 Nutrient uptake

The diffusion-limited nutrient uptake to a single phytoplankton cell, $J$, is given by:

$$J = \psi D (C_b - C_w) \tag{33}$$

where $\psi$ is the diffusion shape factor ($= 4\pi r$ for a sphere), $D$ is the molecular diffusivity of the nutrient, $C_b$ is the average extracellular nutrient concentration, and $C_w$ is the concentration at the wall of the cell. The diffusion shape factor is determined 20 by equating the divergence of the gradient of the concentration field to zero ($\nabla^2 C = 0$).





| Variable | Symbol | Units |
|---|---|---|
| Scalar irradiance | $E_o$ | W m$^{-2}$ |
| Dissolved inorganic nitrogen (DIN) | $N$ | mg N m$^{-3}$ |
| Dissolved inorganic phosphorus (DIP) | $P$ | mg P m$^{-3}$ |
| Dissolved inorganic carbon (DIC) | $DIC$ | mg C m$^{-3}$ |
| Dissolved oxygen | $[O_2]$ | mg O m$^{-3}$ |
| Reserves of nitrogen | $R_N$ | mg N cell$^{-1}$ |
| Reserves of phosphorus | $R_P$ | mg P cell$^{-1}$ |
| Reserves of carbon | $R_C$ | mg C cell$^{-1}$ |
| Maximum reserves of nitrogen | $R_N^{\mathrm{max}}$ | mg N cell$^{-1}$ |
| Maximum reserves of phosphorus | $R_P^{\mathrm{max}}$ | mg P cell$^{-1}$ |
| Maximum reserves of carbon | $R_C^{\mathrm{max}}$ | mg C cell$^{-1}$ |
| Normalised reserves of nitrogen | $R_N^* \equiv R_N/R_N^{\mathrm{max}}$ | - |
| Normalised reserves of phosphorus | $R_P^* \equiv R_P/R_P^{\mathrm{max}}$ | - |
| Normalised reserves of carbon | $R_C^* \equiv R_C/R_C^{\mathrm{max}}$ | - |
| Intracellular Chl $a$ concentration | $c_i$ | mg m$^{-3}$ |
| Structural phytoplankton biomass | $B$ | mg N m$^{-3}$ |
| Absorption cross-section | $\alpha$ | m$^2$ cell$^{-1}$ |
| Diffusion shape factor | $\psi$ | m cell$^{-1}$ |
| Wavelength | $\lambda$ | nm |
| Maximum Chl $a$ synthesis rate | $k_{\mathrm{Chl}}^{\mathrm{max}}$ | mg Chl m$^{-3}$ d$^{-1}$ |
| Photon absorption-weighted opaqueness | $\overline{\Theta}$ | - |
| Non-dimensional absorbance | $\rho_\lambda = \gamma_\lambda c_i r$ | - |

**Table 5.** State and derived variables for the microalgae growth model. DIN is given by the sum of nitrate and ammonia concentrations, $[NO_3]+[NH_4]$.

A semi-empirical correction to Eq. 33, to account for fluid motion around the cell, and the calculation of non-spherical diffusion shape factors, has been applied in earlier work (Baird and Emsley, 1999). For the purposes of biogeochemical modelling these uncertain corrections for small scale turbulence and non-spherical shape are not quantitatively important, and have not been pursued here.

5     Numerous studies have considered diffusion-limited transport to the cell surface at low nutrient concentrations saturating to a physiologically-limited nutrient uptake from the cell wall (Hill and Whittingham, 1955; Pasciak and Gavis, 1975; Mann and Lazier, 2006) at higher concentrations. The physiological limitation is typically considered using a Michaelis-Menton type equation. Here we simply consider the diffusion-limited uptake to be saturated by the filling-up of reserves, $(1 - R^*)$. Thus,



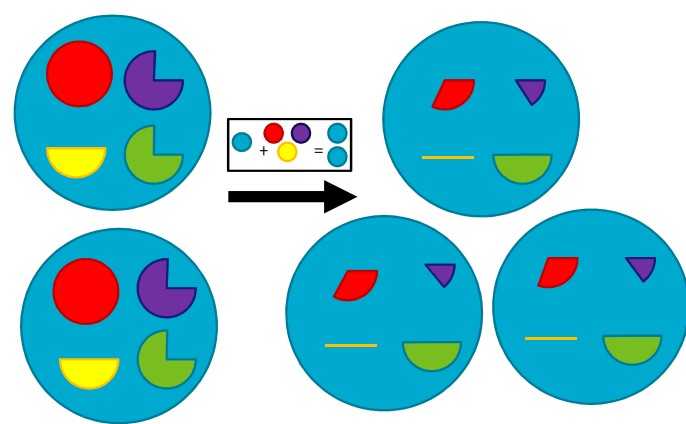

**Figure 12.** Schematic of the process of microalgae growth from internal reserves. Blue circle - structural material; Red pie - nitrogen reserves; Purple pie - phosphorus reserves; Yellow pie - carbon reserves; Green pie - pigment content. Here a circular pie has a value of 1, representing the normalised reserve (a value between 0 and 1). The box shows that to generate structural material for an additional cell requires the equivalent of 100 % internal reserves of carbon, nitrogen and phosphorus of one cell. This figure shows the discrete growth of 2 cells to 3, requiring both the generation of new structural material from reserves and the reserves being diluted as a result of the number of cells in which they are divided increasing from 2 to 3. Thus the internal reserves for nitrogen after the population increases from 2 to 3 is given by: two from the initial 2 cells, minus one for building structural material of the new cell, shared across the 3 offspring, to give 1/3. The same logic applies to carbon and phosphorus reserves, with phosphorus reserves being reduced to 1/6, and carbon reserves being exhausted. In contrast, pigment is not required for structural material so the only reduction is through dilution; the 3/4 content of 2 cells is shared among 3 cells to equal 1/2 in the 3 cells. This schematic shows one limitation of a population-style model whereby reserves are 'shared' across the population (as opposed to individual based modelling, Beckmann and Hense (2004)). A proof of the conservation of mass for this scheme, including under mixing of populations of suspended microalgae, is given in Baird et al. (2004). The model equations also include terms affecting internal reserves through nutrient uptake, light absorption, respiration and mortality that are not shown in this simple schematic.

nutrient uptake is given by:

$$J = \psi D C_b (1 - R^*) \tag{34}$$

where $R^*$ is the reserve of the nutrient being considered. As shown later when considering preferential ammonia uptake, under extreme limitation relative to other nutrients, $R^*$ approaches 0, and uptake approaches the diffusion limitation.





### 3.3.3 Light capture and chlorophyll synthesis

Light absorption by microalgae cells has already been considered above Eq. 6. The same absorption cross-section, $\alpha$, is used to calculate the capture of photons:

$$\frac{\partial R_C^*}{\partial t} = (1 - R_C^*) \frac{(10^9 hc)^{-1}}{A_V} \int \alpha_\lambda E_{o,\lambda} \lambda \, d\lambda \tag{35}$$

where $(1 - R_C^*)$ accounts for the reduced capture of photons as the reserves becomes saturated, and $\frac{(10^9 hc)^{-1}}{A_V}$ converts from energy to photons. The absorption cross-section is a function of intracellular pigment concentration, which is a dynamic variable determined below. While a drop-off of photosynthesis occurs as the carbon reserves become replete, this formulation does not consider photoinhibition due to photooxidation, although it has been considered elsewhere for zooxanthallae (Baird et al., 2018).

The dynamic C:Chl component determines the rate of synthesis of pigment based on the incremental benefit of adding pigment to the rate of photosynthesis. This calculation includes both the reduced benefit when carbon reserves are replete, $(1 - R_C^*)$, and the reduced benefit due to self-shading, $\chi$. The factor $\chi$ is calculated for the derivative of the absorption cross-section per unit projected area, $\alpha/PA$, with non-dimensional group $\rho = \gamma c_i r$. For a sphere of radius $r$ (Baird et al., 2013):

$$\frac{1}{PA} \frac{\partial \alpha}{\partial \rho} = \frac{1 - e^{-2\rho}(2\rho^2 + 2\rho + 1)}{\rho^3} = \chi \tag{36}$$

where $\chi$ represents the area-specific incremental rate of change of absorption with $\rho$. The rate of chlorophyll synthesis is given by:

$$\frac{\partial c_i}{\partial t} = k_{\text{Chl}}^{\max}(1 - R_C^*)\overline{\chi} \quad \text{if } \mathrm{C:Chl} > \theta_{min} \tag{37}$$

where $k_{\text{Chl}}^{\max}$ is the maximum rate of synthesis and $\theta_{min}$ is the minimum C:Chl ratio. Below $\theta_{min}$, pigment synthesis is zero.
Both self-shading, and the rate of photosynthesis itself, are based on photon absorption rather than energy absorption (Table 6), as experimentally shown in Nielsen and Sakshaug (1993).

For each phytoplankton type the model considers multiple pigments with distinct absorbance spectra. The model needs to represent all photo-absorbing pigments as the C:Chl model calculates the pigment concentration based on that required to maximise photosynthesis. If only Chl $a$ was represented, the model would predict a Chl $a$ concentration that was accounting for
the absorption of Chl $a$ and the auxiliary pigments, thus over-predicting the Chl $a$ concentration when compared to observations. Thus the Chl-a predicted by the model is like a HPLC-determined Chl-a concentration, and not the sum of the photosynthetic pigments.

The state variables, equations and parameter values are listed in Tables 5, 6 and 7 respectively. The equations in Table 5 described nitrogen uptake from the DIN pool, where the partitioning between nitrate and ammonia due to preferential ammonia
uptake is described in Sec. 6.1.





### 3.3.4 Carbon fixation / respiration

When photons are captured (photosynthesis) there is an increase in reserves of carbon, $k_I(1 - R_C^*)$, and an accompanying uptake of dissolved inorganic carbon, $\frac{106}{1060}12k_I(1 - R_C^*)$, and release of oxygen, $\frac{106}{1060}32k_I(1 - R_C^*)$, per cell to the water column (Table 6).

Additionally, there is a basal respiration, representing a constant cost of cell maintenance. The loss of internal reserves, $\mu_B^{\max}m_C\phi R_C^*$, results in a gain of water column dissolved inorganic carbon per cell, $\frac{106}{1060}\frac{12}{14}\mu_B^{\max}\phi R_C^*$, as well as a loss in water column dissolved oxygen per cell, $\frac{106}{1060}\frac{32}{14}\mu_B^{\max}\phi R_C^*$ (Table 6). The loss in water column dissolved oxygen per cell represents an instantaneous respiration of the fixed carbon of the reserves. Basal respiration decreases internal reserves, and therefore growth rate, but does not directly lead to cell mortality at zero carbon reserves. Implicit in this scheme is that the
basal cost is higher when the cell has more carbon reserves, $R_C^*$.

A linear mortality term, resulting in the loss of structural material and carbon reserves, is considered later.

### 3.3.5 Conservation of mass of microalgae model

The conservation of mass during transport, growth and mortality is proven in Baird et al. (2004). Briefly, for microalgal growth, total concentration of nitrogen in microalgae cells is given by $B + BR_N^*$. For conservation of mass, the time derivatives must
equate to zero:

$$\frac{\partial B}{\partial t} + \frac{\partial\left(R_N B/R_N^{max}\right)}{\partial t} = 0. \tag{38}$$

using the product rule to differentiate the second term on the LHS:

$$\frac{\partial B}{\partial t} + \frac{\partial B}{\partial t}\frac{R_N}{R_N^{max}} + \frac{B}{R_N^{max}}\frac{\partial R_N}{\partial t} = 0 \tag{39}$$

Where:

$$\frac{\partial B}{\partial t} = +\mu_B^{max}R_C^*R_N^*R_P^*B \tag{40}$$

$$\frac{\partial B}{\partial t}\frac{R_N}{R_N^{max}} = +\mu_B^{max}R_C^*R_N^*R_P^*B\frac{R_N}{R_N^{max}} \tag{41}$$

$$\frac{B}{R_N^{max}}\frac{\partial R_N}{\partial t} = -B(1 + R_N^*)\mu_B^{max}R_C^*R_N^*R_P^*\frac{R_N}{R_N^{max}} \tag{42}$$

Thus demonstrating conservation of mass when $m_{B,N} = R_N^{max}$, as used here.

Earlier published versions of the microalgae model are described with multiple nutrient limitation (Baird et al., 2001), with variable C:N ratios (Wild-Allen et al., 2010) and variable C:Chl ratios (Baird et al., 2013). Further, demonstration of the conservation of mass during transport is given in Baird et al. (2004). Here the microalgae model is presented with variable





$$\frac{\partial N}{\partial t} = -\psi D_N N (1 - R_N^*)(B/m_N) \tag{43}$$

$$\frac{\partial P}{\partial t} = -\psi D_P P (1 - R_P^*)(B/m_N) \tag{44}$$

$$\frac{\partial DIC}{\partial t} = -\left(\frac{106}{1060}12 k_I (1 - R_C^*) - \frac{106}{16}\frac{12}{14}\mu_B^{\max}\phi R_C^*\right)(B/m_N) \tag{45}$$

$$\frac{\partial [O_2]}{\partial t} = \left(\frac{106}{1060}32 k_I (1 - R_C^*) - \frac{106}{16}\frac{32}{14}\mu_B^{\max}\phi R_C^*\right)(B/m_N) \tag{46}$$

$$\frac{\partial R_N}{\partial t} = \psi D_N N (1 - R_N^*) - \mu_B^{\max}(m_N + R_N) R_P^* R_N^* R_C^* \tag{47}$$

$$\frac{\partial R_P}{\partial t} = \psi D_P P (1 - R_P^*) - \mu_B^{\max}(m_P + R_P) R_P^* R_N^* R_C^* \tag{48}$$

$$\frac{\partial R_C}{\partial t} = k_I (1 - R_C^*) - \mu_B^{\max}(m_C + R_C) R_P^* R_N^* R_C^* - \mu_B^{\max}\phi m_C R_C^* \tag{49}$$

$$\frac{\partial B}{\partial t} = \mu_B^{\max} R_P^* R_N^* R_C^* B \tag{50}$$

$$\frac{\partial c_i}{\partial t} = k_{\mathrm{Chl}}^{\max}(1 - R_C^*)\overline{\chi} - \mu_P^{\max} R_P^* R_N^* R_C^* c_i \tag{51}$$

$$\psi = 4\pi r \tag{52}$$

$$k_I = \frac{(10^9 hc)^{-1}}{A_V} \int \alpha_\lambda E_{o,\lambda} \lambda \, d\lambda \tag{53}$$

$$\alpha_\lambda = \pi r^2 \left(1 - \frac{2(1 - (1 + 2\rho_\lambda)e^{-2\rho_\lambda})}{4\rho_\lambda^2}\right) \tag{54}$$

$$\overline{\chi} = \int \chi_\lambda E_{o,\lambda} \lambda \, d\lambda \left/ \int E_{o,\lambda} \lambda \, d\lambda \right. \tag{55}$$

$$\chi_\lambda = \frac{1}{\pi r^2}\frac{\partial \alpha_\lambda}{\partial \rho_\lambda} = \frac{1 - e^{-2\rho_\lambda}(2\rho_\lambda^2 + 2\rho_\lambda + 1)}{\rho_\lambda^3} \tag{56}$$

$$\rho_\lambda = \gamma c_i r \tag{57}$$

**Table 6.** Microalgae growth model equations. The term $B/m_N$ is the concentration of cells. The equation for organic matter formation gives the stoichiometric constants; 12 g C mol C$^{-1}$; 32 g O mol O$_2^{-1}$. The equations are for scalar irradiance specified as an energy flux.



| | Symbol | Value |
|---|---|---|
| *Constants* | | |
| Molecular diffusivity of NO$_3$ | $D_N$ | $f(T,S)$ m$^2$ s$^{-1}$ |
| Molecular diffusivity of PO$_4$ | $D_P$ | $f(T,S)$ m$^2$ s$^{-1}$ |
| Speed of light | $c$ | $2.998 \times 10^8$ m s$^{-1}$ |
| Planck constant | $h$ | $6.626 \times 10^{-34}$ J s$^{-1}$ |
| Avogadro constant | $A_V$ | $6.02 \times 10^{23}$ mol$^{-1}$ |
| [a]Pigment-specific absorption coefficient | $\gamma_{\text{pig},\lambda}$ | $f(\text{pig},\lambda)$ m$^{-1}$ $\left(\text{mg m}^{-3}\right)^{-1}$ |
| [d]Minimum C:Chl ratio | $\theta_{min}$ | 20.0 wt/wt |
| *Allometric relationships* | | |
| [b]Carbon content | $m_C$ | $12010 \times 9.14 \times 10^3 V$ mg C cell$^{-1}$ |
| [c]Maximum intracellular Chl *a* concentration | $c_i^{\max}$ | $2.09 \times 10^7 V^{-0.310}$ mg Chl m$^{-3}$ |
| Nitrogen content of phytoplankton | $m_N$ | $\frac{14}{12}\frac{16}{106}m_C$ mg N cell$^{-1}$ |

**Table 7.** Constants and parameter values used in the microalgae model. $V$ is cell volume in $\mu$m$^3$. [a] Figs. 6 7 & 8,[b]Straile (1997),[c]Finkel (2001), Sathyendranath et al. (2009) using HPLC-determination which isolate Chl-a.

C:Chl ratios (with an additional auxiliary pigment), and both nitrogen and phosphorus limitation, and a preference for ammonia uptake when compared to nitrate. The strategy of dynamic supply and consumption rates of elements is a simple version of what is often called Dynamic Energy Budget (DEB) models in the ecological modelling literature (Kooijman, 2010).

### 3.4 Nitrogen-fixing *Trichodesmium*

5 The growth of *Trichodesmium* follows the microalgae growth and C:Chl model above, with the following additional processes of nitrogen fixation and physiological-dependent buoyancy adjustment, as described in Robson et al. (2013). Additional parameter values for *Trichodesmium* are given in Table 8.

### 3.4.1 Nitrogen fixation

Nitrogen fixation occurs when the DIN concentration falls below a critical concentration, $DIN_{crit}$, typically 0.3 to 1.6 $\mu$mol
10 L$^{-1}$ (i.e. 4 to 20 mg N m$^{-3}$, Robson et al. (2013)), at which point *Trichodesmium* produce nitrogenase to allow fixation of N$_2$. It is assumed that nitrogenase becomes available whenever ambient DIN falls below the value of $DIN_{crit}$ and carbon and phosphorus are available to support nitrogen uptake. The rate of change of internal reserves of nitrogen, $R_N$, due to nitrogen fixation if $DIN < DIN_{crit}$ is given by:

$$N_{fix} = \frac{\partial R_N}{\partial t}\Big|_{N_{fix}} = \max\left(4\pi r D_{\text{NO}_3} DIN_{crit} R_P^* R_C^* (1 - R_N^*) - 4\pi r D_{\text{NO}_3} [\text{NO}_3 + \text{NH}_4](1 - R_N^*), 0\right) \tag{58}$$





| | Symbol | Value |
|---|---|---|
| Maximum growth rate | $\mu^{max}$ | 0.2 d$^{-1}$ |
| [b]Ratio of xanthophyll to Chl $a$ | $f_{xan}$ | 0.5 |
| Linear mortality | $m_L$ | 0.10 d$^{-1}$ |
| Quadratic mortality | $m_Q$ | 0.10 d$^{-1}$ (mg N m$^{-3}$)$^{-1}$ |
| Cell radius | $r$ | 5 $\mu$m |
| Colony radius | $r_{col}$ | 5 $\mu$m |
| Max. cell density | $\rho_{max}$ | 1050 kg m$^{-3}$ |
| Min. cell density | $\rho_{min}$ | 900 kg m$^{-3}$ |
| Critical threshold for N fixation | $DIN_{crit}$ | 10 mg N m$^{-3}$ |
| Fraction of energy used for nitrogenase | $f_{nitrogenase}$ | 0.07 |
| Fraction of energy used for N fixation | $f_{Nfix}$ | 0.33 |
| Nitrogen gas in equilibrium with atm. | [N$_2$] | $2 \times 10^4$ mg N m$^{-3}$ |

**Table 8.** Parameter values used in the *Trichodesmium* model (Robson et al., 2013). [b] The major accessory pigments in *Trichodesmium* are the red-ish phycourobilin and phycoerythrobilin (Subramaniam et al., 1999). For simplicity in this model their absorption cross-section is approximated by photosynthetic xanthophyll, which has an absorption peak approximately 10 nm less than the phycourobilin.

where $N_{fix}$ is the rate of nitrogen fixation per cell and $r$ is the radius of the individual cell. Using this formulation, *Trichodesmium* is able to maintain its nitrogen uptake rate at that achieved through diffusion limited uptake at $DIN_{crit}$ even when $DIN$ drops below $DIN_{crit}$, provided phosphorus and carbon reserves, $R_P^*$ and $R_C^*$ respectively, are available.

The energetic cost of nitrogen fixation is represented as a fixed proportion of carbon fixation, $f_{Nfix}$, equivalent to a reduction
5 in quantum efficiency, and as a proportion, $f_{nitrogenase}$, of the nitrogen fixed:

$$\frac{\partial R_C}{\partial t} = -(1 - f_{Nfix})(1 - f_{nitrogenase})k_I \tag{59}$$

where $k_I$ is the rate of photon absorption per cell obtain from the microalgal growth model (Table 6).

### 3.4.2 Buoyancy adjustment

The rate of change of *Trichodesmium* biomass, $B$, as a result of density difference between the cell and the water, is approxi-
10 mated by Stokes' Law:

$$\frac{\partial B}{\partial t} = -\frac{2}{9} \frac{gr_{col}^2}{\mu} (\rho - \rho_w) \frac{\partial B}{\partial z} \tag{60}$$





where $z$ is the distance in the vertical (+ve up), $\mu$ is the dynamic viscosity of water, $g$ is acceleration due to gravity, $r_{col}$ is the equivalent spherical radius of the sinking mass representing a colony radius, $\rho_w$ is the density of water, and $\rho$ is the cell density is given by:

$$\rho = \rho_{min} + R_C^* \left( \rho_{max} - \rho_{min} \right) \tag{61}$$

where $R_C^*$ is the normalised carbon reserves of the cell (see above), and $\rho_{min}$ and $\rho_{max}$ are the densities of the cell when there is no carbon reserves and full carbon reserves respectively. Thus, when light reserves are depleted, the cell is more buoyant, facilitating the retention of *Trichodesmium* in the surface waters.

## 3.5 Water column inorganic chemistry

### 3.5.1 Carbon chemistry

The major pools of dissolved inorganic carbon species in the ocean are $HCO_3^-$, $CO_3^-$, and dissolved $CO_2$, which influence the speciation of $H^+$, and $OH^-$ ions, and therefore pH. The interaction of these ions reaches an equilibrium in seawater within a few tens of seconds (Zeebe and Wolf-Gladrow, 2001). In the biogeochemical model here, where calculation timesteps are of order tens of minutes, it is reasonable to assume that the carbon chemistry system is at equilibrium.

The Ocean-Carbon Cycle Model Intercomparison Project (OCMIP) has developed numerical methods to quantify air-sea carbon fluxes and carbon dioxide system equilibria (Najjar and Orr, 1999). Here we use a modified version of the OCMIP-2 Fortran code developed for MOM4 (GFDL Modular Ocean Model version 4, (Griffies et al., 2004)). The OCMIP procedures quantify the state of the carbon dioxide ($CO_2$) system using two prognostic variables, the concentration of dissolved inorganic carbon, $DIC$, and total alkalinity, $A_T$. The value of these prognostic variables, along with salinity and temperature, are used to calculate the pH and partial pressure of carbon dioxide, $pCO_2$, in the surface waters using a set of governing chemical equations which are solved using a Newton-Raphson method (Najjar and Orr, 1999).

One alteration from the global implementation of the OCMIP scheme is to increase the search space for the iterative scheme from $\pm 0.5$ pH units (appropriate for global models) to $\pm 2.5$. With this change, the OCMIP scheme converges over a broad range of $DIC$ and $A_T$ values (Munhoven, 2013).

For more details see Mongin and Baird (2014); Mongin et al. (2016b).

### 3.5.2 Nitrification

Nitrification is the oxidation of ammonia with oxygen, to form nitrite followed by the rapid oxidation of these nitrites into nitrates. This is represented in a one step processes, with the rate of nitrification given by:

$$\frac{\partial [NH_4]}{\partial t} = -\tau_{nit,wc}[NH_4]\frac{[O_2]}{K_{nit,O} + [O_2]} \tag{68}$$

where the equations and parameter values are defined in Tables 10 and 11.





| Variable | Symbol | Units |
|---|---|---|
| Ammonia concentration | $[NH_4]$ | mg N m$^{-3}$ |
| Water column Dissolved Inorganic Carbon (DIC) | $DIC$ | mg C m$^{-3}$ |
| Water column Dissolved Inorganic Phosphorus (DIP) | $P$ | mg P m$^{-3}$ |
| Water column Particulate Inorganic Phosphorus (PIP) | $PIP$ | mg P m$^{-3}$ |
| Water column Non-Algal Particulates (NAP) | $NAP$ | kg m$^{-3}$ |
| Water column dissolved oxygen concentration | $[O_2]$ | mg O m$^{-3}$ |

**Table 9.** State and derived variables for the water column inorganic chemistry model.

$$NH_4^+ + 2O_2 \longrightarrow NO_3^- + H_2O + 2H^+ \tag{62}$$

$$\frac{\partial [NH_4]}{\partial t} = -\tau_{nit,wc}[NH_4]\frac{[O_2]}{K_{nit,O}+[O_2]} \tag{63}$$

$$\frac{\partial [O_2]}{\partial t} = -2\tau_{nit,wc}[NH_4]\frac{[O_2]}{K_{nit,O}+[O_2]} \tag{64}$$

$$\frac{\partial [NO_3]}{\partial t} = \tau_{nit,wc}[NH_4]\frac{[O_2]}{K_{nit,O}+[O_2]} \tag{65}$$

$$\frac{\partial P}{\partial t} = \tau_{Pabs}\left(\frac{PIP}{k_{Pads,wc}NAP}-\frac{[O_2]P}{K_{O_2,abs}+[O_2]}\right) \tag{66}$$

$$\frac{\partial PIP}{\partial t} = -\tau_{Pabs}\left(\frac{PIP}{k_{Pads,wc}NAP}-\frac{[O_2]P}{K_{O_2,abs}+[O_2]}\right) \tag{67}$$

**Table 10.** Equations for the water column inorganic chemistry.

| Description | Symbol | Units |
|---|---|---|
| Maximum rate of nitrification in the water column | $\tau_{nit,wc}$ | 0.1 d$^{-1}$ |
| Oxygen half-saturation constant for nitrification | $K_{nit,O}$ | 500 mg O m$^{-3}$ |
| Rate of P adsorbed/desorbed equilibrium | $\tau_{Pabs}$ | 0.004 d$^{-1}$ |
| Isothermic const. P adsorption for NAP | $k_{Pads,wc}$ | 30 kg NAP$^{-1}$ |
| Oxygen half-saturation for P adsorption | $K_{O_2,abs}$ | 2000 mg O m$^{-3}$ |

**Table 11.** Constants and parameter values used in the water column inorganic chemistry.

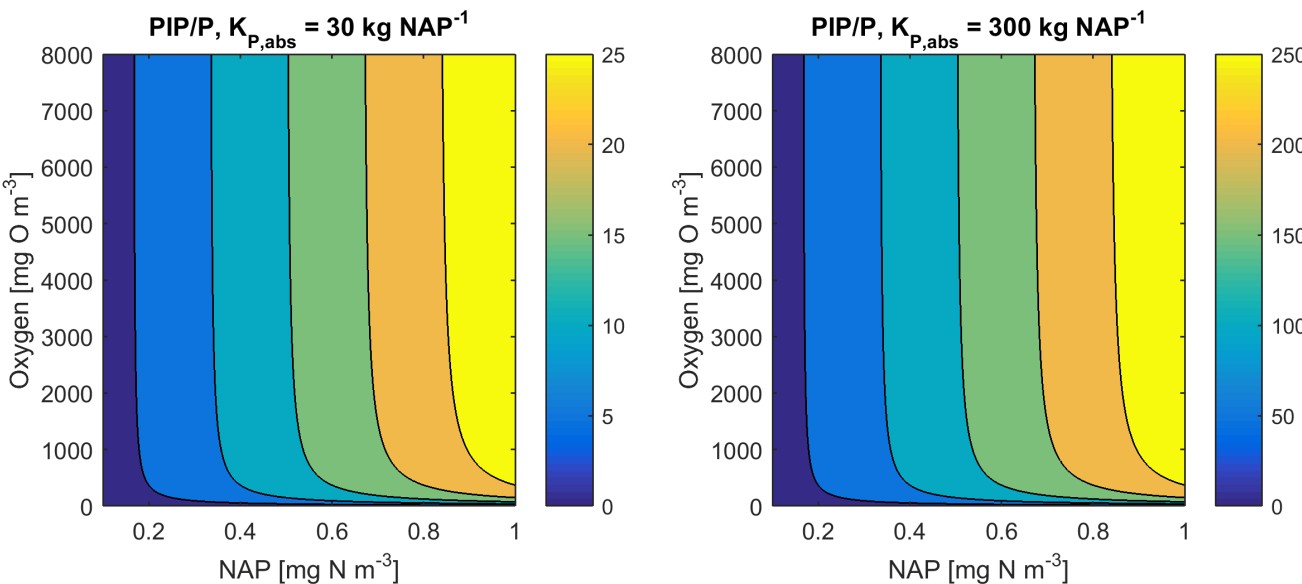

**Figure 13.** Phosphorus adsorption - desorption equilibria, $K_{\mathrm{O_2,abs}}$ = 74 mg O m$^{-3}$.

### 3.5.3 Phosphorus absorption - desorption

The rate of phosphorus desorption from particulates is given by:

$$\frac{\partial P}{\partial t} = \tau_{Pabs} \left( \frac{PIP}{k_{Pads,wc} NAP} - \frac{[\mathrm{O_2}] P}{K_{\mathrm{O_2,abs}} + [\mathrm{O_2}]} \right) = -\frac{\partial PIP}{\partial t} \tag{69}$$

where $[\mathrm{O_2}]$ is the concentration of oxygen, $P$ is the concentration of dissolved inorganic phosphorus, $PIP$ is the concentration

5 of particulate inorganic phosphorus, $NAP$ is the sum of the non-algal inorganic particulate concentrations, and $\tau_{Pabs}$, $k_{Pads,wc}$ and $K_{\mathrm{O_2,abs}}$ are model parameters described in Table 11.

At steady-state, the $PIP$ concentration is given by:

$$PIP = k_{Pads,wc} P \frac{[\mathrm{O_2}]}{K_{\mathrm{O_2,abs}} + [\mathrm{O_2}]} NAP \tag{70}$$

As an example for rivers flowing into the eReefs configuration, $[\mathrm{O_2}]$ = 7411 mg m$^{-3}$ (90% saturation at T = 25, S = 0), $NAP$

10 = 0.231 kg m$^{-3}$, $k_{Pads,wc}$ = 30 kg NAP$^{-1}$, $K_{\mathrm{O_2,abs}}$ = 74 mg O m$^{-3}$, $P$ = 4.2 mg m$^{-3}$, thus the ratio $PIP/DIP$ = 6.86 (see Fig. 13).

Limited available observations of absorption-desorption include from the Johnstone River (Pailles and Moody, 1992) and the GBR (Monbet et al., 2007).





### 3.6 Zooplankton herbivory

The simple food web of the model involves small zooplankton consuming small phytoplankton, and large zooplankton consuming large phytoplankton, microphytobenthos and *Trichodesmium*. For simplicity the state variables and equations are only given for small plankton grazing (Tables 12, 14), but the parameters are given for all grazing terms (Table 13).

The rate of zooplankton grazing is determined by the encounter rate of the predator and all its prey up until the point at which it saturates the growth of the zooplankton, and then it is constant. This is effectively a Hollings Type I grazing response (Gentleman, 2002). Under the condition of multiple prey types, there is no preferential grazing other than that determined by the chance of encounter. The encounter rate is the result of the relative motion brought about by diffusive, shear, and swimming-determined relative velocities (Jackson, 1995; Baird, 2003).

This formulation of grazing, originally proposed by Jackson (1995) but rarely used in biogeochemical modelling, is developed from considering the encounter of individuals, not populations. One particular advantage of formulating the encounter on individuals is that should the number of populations considered in the model change (i.e. an additional phytoplankton class is added), there is no need to re-parameterise. In contrast, almost all biogeochemical models, as typified by Fasham et al. (1990), consider the grazing of populations of plankton, parameterised using a saturating curve constrained by a half saturation con-

stant. Awkwardly, the half saturation constant only has meaning for one particular diet of phytoplankton. This is best illustrated by dividing a single population into two identical populations of half the number, in which case, for the same specification of half-saturation constant, the grazing rate increases. That is:

$$\frac{\mu P}{k+P} \neq \frac{\mu P/2}{k+P/2} + \frac{\mu P/2}{k+P/2} \tag{71}$$

As the zooplankton are grazing on the phytoplankton that contain internal reserves of nutrients an addition flux of dissolved

inorganic nutrients ($gR_N^*$ for nitrogen) is returned to the water column (for more details see Sec. 3.6.1).

### 3.6.1 Conservation of mass in zooplankton grazing

It is important to note that the microalgae model presented above represents internal reserves of nutrients, carbon and chlorophyll as a per cell quantity. Using this representation there are no losses of internal quantities with either grazing or mortality. However the implication of their presence is represented in the ($gR_N^*$) terms (Table 14) that return the reserves to the water

column. These terms represent the fast return of a fraction of phytoplankton nitrogen due to processes like "sloppy eating".

     An alternative and equivalent formulation would be to consider total concentration of microalgal reserves in the water column, then the change in water column concentration of reserves due to mortality (either grazing or natural mortality) must be considered. This alternate representation will not be undertaken here as the above considered equations are fully consistent, but it is worth noting that the numerical solution of the model within the EMS package represents total water

column concentrations of internal reserves, and therefore must include the appropriate loss terms due to mortality.





| Variable | Symbol | Units |
|---|---|---|
| Ammonia concentration | $[\text{NH}_4]$ | mg N m$^{-3}$ |
| Water column dissolved Inorganic Carbon (DIC) | $DIC$ | mg C m$^{-3}$ |
| Water column dissolved Inorganic Phosphorus (DIP) | $P$ | mg P m$^{-3}$ |
| Water column dissolved oxygen concentration | $[\text{O}_2]$ | mg O m$^{-3}$ |
| Reserves of phytoplankton nitrogen | $R_N$ | mg N cell$^{-1}$ |
| Reserves of phytoplankton phosphorus | $R_P$ | mg P cell$^{-1}$ |
| Reserves of phytoplankton carbon | $R_C$ | mmol photon cell$^{-1}$ |
| Maximum reserves of nitrogen | $R_N^{\max}$ | mg N cell$^{-1}$ |
| Maximum reserves of phosphorus | $R_P^{\max}$ | mg P cell$^{-1}$ |
| Maximum reserves of carbon | $R_C^{\max}$ | mmol photon cell$^{-1}$ |
| Normalised reserves of nitrogen | $R_N^* \equiv R_N/R_N^{\max}$ | - |
| Normalised reserves of phosphorus | $R_P^* \equiv R_P/R_P^{\max}$ | - |
| Normalised reserves of carbon | $R_C^* \equiv R_C/R_C^{\max}$ | - |
| Phytoplankton structural biomass | $B$ | mg N m$^{-3}$ |
| Zooplankton biomass | $Z$ | mg N m$^{-3}$ |
| Detritus at the Redfield ratio | $D_{Red}$ | mg N m$^{-3}$ |
| | | |
| Zooplankton grazing rate | $g$ | mg N m$^{-3}$ s$^{-1}$ |
| Encounter rate coefficient due to molecular diffusion | $\phi_{diff}$ | m$^3$ s$^{-1}$ cell Z$^{-1}$ |
| Encounter rate coefficient due to relative motion | $\phi_{rel}$ | m$^3$ s$^{-1}$ cell Z$^{-1}$ |
| Encounter rate coefficient due to turbulent shear | $\phi_{shear}$ | m$^3$ s$^{-1}$ cell Z$^{-1}$ |
| Phytoplankton cell mass | $m_B$ | mg N cell$^{-1}$ |
| Zooplankton cell mass | $m_Z$ | mg N cell$^{-1}$ |

**Table 12.** State and derived variables for the zooplankton grazing. Zooplankton cell mass, $m_Z = 16000 \times 14.01 \times 10.5 V_Z$ mg N cell$^{-1}$, where $V_Z$ is the volume of zooplankton (Hansen et al., 1997).





| Description | Symbol | Small | Large |
|---|---|---|---|
| Maximum growth rate of zooplankton at $T_{ref}$ (d$^{-1}$) | $\mu_Z$ | 4.0 | 1.33 |
| Nominal cell radius of zooplankton ($\mu$m) | $r_Z$ | 5 | 320 |
| Growth efficiency of zooplankton | $E_Z$ | 0.462 | 0.426 |
| Fraction of growth inefficiency lost to detritus | $\gamma_Z$ | 0.5 | 0.5 |
| Swimming velocity ($\mu$m s$^{-1}$) | $U_Z$ | 200 | 3000 |
| | | | |
| Constants | | | |
| Boltzmann's constant | $\kappa$ | $1.38066 \times 10^{-23}$ | J K$^{-1}$ |
| Viscosity | $\nu$ | $10^{-6}$ | m$^2$ s$^{-1}$ |
| Dissipation rate of TKE | $\epsilon$ | $10^{-6}$ | m$^3$ s$^{-1}$ |
| Oxygen half-saturation for aerobic respiration | $K_{OA}$ | 256 | mg O m$^{-3}$ |

**Table 13.** Constants and parameter values used for zooplankton grazing. Dissipation rate of turbulent kinetic energy (TKE) is considered constant.

## 3.7 Zooplankton carnivory

Large zooplankton consume small zooplankton. This process uses similar encounter rate and consumption rate limitations calculated for zooplankton herbivory (Table 14). As zooplankton contain no internal reserves, the equations are simplified from the herbivory case to those listed in Table 15). Assuming that the efficiency of herbivory, $\gamma$, is equal to that of carnivory, and therefore assigned the same parameter, the additional process of carnivory adds no new parameters to the biogeochemical model.

## 3.8 Zooplankton respiration

In the model there is no change in water column oxygen concentration if organic material is exchanged between pools with the same elemental ratio. Thus, when zooplankton plankton consume phytoplankton no oxygen is consumed due to the consumption of phytoplankton structural material ($B_P$). However, the excess carbon reserves represent a pool of fixed carbon, which when released from the phytoplankton must consume oxygen. Further, zooplankton mortality and growth inefficiency results in detrital production, which when remineralised consumes oxygen. Additionally, carbon released to the dissolved inorganic pool during inefficiency grazing on phytoplankton structural material also consumes oxygen. Thus zooplankton respiration is implicitly captured in these associated processes.

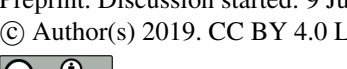



$$\frac{\partial [\mathrm{NH_4}]}{\partial t} = g(1-E)(1-\gamma) + gR_N^* \tag{72}$$

$$\frac{\partial P}{\partial t} = g\frac{1}{16}\frac{31}{14}(1-E)(1-\gamma) + \frac{1}{16}\frac{31}{14}gR_P^* \tag{73}$$

$$\frac{\partial DIC}{\partial t} = g\frac{106}{16}\frac{12}{14}(1-E)(1-\gamma) + \frac{106}{16}\frac{12}{14}gR_C^* \tag{74}$$

$$\frac{\partial B}{\partial t} = -g \tag{75}$$

$$\frac{\partial Z}{\partial t} = Eg \tag{76}$$

$$\frac{\partial D_{Red}}{\partial t} = g(1-E)\gamma \tag{77}$$

$$\frac{\partial [\mathrm{O_2}]}{\partial t} = -\frac{\partial DIC}{\partial t}\frac{32}{12}\frac{[\mathrm{O_2}]}{K_{OA}+[\mathrm{O_2}]} \tag{78}$$

$$g = \min\left[\mu_Z^{max}Z/E, \frac{Z}{m_{Z_L}}(\phi_{diff}+\phi_{rel}+\phi_{shear})B\right] \tag{79}$$

$$\phi = \phi_{diff}+\phi_{rel}+\phi_{shear} \tag{80}$$

$$\phi_{diff} = (2\kappa T/(3\rho\nu))(1/r_Z+1/r_B)(r_B+r_Z) \tag{81}$$

$$\phi_{rel} = \pi(r_Z+r_B)^2 U_{eff} \tag{82}$$

$$\phi_{shear} = 1.3\sqrt{\epsilon/\nu}(r_Z+r_B)^3 \tag{83}$$

$$U_{eff} = (U_B^2+3U_Z^2)/3U_Z \tag{84}$$

**Table 14.** Equations for zooplankton grazing. The terms represent a predator Z consuming a phytoplankton B. Notes (1) If the zooplankton diet contains multiple phytoplankton classes, and grazing is prey saturated, then phytoplankton loss must be reduced to account for the saturation by other types of microalgae; (2) $\frac{Z}{m_Z}$ is the number of individual zooplankton; (3) Phytoplankton pigment is lost to water column without being conserved. Chl $a$ has chemical formulae $C_{55}H_{72}O_5N_4Mg$, and a molecular weight of 893.49 g mol$^{-1}$. The uptake (and subsequent remineralisation) of molecules for chlorophyll synthesis could make up a maximum (at C:Chl = 20) of $(660/893)/20$ and $(56/893)/20 \times (16/106) \times (14/12))$, or ~4 and ~2 per cent of the exchange of C and N between the cell and water column, and will cancel out over the lifetime of a cell. Thus the error in ignoring chlorophyll loss to the water column is small.





$$\frac{\partial [\text{NH}_4]}{\partial t} = g(1-E)(1-\gamma) \tag{85}$$

$$\frac{\partial P}{\partial t} = g\frac{1}{16}\frac{31}{14}(1-E)(1-\gamma) \tag{86}$$

$$\frac{\partial DIC}{\partial t} = g\frac{106}{16}\frac{12}{14}(1-E)(1-\gamma) \tag{87}$$

$$\frac{\partial Z_S}{\partial t} = -g \tag{88}$$

$$\frac{\partial Z_L}{\partial t} = Eg \tag{89}$$

$$\frac{\partial D_{Red}}{\partial t} = g(1-E)\gamma \tag{90}$$

$$\frac{\partial [\text{O}_2]}{\partial t} = -\frac{\partial DIC}{\partial t}\frac{32}{12}\frac{[\text{O}_2]}{K_{OA}+[\text{O}_2]} \tag{91}$$

$$g = \min\left[\mu_{Z_L}^{max}Z_L/E,\frac{Z_L}{m_Z}\left(\phi_{diff}+\phi_{rel}\phi_{shear}\right)Z_S\right] \tag{92}$$

**Table 15.** Equations for zooplankton carnivory, represent large zooplankton $Z_L$ consuming small zooplankton $Z_S$. The parameters values and symbols are given in Table 13 and Table 12

.

### 3.9 Non-grazing plankton mortality

The rate of change of phytoplankton biomass, $B$, as a result of natural mortality is given by:

$$\frac{\partial B}{\partial t} = -m_L B - m_Q B^2 \tag{93}$$

where $m_L$ is the linear mortality coefficient and $m_Q$ is the quadratic mortality coefficient.

5      A combination of linear and quadratic mortality rates are used in the model. When the mortality term is the sole loss term, such as zooplankton in the water column or benthic microalgae in the sediments, a quadratic term is employed to represent increasing predation / viral disease losses in dense populations.

Linear terms have been used to represent a basal respiration rate.

As described in Sec 3.3.5, the mortality terms need to account for the internal properties of lost microalgae.

10      For definitions of the state variables see Tables 16 & 17.

### 3.10 Gas exchange

Gas exchange is calculated using wind speed (we choose a cubic relationship, Wanninkhof and McGillis (1999)), saturation state of the gas (described below) and the Schmidt number of the gas (Wanninkhof, 1992). The transfer coefficient, $k$, is given





| Description | water column | | sediment | |
| --- | --- | --- | --- | --- |
| | linear | quadratic | linear | quadratic |
| | $d^{-1}$ | $d^{-1}$ $(mg\ N\ m^{-3})^{-1}$ | $d^{-1}$ | $d^{-1}$ $(mg\ N\ m^{-3})^{-1}$ |
| Small phytoplankton | 0.1 | - | 1 | - |
| Large phytoplankton | 0.1 | - | 10 | - |
| Microphytobenthos | - | - | - | 0.0001 |
| *Trichodesmium* | 0.1 | 0.1 | - | - |

**Table 16.** Constants and parameter values used for plankton mortality.

$$\frac{\partial [\text{NH}_4]}{\partial t} = m_{L,B} B R_N^* \tag{94}$$

$$\frac{\partial DIP}{\partial t} = \frac{1}{16}\frac{31}{14} m_{L,B} B R_P^* \tag{95}$$

$$\frac{\partial DIC}{\partial t} = \frac{106}{16}\frac{12}{14} m_{L,B} B R_C^* \tag{96}$$

$$\frac{\partial [\text{O}_2]}{\partial t} = -\frac{\partial DIC}{\partial t}\frac{32}{12}\frac{[\text{O}_2]}{K_{OA} + [\text{O}_2]} \tag{97}$$

$$\frac{\partial B}{\partial t} = -m_{L,B} B \tag{98}$$

$$\frac{\partial D_{Red}}{\partial t} = m_{L,B} B \tag{99}$$

**Table 17.** Equations for linear phytoplankton mortality.

$$\frac{\partial Z_S}{\partial t} = -m_{Q,ZS} Z_S^2 \tag{100}$$

$$\frac{\partial Z_L}{\partial t} = -m_{Q,ZL} Z_L^2 \tag{101}$$

$$\frac{\partial D_{Red}}{\partial t} = f_{Z2det}\left(m_{Q,ZS} Z_S^2 + m_{Q,ZL} Z_L^2\right) \tag{102}$$

$$\frac{\partial [\text{NH}_4]}{\partial t} = (1 - f_{Z2det})\left(m_{Q,ZS} Z_S^2 + m_{Q,ZL} Z_L^2\right) \tag{103}$$

**Table 18.** Equations for the zooplankton mortality. $f_{Z2det}$ is the fraction of zooplankton mortality that is remineralised, and is equal to 0.5 for both small and large zooplankton.





by:

$$k = \frac{0.0283}{360000} u_{10}^3 \left( \text{Sc}/660 \right)^{-1/2} \tag{104}$$

where $0.0283$ cm hr$^{-1}$ is an empirically-determined constant (Wanninkhof and McGillis, 1999), $u_{10}^3$ is the short-term steady wind at 10 m above the sea surface [m s$^{-1}$], the Schmidt number, Sc, is the ratio of the diffusivity of momentum and that of the exchanging gas, and is given by a cubic temperature relationship (Wanninkhof, 1992). Finally, a conversion factor of 360000 m s$^{-1}$ (cm hr$^{-1}$)$^{-1}$ is used.

In practice the hydrodynamic model can contain thin surface layers as the surface elevation moves between z-levels. Further, physical processes of advection and diffusion and gas fluxes are done sequentially, allowing concentrations to build up through a single timestep. To avoid unrealistic changes in the concentration of gases in thin surface layers, the shallowest layer thicker than 20 cm receives all the surface fluxes.

### 3.10.1 Oxygen

The saturation state of oxygen $[O_2]_{sat}$ is determined as a function of temperature and salinity following Weiss (1970). The change in concentration of oxygen in the surface layer due to a sea-air oxygen flux (+ve from sea to air) is given by:

$$\frac{\partial [O_2]}{\partial t} = k_{O_2} \left( [O_2]_{sat} - [O_2] \right) / h \tag{105}$$

where $k_{O_2}$ is the transfer coefficient for oxygen (Eq. 104), $[O_2]$ is the dissolved oxygen concentration in the surface waters, and $h$ is the thickness of the surface layer of the model into which sea-air flux flows.

### 3.10.2 Carbon dioxide

The change in surface dissolved inorganic carbon concentration, $DIC$, resulting from the sea-air flux (+ve from sea to air) of carbon dioxide is given by:

$$\frac{\partial DIC}{\partial t} = k_{CO_2} \left( [CO_2]_{atm} - [CO_2] \right) / h \tag{106}$$

where $k_{CO_2}$ the transfer coefficient for carbon dioxide (Eq. 104), $[CO_2]$ is the dissolved carbon dioxide concentration in the surface waters determined from $DIC$ and $A_T$ using the carbon chemistry equilibria calculations described in Sec 3.5.1, $[CO_2]_{atm}$ is the partial pressure of carbon dioxide in the atmosphere, and $h$ is the thickness of the surface layer of the model into which sea-air flux flows.

Note the carbon dioxide flux is not determined by the gradient in DIC, but the gradient in $[CO_2]$. At pH values around 8, $[CO_2]$ makes up only approximately 1/200th of DIC in seawater, significantly reducing the air-sea exchange. Counteracting this reduced gradient, note that changing DIC results in an approximately 10 fold change in $[CO_2]$ (quantified by the Revelle factor (Zeebe and Wolf-Gladrow, 2001)). Thus, the gas exchange of $CO_2$ is approximately $1/200 \times 10 = 1/20$ of the oxygen flux for the same proportional perturbation in DIC and oxygen. At a Sc number of 524 (25°C seawater) and a wind speed of 12 m s$^{-1}$, 1 m of water equilibrates with $CO_2$ in the atmosphere with an e-folding timescale of approximately 1 day.

The model has no processes such as seed or gamete dispersion that result in new habitats being created.





## 4 Epibenthic processes

In the model, benthic communities are quantified as a biomass per unit area, or areal biomass. At low biomass, the community is composed of a few specimens spread over a small fraction of the bottom, with no interaction between the nutrient and energy acquisition of individuals. Thus, at low biomass the areal fluxes are a linear function of the biomass.

As biomass increases, the individuals begin to cover a significant fraction of the bottom. For nutrient and light fluxes that are constant per unit area, such as downwelling irradiance and sediment releases, the flux per unit biomass decreases with increasing biomass. Some processes, such as photosynthesis in a thick seagrass meadow or nutrient uptake by a coral reef, become independent of biomass (Atkinson, 1992) as the bottom becomes completely covered. To capture the non-linear effect of biomass on benthic processes, we use an effective projected area fraction, $A_{eff}$.

To restate, at low biomass, the area on the bottom covered by the benthic community is a linear function of biomass. As the total leaf area approaches and exceeds the projected area, the projected area for the calculation of water-community exchange approaches 1, and becomes independent of biomass. This is represented using:

$$A_{eff} = 1 - \exp(-\Omega_B \ B) \tag{107}$$

where $A_{eff}$ is the effective projected area fraction of the benthic community (m$^2$ m$^{-2}$), $B$ is the biomass of the benthic
community (g N m$^{-2}$), and $\Omega_B$ is the nitrogen-specific leaf area coefficient (m$^2$ g N$^{-1}$). For further explanation of $\Omega_B$ see Baird et al. (2016a).

The parameter $\Omega_B$ is critical: it provides a means of converting between biomass and fractions of the bottom covered, and is used in calculating the absorption cross-section of the leaf and the nutrient uptake of corals and macroalgae. That $\Omega_B$ has a simple physical explanation, and can be determined from commonly undertaken morphological measurement (see below),
gives us confidence in its use throughout the model.

### 4.1 Epibenthic optical model

The spectrally-resolved light field at the base of the water column is attenuated, in vertical order, by macroalgae, seagrass (*Zostera* then shallow and then deep forms of *Halophila*), followed by the zooxanthellae in corals. The downwelling irradiance at wavelength $\lambda$ after passing through each macroalgae and seagrass species is given by, $E_{below,\lambda}$:

$$E_{below,\lambda} = E_{d,above,\lambda} e^{-A_\lambda \Omega_X X} \tag{108}$$

where $E_{above,\lambda}$ for macroalgae is $E_{d,bot,\lambda}$, the downwelling irradiance of the bottom water column layer, $A_\lambda$ is the absorbance of the leaf, $\Omega$ is the nitrogen specific leaf area, and $X$ is the leaf nitrogen biomass.

The light absorbed by corals is assumed to be entirely due to zooxanthellae, and is given by:

$$E_{below,\lambda} = E_{above,\lambda} e^{-n\alpha_\lambda} \tag{109}$$

where $n = CS/m_{N,CS}$ is the areal density of zooxanthellae cells and $\alpha_\lambda$ is the absorption cross-section of a cell a result of the absorption of multiple pigment types.





| Variable | Symbol | Units |
|---|---|---|
| Downwelling irradiance | $E_d$ | W m$^{-2}$ |
| Macroalgae biomass | $MA$ | g N m$^{-2}$ |
| Water column detritus, C:N:P = 550:30:1 | $D_{Atk}$ | g N m$^{-3}$ |
| | | |
| Effective projected area of macroalgae | $A_{eff}$ | m$^2$ m$^{-2}$ |
| Leaf absorbance | $A_{L,\lambda}$ | - |
| Bottom stress | $\tau$ | N m$^{-2}$ |
| Wavelength | $\lambda$ | nm |
| Bottom water layer thickness | $h_{wc}$ | m |

**Table 19.** State and derived variables for the macroalgae model. For simplicity in the equations all dissolved constituents are given in grams, although elsewhere they are shown in milligrams.

The optical model for microphytobenthic algae, and the bottom reflectance due to sediment and bottom types, is described in Sec. 5.1.

### 4.2 Macroalgae

The macroalgae model considers the diffusion-limited supply of dissolved inorganic nutrients (N and P) and the absorption of
light, delivering N, P and fixed C respectively. Unlike the microalgae model, no internal reserves are considered, implying that the macroalgae has a fixed stoichiometry that can be specified as:

$$550\text{CO}_2 + 30\text{NO}_3^- + \text{PO}_4^{3-} + 792\text{H}_2\text{O} \xrightarrow{5500 \text{ photons}} (\text{CH}_2\text{O})_{550}(\text{NH}_3)_{30}\text{H}_3\text{PO}_4 + 716\text{O}_2 \tag{110}$$

where the stoichiometry is based on Atkinson and Smith (1983) (see also Baird and Middleton (2004); Hadley et al. (2015a, b)). Note that when ammonia is taken up instead of nitrate there is a slightly different O$_2$ balance (Sec. 6.1). In the next section
will consider the maximum nutrient uptake and light absorption, and then bring them together to determine the realised growth rate.

### 4.2.1 Nutrient uptake

Nutrient uptake by macroalgae is a function of nutrient concentration, water motion (Hurd, 2000) and internal physiology. The maximum flux of nutrients is specified as a mass transfer limit per projected area of macroalgae and is given by (Falter et al.,
2004; Zhang et al., 2011):

$$S_{\text{x}} = 2850 \left( \frac{2\tau}{\rho} \right)^{0.38} \text{Sc}_{\text{x}}^{-0.6}, \text{Sc}_{\text{x}} = \frac{\nu}{D_{\text{x}}} \tag{111}$$

where $S_{\text{x}}$ is the mass transfer rate coefficient of element x = N, P, $\tau$ is the shear stress on the bottom, $\rho$ is the density of water and Sc$_{\text{x}}$ is the Schmidt number. The Schmidt number is the ratio of the diffusivity of momentum, $\nu$, and mass, $D_{\text{x}}$, and varies





with temperature, salinity and nutrient species. The rate constant $S$ can be thought of as the height of water cleared of mass per unit of time by the water-macroalgae exchange.

### 4.2.2 Light capture

The calculation of light capture by macroalgae involves estimating the fraction of light that is incident upon the leaves, and the

fraction that is absorbed. The rate of photon capture is given by:

$$k_I = \frac{\left(10^9 hc\right)^{-1}}{A_V} \int E_{d,\lambda} \left(1 - \exp\left(-A_{L,\lambda}\Omega_{MA}MA\right)\right) \lambda d\lambda \tag{112}$$

where $h$, $c$ and $A_V$ are fundamental constants, $10^9$ nm m$^{-1}$ accounts for the typical representation of wavelength, $\lambda$ in nm, and $A_{L,\lambda}$ is the spectrally-resolved absorbance of the leaf. As shown in Eq. 107, the term $1 - \exp\left(-\Omega_{MA}MA\right)$ gives the effective projected area fraction of the community. In the case of light absorption of macroalgae, the exponent is multiplied by the leaf

absorbance, $A_{L,\lambda}$, to account for the transparency of the leaves. At low macroalgae biomass, absorption at wavelength $\lambda$ is equal to $E_{d,\lambda}A_{L,\lambda}\Omega_{MA}MA$, increasing linearly with biomass as all leaves at low biomass are exposed to full light (i.e. there is no self-shading). At high biomass, the absorption by the community asymptotes to $E_{d,\lambda}$, at which point increasing biomass does not increase the absorption as all light is already absorbed.

For more details on the calculation of $\Omega_{MA}$ see Baird et al. (2016a).

### 4.2.3 Growth

The growth rate combines nutrient, light and maximum organic matter synthesis rates following:

$$\mu_{MA} = \min\left[\mu_{MA}^{max}, \frac{30}{5500}14\frac{k_I}{MA}, \frac{S_N A_{eff} N}{MA}, \frac{30}{1}\frac{14}{31}\frac{S_P A_{eff} P}{MA}\right] \tag{113}$$

and the production of macroalgae is given by $\mu_{MA}MA$. Note, as per seagrass, that the maximum growth rates sits within the minimum operator. This allows the growth of macroalgae to the independent of temperature at low light, but still have an

exponential dependence at maximum growth rates (Baird et al., 2003).

### 4.2.4 Mortality

Mortality is defined as a simple linear function of biomass:

$$\frac{\partial MA}{\partial t} = -\zeta_{MA}MA \tag{125}$$

A quadratic formulation is not necessary as both the nutrient and light capture rates become independent of biomass as $MA \gg$

$1/\Omega_{MA}$. Thus the steady-state biomass of macroalgae under nutrient limitation is given by:

$$(MA)_{SS} = \frac{S_N A_{eff} N}{\zeta} \tag{126}$$

and for light-limited growth by:

$$(MA)_{SS} = \frac{k_I}{\zeta} \tag{127}$$





$$\frac{\partial N}{\partial t} = -\mu_{MA}MA/h_{wc} \tag{114}$$

$$\frac{\partial P}{\partial t} = -\frac{1}{30}\frac{31}{14}\mu_{MA}MA/h_{wc} \tag{115}$$

$$\frac{\partial DIC}{\partial t} = -\frac{550}{30}\frac{12}{14}\mu_{MA}MA/h_{wc} \tag{116}$$

$$\frac{\partial [O_2]}{\partial t} = \frac{716}{30}\frac{32}{14}\left(\mu_{MA}MA\right)/h_{wc} \tag{117}$$

$$\frac{\partial MA}{\partial t} = \mu_{MA}MA - \zeta_{MA}MA \tag{118}$$

$$\frac{\partial D_{Atk}}{\partial t} = \zeta_{MA}MA/h_{wc} \tag{119}$$

$$\mu_{MA} = \min\left[\mu_{MA}^{max}, \frac{30}{5500}14\frac{k_I}{MA}, \frac{S_N A_{eff} N}{MA}, \frac{30}{1}\frac{14}{31}\frac{S_P A_{eff} P}{MA}\right] \tag{120}$$

$$S_x = 2850\left(\frac{2\tau}{\rho}\right)^{0.38} Sc^{-0.6}, Sc = \frac{\nu}{D_x} \tag{121}$$

$$k_I = \frac{\left(10^9 hc\right)^{-1}}{A_V}\int E_{d,\lambda}\left(1 - \exp\left(-A_{L,\lambda}\Omega_{MA}MA\right)\right)\lambda d\lambda \tag{122}$$

$$A_{eff} = 1 - \exp(-\Omega_{MA}\ MA) \tag{123}$$

$$550CO_2 + 30NO_3^- + PO_4^{3-} + 792H_2O \xrightarrow{5500\ \text{photons}} (CH_2O)_{550}(NH_3)_{30}H_3PO_4 + 716O_2 + 391H^+ \tag{124}$$

**Table 20.** Equations for the macroalgae model. Other constants and parameters are defined in Table 21. 14 g N mol $N^{-1}$; 12 g C mol $C^{-1}$; 31 g P mol $P^{-1}$; 32 g O mol $O_2^{-1}$. Uptake shown here is for nitrate, see Sec. 6.1 for ammonia uptake.

| | Symbol | Value | Units |
|---|---|---|---|
| *Parameters* | | | |
| Maximum growth rate of macroalgae | $\mu_{MA}^{max}$ | 1.0 | $d^{-1}$ |
| Nitrogen-specific area of macroalgae | $\Omega_{MA}$ | 1.0 | $(g N m^{-2})^{-1}$ |
| [a]Leaf absorbance | $A_{L,\lambda}$ | $\sim 0.7$ | - |
| Mortality rate | $\zeta_{MA_A}$ | 0.01 | $d^{-1}$ |

**Table 21.** Constants and parameter values used to model macroalgae. [a]Spectrally-resolved values

The full macroalgae equations, parameters and symbols are listed in Tables 19, 20 and 21.





| Variable | Symbol | Units |
|---|---|---|
| Downwelling irradiance | $E_d$ | W m$^{-2}$ |
| Porewater DIN concentration | $N_s$ | g N m$^{-3}$ |
| Porewater DIP concentration | $P_s$ | g P m$^{-3}$ |
| Water column DIC concentration | $DIC$ | g C m$^{-3}$ |
| Water column oxygen concentration | $[O_2]$ | g O m$^{-3}$ |
| Above-ground seagrass biomass | $SG_A$ | g N m$^{-2}$ |
| Below-ground seagrass biomass | $SG_B$ | g N m$^{-2}$ |
| Detritus at 550:30:1 in sediment | $D_{Atk,sed}$ | g N m$^{-3}$ |
| | | |
| Effective projected area of seagrass | $A_{eff}$ | m$^2$ m$^{-2}$ |
| Bottom stress | $\tau$ | N m$^{-2}$ |
| Thickness of sediment layer $l$ | $h_{s,l}$ | m |
| Bottom water layer thickness | $h_{wc}$ | m |
| Wavelength | $\lambda$ | nm |
| Translocation rate | $\Upsilon$ | g N m$^{-2}$ s$^{-1}$ |
| Porosity | $\phi$ | - |

**Table 22.** State and derived variables for the seagrass model. For simplicity in the equations all dissolved constituents are given in grams, although elsewhere they are shown in milligrams. The bottom water column thickness varies is spatially-variable, depending on bathymetry.

## 4.3 Seagrass

Seagrasses are quantified per m$^2$ with a constant stoichiometry (C:N:P = 550:30:1) for both above-ground, $SG_A$, and below-ground, $SG_B$, biomass, and can translocate organic matter at this constant stoichiometry between the two stores of biomass. Growth occurs only in the above-ground biomass, but losses (grazing, decay etc.) occur in both. Multiple seagrass varieties are represented. The varieties are modelled using the same equations for growth, respiration and mortality, but with different parameter values.





$$\frac{\partial N_w}{\partial t} = -\left(\mu_{SG} - \frac{\mu_{SG}^{max}\overline{N_s}}{K_{SG,N} + \overline{N_s}}\right)/h_{wc} \tag{128}$$

$$\frac{\partial P_w}{\partial t} = -\left(\frac{1}{30}\frac{31}{14}\mu_{SG} - \frac{\mu_{SG}^{max}\overline{P_s}}{K_{SG,P} + \overline{P_s}}\right)/h_{wc} \tag{129}$$

$$\frac{\partial N_{s,l}}{\partial t} = -f_{N,l}/(h_{s,l}\phi_l) \tag{130}$$

$$\frac{\partial P_{s,l}}{\partial t} = -\frac{1}{30}\frac{31}{14}f_{P,l}/(h_{s,l}\phi_l) \tag{131}$$

$$\frac{\partial DIC}{\partial t} = -\frac{550}{30}\frac{12}{14}\left(\mu_{SG_A}SG_A\right)/h_{wc} \tag{132}$$

$$\frac{\partial [O_2]}{\partial t} = \frac{716}{30}\frac{32}{14}\left(\mu_{SG_A}SG_A\right)/h_{wc} \tag{133}$$

$$\frac{\partial SG_A}{\partial t} = \mu_{SG_A}SG_A - (\zeta_{SG_A} + \zeta_{SG,\tau})\left(SG_A - \frac{f_{seed}}{\Omega_{SG}}(1 - f_{below})\right) - \Upsilon \tag{134}$$

$$\frac{\partial SG_B}{\partial t} = -(\zeta_{SG_B} + \zeta_{SG,\tau})\left(SG_B - \frac{f_{seed}}{\Omega_{SG}}f_{below}\right) + \Upsilon \tag{135}$$

$$\frac{\partial D_{Atk,sed}}{\partial t} = \left((\zeta_{SG_A} + \zeta_{SG,\tau})\left(SG_A - \frac{f_{seed}}{\Omega_{SG}}(1 - f_{below})\right)\right)/(h_{sed}\phi) \tag{136}$$
$$+ \left((\zeta_{SG_B} + \zeta_{SG,\tau})\left(SG_B - \frac{f_{seed}}{\Omega_{SG}}f_{below}\right)\right)/(h_{sed}\phi)$$

$$\mu_{SG_A} = \min\left[\frac{\mu_{SG}^{max}\overline{N_s}}{K_{SG,N} + \overline{N_s}} + S_N A_{eff}N, \frac{\mu_{SG}^{max}\overline{P_s}}{K_{SG,P} + \overline{P_s}} + S_P A_{eff}P, \frac{30}{5500}14\frac{\max(0, k_I - k_{resp})}{SG_A}\right] \tag{137}$$

$$\overline{N_s} = \frac{\sum_{l=1}^{L} N_{s,l}h_{s,l}\phi_l}{\sum_{l=1}^{L} h_{s,l}\phi_l} \tag{138}$$

$$\overline{P_s} = \frac{\sum_{l=1}^{L} P_{s,l}h_{s,l}\phi_l}{\sum_{l=1}^{L} h_{s,l}\phi_l} \tag{139}$$

$$f_{N,l} = \frac{N_{s,l}h_{s,l}\phi_l}{\sum_{l=1}^{L} N_{s,l}h_{s,l}\phi_l}\mu_{SG}SG_A \tag{140}$$

$$f_{P,l} = \frac{P_{s,l}h_{s,l}\phi_l}{\sum_{l=1}^{L} P_{s,l}h_{s,l}\phi_l}\mu_{SG}SG_A \tag{141}$$

$$k_I = \frac{(10^9 hc)^{-1}}{A_V}\int E_{d,\lambda}\left(1 - \exp\left(-A_{L,\lambda}\Omega_{SG}SG_A\sin\beta_{blade}\right)\right)\lambda d\lambda \tag{142}$$

$$k_{resp} = 2\left(E_{comp}A_L\Omega_{SG}\sin\beta_{blade} - \frac{5500}{30}\frac{1}{14}\zeta_{SG_A}\right)SG_A \tag{143}$$

$$\Upsilon = \left(f_{below} - \frac{SG_B}{SG_B + SG_A}\right)(SG_A + SG_B)\tau_{tran} \tag{144}$$

$$550CO_2 + 30NO_3^- + PO_4^{3-} + 792H_2O \xrightarrow{5500\ photons} (CH_2O)_{550}(NH_3)_{30}H_3PO_4 + 716O_2 + 391H^+ \tag{145}$$

**Table 23.** Equations for the seagrass model. Other constants and parameters are defined in Table 24. The equation for organic matter formation gives the stoichiometric constants; 14 g N mol N$^{-1}$; 12 g C mol C$^{-1}$; 31 g P mol P$^{-1}$; 32 g O mol O$_2^{-1}$.





*Parameters*

| | Symbol | Zostera capricorni | Halophila ovalis | Halophila decipens | Units |
|---|---|---|---|---|---|
| [a]Maximum growth rate of seagrass | $\mu_{SG}^{max}$ | 0.4 | 0.4 | 0.4 | $d^{-1}$ |
| [b]Nitrogen-specific area of seagrass | $\Omega_{SG}$ | 1.5 | 1.9 | 1.9 | $(g\,N\,m^{-2})^{-1}$ |
| [c]Leaf absorbance | $A_{L,\lambda}$ | ~0.7 | ~0.7 | ~0.7 | - |
| [d]Fraction biomass below ground | $f_{below}$ | 0.75 | 0.25 | 0.5 | - |
| [e]Translocation rate | $\tau_{tran}$ | 0.033 | 0.033 | 0.033 | $d^{-1}$ |
| [f]Half-saturation P uptake | $K_{SG,P}$ | 96 | 96 | 96 | mg P m$^{-3}$ |
| [g]Half-saturation N uptake | $K_{SG,N}$ | 420 | 420 | 420 | mg N m$^{-3}$ |
| [h]Compensation scalar PAR irradiance | $E_{comp}$ | 4.5 | 2.0 | 1.5 | mol photon m$^{-2}$ d$^{-1}$ |
| [h]Leaf loss rate | $\zeta_{SG_A}$ | 0.04 | 0.08 | 0.06 | $d^{-1}$ |
| [h]Root loss rate | $\zeta_{SG_B}$ | 0.004 | 0.004 | 0.004 | $d^{-1}$ |
| Seed biomass as a fraction of 63 % cover | $f_{seed}$ | 0.01 | 0.01 | 0.01 | - |
| [i]Seagrass root depth | $z_{root}$ | 0.15 | 0.08 | 0.05 | m |
| Sine of nadir canopy bending angle | $\sin\beta_{blade}$ | 0.5 | 1.0 | 1.0 | - |
| Mortality critical shear stress | $\tau_{SG,shear}$ | 1.0 | 1.0 | 1.0 | N m$^{-2}$ |
| Mortality shear stress time-scale | $\tau_{SG,time}$ | 0.5 | 0.5 | 0.5 | d |
| Max. shear stress loss rate | $\zeta_{SG,\tau}^{max}$ | 2 | 2 | 2 | $d^{-1}$ |

**Table 24.** Constants and parameter values used to model seagrass. [a] ×2 for nighttime ×2 for roots; [b] Zostera - calculated from leaf characteristics in (Kemp et al., 1987; Hansen et al., 2000), *Halophia ovalis* - calculated from leaf dimensions in Vermaat et al. (1995) - $\Omega_{SG}$ can also be determined from specific leaf area such as determined in Cambridge and Lambers (1998) for 9 Australian seagrass species; [c] Spectrally-resolved values in Baird et al. (2016a); [d] Duarte and Chiscano (1999); [e] loosely based on Kaldy et al. (2013); [f] *Thalassia testudinum* Gras et al. (2003); [g] *Thalassia testudinum* (Lee and Dunton, 1999); [h] Chartrand et al. (2012); Longstaff (2003), Chartrand et al. (2017); [i] Roberts (1993).





Here we present just the equations for the seagrass submodel. A description of the seagrass processes of growth, translocation between roots and leaves, and mortality has been published in Baird et al. (2016a), along with a comparison to observations from Gladstone Harbour on the northeast Australian coast.

### 4.4 Coral polyps

The coral polyp parameterisation consists of a microalgae growth model to represent zooxanthellae growth based on Baird et al. (2013), and the parameterisation of coral - zooxanthellae interaction based on the host - symbiont model of Gustafsson et al. (2013), a new photoadaptation, photoinhibition and reaction centre dynamics models. The extra detail on the zooxanthellae photosystem is required due to its important role in thermal-stress driven coral bleaching (Yonge, 1930; Suggett et al., 2008).

#### 4.4.1 Coral host, symbiont and the environment

The state variables for the coral polyp model (Table 25) include the biomass of coral tissue, $CH$ (g N m$^{-2}$), and the structure material of the zooxanthellae cells, $CS$ (mg N m$^{-2}$). The structure material of the zooxanthellae, $CS$, in addition to nitrogen, contains carbon and phosphorus at the Redfield ratio. The zooxanthellae cells also contain reserves of nitrogen, $R_N$ (mg N m$^{-2}$), phosphorus, $R_P$ (mg P m$^{-2}$), and carbon, $R_C$ (mg C m$^{-2}$).

The zooxanthellae light absorption capability is resolved by considering the time-varying concentrations of pigments chloro-
15 phyll $a$, $Chl$, diadinoxanthin, $X_p$, and diatoxanthin $X_h$, for which the state variable represents the areal concentration. A further three pigments, chlorophyll $c_2$, peridinin, and $\beta$-carotene are considered in the absorption calculations, but their concentrations are in fixed ratios to chlorophyll $a$. Exchanges between the coral community and the overlying water can alter the water column concentrations of dissolved inorganic carbon, $DIC$, nitrogen, $N$, and phosphorus, $P$, as well as particulate phytoplankton, $B$, zooplankton, $Z$, and detritus, $D$, where multiple nitrogen, plankton and detritus types are resolved (Table 25).

The coral host is able to assimilate particulate organic nitrogen either through translocation from the zooxanthellae cells or through the capture of water column organic detritus and/or plankton. The zooxanthellae varies its intracellular pigment content depending on potential light limitation of growth, and the incremental benefit of adding pigment, allowing for the package effect (Baird et al., 2013). The coral tissue is assumed to have a Redfield C:N:P stoichiometry (Redfield et al., 1963), as shown by Muller-Parker et al. (1994). The zooxanthellae are modelled with variable C:N:P ratios (Muller-Parker et al.,
1994), based on a structure material at the Redfield ratio, but with variable internal reserves. The fluxes of C, N and P with the overlying water column (nutrient uptake and detrital / mucus release) can therefore vary from the Redfield ratio.

An explanation of the individual processes follows, with tables in the Appendix listing all the model state variables (Table 25), derived variables (Table 26), equations (Tables 27, 28, 29 and 30), and parameters values (Tables 31 and 32).

Here we present just the equations for the coral submodel. The description of the coral processes has been published in Baird
et al. (2018), along with a comparison to observations from the Great Barrier Reef on the northeast Australian coast. The effect of coral calcification on water column properties is described below.



| Variable | Symbol | Units |
|---|---|---|
| Dissolved inorganic nitrogen (DIN) | $N$ | mg N m$^{-3}$ |
| Dissolved inorganic phosphorus (DIP) | $P$ | mg P m$^{-3}$ |
| Zooxanthellae biomass | $CS$ | mg N m$^{-2}$ |
| Reserves of nitrogen | $R_N$ | mg N cell$^{-1}$ |
| Reserves of phosphorus | $R_P$ | mg P cell$^{-1}$ |
| Reserves of carbon | $R_C$ | mg C cell$^{-1}$ |
| Coral biomass | $CH$ | g N m$^{-2}$ |
| Suspended phytoplankton biomass | $B$ | mg N m$^{-3}$ |
| Suspended zoooplankton biomass | $Z$ | mg N m$^{-3}$ |
| Suspended detritus at 106:16:1 | $D_{Red}$ | mg N m$^{-3}$ |
| Macroalgae biomass | MA | mg N m$^{-3}$ |
| Temperature | $T$ | °C |
| Absolute salinity | $S_A$ | kg m$^{-3}$ |
| zooxanthellae chlorophyll *a* concentration | $Chl$ | mg m$^{-2}$ |
| zooxanthellae diadinoxanthin concentration | $X_p$ | mg m$^{-2}$ |
| zooxanthellae diatoxanthin concentration | $X_h$ | mg m$^{-2}$ |
| Oxidised reaction centre concentration | $Q_{ox}$ | mg m$^{-2}$ |
| Reduced reaction centre concentration | $Q_{red}$ | mg m$^{-2}$ |
| Inhibited reaction centre concentration | $Q_{in}$ | mg m$^{-2}$ |
| Reactive oxygen species concentration | [ROS] | mg m$^{-2}$ |
| Chemical oxygen demand | $COD$ | mg O$_2$ m$^{-3}$ |

**Table 25.** Model state variables for the coral polyp model. Note that water column variables are 3 dimensional, benthic variables are 2 dimensional, and unnormalised reserves are per cell.

### 4.4.2 Coral calcification

The rate of coral calcification is a function of the water column aragonite saturation, $\Omega_a$, and the normalised reserves of fixed carbon in the symbiont, $R_C^*$. The rates of change of DIC and total alkalinity, $A_T$, in the bottom water column layer of thickness $h_{wc}$ due to calcification becomes:

$$\frac{\partial DIC}{\partial t} = -12gA_{eff}/h_{wc} \tag{190}$$

$$\frac{\partial A_T}{\partial t} = -2gA_{eff}/h_{wc} \tag{191}$$





| Variable | Symbol | Units |
|---|---|---|
| Downwelling irradiance | $E_d$ | $\text{W m}^{-2}$ |
| Maximum reserves of nitrogen | $R_N^{\max}$ | $\text{mg N cell}^{-1}$ |
| Maximum reserves of phosphorus | $R_P^{\max}$ | $\text{mg P cell}^{-1}$ |
| Maximum reserves of carbon | $R_C^{\max}$ | $\text{mg C cell}^{-1}$ |
| Normalised reserves of nitrogen | $R_N^* \equiv R_N / R_N^{\max}$ | - |
| Normalised reserves of phosphorus | $R_P^* \equiv R_P / R_P^{\max}$ | - |
| Normalised reserves of carbon | $R_C^* \equiv R_C / R_C^{\max}$ | - |
| Intracellular chlorophyll $a$ concentration | $c_i$ | $\text{mg m}^{-3}$ |
| Intracellular diadinoxanthin concentration | $x_p$ | $\text{mg m}^{-3}$ |
| Intracellular diatoxanthin concentration | $x_h$ | $\text{mg m}^{-3}$ |
| Total reaction centre concentration | $Q_\text{T}$ | $\text{mg m}^{-2}$ |
| Total active reaction centre concentration | $Q_\text{a}$ | $\text{mg m}^{-2}$ |
| Concentration of zooxanthellae cells | $n$ | $\text{cell m}^{-2}$ |
| Thickness of the bottom water column layer | $h_{wc}$ | m |
| Effective projected area fraction | $A_{eff}$ | $\text{m}^2 \text{ m}^{-2}$ |
| Area density of zooxanthellae cells | $n_{CS}$ | $\text{cell m}^{-2}$ |
| Absorption cross-section | $\alpha$ | $\text{m}^2 \text{ cell}^{-1}$ |
| Rate of photon absorption | $k_I$ | $\text{mol photon cell}^{-1} \text{ s}^{-1}$ |
| Photon-weighted average opaqueness | $\overline{\chi}$ | - |
| Maximum Chl. synthesis rate | $k_\text{Chl}^{\max}$ | $\text{mg Chl m}^{-3} \text{ d}^{-1}$ |
| Density of water | $\rho$ | $\text{kg m}^{-3}$ |
| Bottom stress | $\tau$ | $\text{N m}^{-2}$ |
| Schmidt number | Sc | - |
| Mass transfer rate coefficient for particles | $S_{part}$ | $\text{m d}^{-1}$ |
| Heterotrophic feeding rate | $G$ | $\text{g N m}^{-2} \text{ d}^{-1}$ |
| Wavelength | $\lambda$ | nm |
| Translocation fraction | $f_{tran}$ | - |
| Active fraction of oxidised reaction centres | $a_{Q_{ox}}^*$ | - |

**Table 26.** Derived variables for the coral polyp model.





$$\frac{\partial N}{\partial t} = -S_N N (1 - R_N^*) A_{eff} \tag{146}$$

$$\frac{\partial P}{\partial t} = -S_P P (1 - R_P^*) A_{eff} \tag{147}$$

$$\frac{\partial DIC}{\partial t} = -\left( \frac{106}{1060} 12 k_I \frac{Q_{ox}}{Q_T} a_{Q_{ox}}^* (1 - R_C^*) - \frac{106}{16} \frac{12}{14} \mu_{CS}^{\max} \phi R_C^* \right) (CS/m_N) \tag{148}$$

$$\frac{\partial [O_2]}{\partial t} = \left( \frac{106}{1060} 32 k_I \frac{Q_{ox}}{Q_T} a_{Q_{ox}}^* (1 - R_C^*) - \frac{106}{16} \frac{32}{14} \mu_{CS}^{\max} \phi R_C^* \right) (CS/m_N) \tag{149}$$

$$\frac{\partial R_N}{\partial t} = S_N N (1 - R_N^*)/(CS/m_N) - \mu_{CS}^{\max} R_P^* R_N^* R_C^* (m_N + R_N) \tag{150}$$

$$\frac{\partial R_P}{\partial t} = S_P P (1 - R_P^*)/(CS/m_N) - \mu_{CS}^{\max} R_P^* R_N^* R_C^* (m_P + R_P) \tag{151}$$

$$\frac{\partial R_C}{\partial t} = k_I \left( \frac{Q_{ox}}{Q_T} \right) a_{Q_{ox}}^* (1 - R_C^*) - \mu_{CS}^{\max} R_P^* R_N^* R_C^* (m_C + R_C)$$
$$- \mu_{CS}^{\max} \phi m_C R_C^* \tag{152}$$

$$\frac{\partial CS}{\partial t} = \mu_{CS}^{\max} R_P^* R_N^* R_C^* CS - \zeta_{CS} CS \tag{153}$$

$$\frac{\partial c_i}{\partial t} = \left( k_{\mathrm{Chl}}^{\max} (1 - R_C^*)(1 - Q_{in}/Q_T) \overline{\chi} - \mu_P^{\max} R_P^* R_N^* R_C^* c_i \right)(CS/m_N) \tag{154}$$

$$\frac{\partial X_p}{\partial t} = \Theta_{xan2chl} \left( k_{\mathrm{Chl}}^{\max} (1 - R_C^*)(1 - Q_{in}/Q_T) \overline{\chi} \right)$$
$$- 8 \left( Q_{in}/Q_t - 0.5 \right)^3 \tau_{xan} \Phi (X_p + X_h) \tag{155}$$

$$\frac{\partial X_h}{\partial t} = 8 \left( Q_{in}/Q_T - 0.5 \right)^3 \tau_{xan} \Phi (X_p + X_h) \tag{157}$$

$$\frac{\partial CS}{\partial t} = (1 - f_{tran}) \mu_{CS} CS - \zeta_{CS} CS + f_{remin} \frac{\zeta_{CH}}{A_{eff}} CH^2 \tag{158}$$

$$k_I = \frac{(10^9 hc)^{-1}}{A_V} \int \alpha_\lambda E_{d,\lambda} \lambda \, d\lambda \tag{159}$$

$$S_x = 2850 \left( \frac{2\tau}{\rho} \right)^{0.38} \mathrm{Sc}_x^{-0.6}, \mathrm{Sc}_x = \frac{\nu}{D_x} \tag{160}$$

$$\Phi = 1 - 4 \left( \frac{X_p}{X_p + X_h} - 0.5 \right)^2 \text{ or } \Phi = 1 \tag{161}$$

**Table 27.** Equations for the interactions of coral host, symbiont and environment excluding bleaching loss terms that appear in Table 30. The term $CS/m_N$ is the concentration of zoothanxellae cells. The equation for organic matter formation gives the stoichiometric constants; 12 g C mol C$^{-1}$; 32 g O mol O$_2^{-1}$.





$$\frac{\partial CH}{\partial t} = G' - \frac{\zeta_{CH}}{A_{eff}}CH^2 \tag{162}$$

$$\frac{\partial B}{\partial t} = -S_{part}A_{eff}B\frac{G'}{G}/h_{wc} \tag{163}$$

$$\frac{\partial Z}{\partial t} = -S_{part}A_{eff}Z\frac{G'}{G}/h_{wc} \tag{164}$$

$$\frac{\partial D_{Red}}{\partial t} = \left(-S_{part}A_{eff}D_{Red}\frac{G'}{G} + (1-f_{remin})\frac{\zeta_{CH}}{A_{eff}}CH^2\right)/h_{wc} \tag{165}$$

$$f_{tran} = \frac{\pi r_{CS}^2 n_{CS}}{2CH\Omega_{CH}} \tag{166}$$

$$G = S_{part}A_{eff}(B+Z+D_{Red}) \tag{167}$$

$$G' = \min\left[\min\left[\mu_{CH}^{max}CH - f_{tran}\mu_{CS}CS - \zeta_{CS}CS,0\right],G\right] \tag{168}$$

$$A_{eff} = 1-\exp(-\Omega_{CH}CH) \tag{169}$$

**Table 28.** Equations for the coral polyp model. The term $CS/m_N$ is the concentration of zoothanxellae cells. The equation for organic matter formation gives the stoichiometric constants; 12 g C mol C$^{-1}$; 32 g O mol O$_2^{-1}$. Other constants and parameters are defined in Table 32.

$$\frac{\partial Q_{\text{ox}}}{\partial t} = -k_I n\, m_{\text{RCII}}\left(\frac{Q_{\text{ox}}}{Q_{\text{T}}}\right)\left(1-a_{Q_{ox}}^*\left(1-R_C^*\right)\right) + f_2(T)R_N^*R_P^*R_C^*Q_{\text{in}} \tag{170}$$

$$\frac{\partial Q_{\text{red}}}{\partial t} = k_I n\, m_{\text{RCII}}\left(\frac{Q_{\text{ox}}}{Q_{\text{T}}}\right)\left(1-a_{Q_{ox}}^*\left(1-R_C^*\right)\right) - k_I n m_{RCII}\frac{Q_{\text{red}}}{Q_T} \tag{171}$$

$$\frac{\partial Q_{\text{in}}}{\partial t} = -268\, m_{RCII}Q_{\text{in}} + k_I n m_{RCII}\frac{Q_{\text{red}}}{Q_T} \tag{172}$$

$$\frac{\partial[\text{ROS}]}{\partial t} = -f(T)R_N^*R_P^*R_C^*[\text{ROS}] + 32\frac{1}{10}k_I n\, m_{\text{RCII}}\left(\frac{Q_{\text{in}}}{Q_{\text{T}}}\right) \tag{173}$$

**Table 29.** Equations for symbiont reaction centre dynamics. Bleaching loss terms appear in Table 30.





$$\frac{\partial[\text{NH}_4]}{\partial t} = \min\left[\gamma, \max\left[0, \frac{[\text{ROS}] - [\text{ROS}_{threshold}]}{m_O}\right]\right] CSR_N^*/h_{wc} \tag{174}$$

$$\frac{\partial P}{\partial t} = \frac{1}{16}\frac{31}{14}\min\left[\gamma, \max\left[0, \frac{[\text{ROS}] - [\text{ROS}_{threshold}]}{m_O}\right]\right] CSR_P^*/h_{wc} \tag{175}$$

$$\frac{\partial DIC}{\partial t} = \frac{106}{16}\frac{12}{14}\min\left[\gamma, \max\left[0, \frac{[\text{ROS}] - [\text{ROS}_{threshold}]}{m_O}\right]\right] CSR_C^*/h_{wc} \tag{176}$$

$$\frac{\partial[\text{O}_2]}{\partial t} = -\frac{\partial DIC}{\partial t}\frac{32}{12}\frac{[\text{O}_2]^2}{K_{OA}^2 + [\text{O}_2]^2} \tag{177}$$

$$\frac{\partial[\text{COD}]}{\partial t} = \frac{\partial DIC}{\partial t}\frac{32}{12}\left(1 - \frac{[\text{O}_2]^2}{K_{OA}^2 + [\text{O}_2]^2}\right) \tag{178}$$

$$\frac{\partial CS}{\partial t} = -\min\left[\gamma, \max\left[0, \frac{[\text{ROS}] - [\text{ROS}_{threshold}]}{m_O}\right]\right] CS \tag{179}$$

$$\frac{\partial R_N}{\partial t} = -\min\left[\gamma, \max\left[0, \frac{[\text{ROS}] - [\text{ROS}_{threshold}]}{m_O}\right]\right] R_N \tag{180}$$

$$\frac{\partial R_P}{\partial t} = -\min\left[\gamma, \max\left[0, \frac{[\text{ROS}] - [\text{ROS}_{threshold}]}{m_O}\right]\right] R_P \tag{181}$$

$$\frac{\partial R_C}{\partial t} = -\min\left[\gamma, \max\left[0, \frac{[\text{ROS}] - [\text{ROS}_{threshold}]}{m_O}\right]\right] R_C \tag{182}$$

$$\frac{\partial Chl}{\partial t} = -\min\left[\gamma, \max\left[0, \frac{[\text{ROS}] - [\text{ROS}_{threshold}]}{m_O}\right]\right] Chl \tag{183}$$

$$\frac{\partial X_p}{\partial t} = -\min\left[\gamma, \max\left[0, \frac{[\text{ROS}] - [\text{ROS}_{threshold}]}{m_O}\right]\right] X_p \tag{184}$$

$$\frac{\partial X_h}{\partial t} = -\min\left[\gamma, \max\left[0, \frac{[\text{ROS}] - [\text{ROS}_{threshold}]}{m_O}\right]\right] X_h \tag{185}$$

$$\frac{\partial Q_{\text{ox}}}{\partial t} = -\min\left[\gamma, \max\left[0, \frac{[\text{ROS}] - [\text{ROS}_{threshold}]}{m_O}\right]\right] Q_{\text{ox}} \tag{186}$$

$$\frac{\partial Q_{\text{red}}}{\partial t} = -\min\left[\gamma, \max\left[0, \frac{[\text{ROS}] - [\text{ROS}_{threshold}]}{m_O}\right]\right] Q_{\text{red}} \tag{187}$$

$$\frac{\partial Q_{\text{in}}}{\partial t} = -\min\left[\gamma, \max\left[0, \frac{[\text{ROS}] - [\text{ROS}_{threshold}]}{m_O}\right]\right] Q_{\text{in}} \tag{188}$$

$$\frac{\partial D_{Red}}{\partial t} = \min\left[\gamma, \max\left[0, \frac{[\text{ROS}] - [\text{ROS}_{threshold}]}{m_O}\right]\right] CS/h_{wc} \tag{189}$$

**Table 30.** Equations describing the expulsion of zooxanthellae, and the resulting release of inorganic and organic molecules into the bottom water column layer.



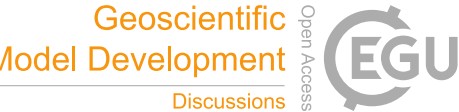

| | Symbol | Value |
|---|---|---|
| *Constants* | | |
| Molecular diffusivity of $NO_3$ | $D$ | $f(T, S_A) \sim 17.5 \times 10^{-10}$ m$^2$ s$^{-1}$ |
| Speed of light | $c$ | $2.998 \times 10^8$ m s$^{-1}$ |
| Planck constant | $h$ | $6.626 \times 10^{-34}$ J s$^{-1}$ |
| Avogadro constant | $A_V$ | $6.02 \times 10^{23}$ mol$^{-1}$ |
| [a]Pigment-specific absorption coefficients | $\gamma_\lambda$ | $f(\text{pig}, \lambda)$ m$^{-1}$ $\left(\text{mg m}^{-3}\right)^{-1}$ |
| Kinematic viscosity of water | $\nu$ | $f(T, S_A) \sim 1.05 \times 10^{-6}$ m$^2$ s$^{-1}$ |
| | | |
| *Parameters* | | |
| [b]Nitrogen content of zooxanthellae cells | $m_N$ | $5.77 \times 10^{-12}$ mol N cell$^{-1}$ |
| [c]Carbon content of zooxanthellae cells | $m_C$ | $(106/16)\, m_N$ mol C cell$^{-1}$ |
| [d]Maximum intracellular Chl concentration | $c_i^{\max}$ | $3.15 \times 10^6$ mg Chl m$^{-3}$ |
| Radius of zooxanthellae cells | $r_{CS}$ | 5 $\mu$m |
| Maximum growth rate of coral | $\mu_{CH}^{max}$ | 0.05 d$^{-1}$ |
| [e]Rate coefficient of particle capture | $S_{part}$ | 3.0 m d$^{-1}$ |
| Maximum growth rate of zooxanthellae | $\mu_{CS}^{max}$ | 0.4 d$^{-1}$ |
| Quadratic mortality coefficient of polyps | $\zeta_{CH}$ | 0.01 d$^{-1}$ (g N m$^{-2}$)$^{-1}$ |
| Linear mortality of zooxanthellae | $\zeta_{CS}$ | 0.04 d$^{-1}$ |
| [g]Remineralised fraction of coral mortality | $f_{remin}$ | 0.5 |
| Nitrogen-specific host area coefficient of polyps | $\Omega_{CH}$ | 2.0 m$^2$ g N$^{-1}$ |
| | | |
| Fractional (of $\mu_{CS}^{max}$) respiration rate | $\phi$ | 0.1 |

**Table 31.** Constants and parameter values used to model coral polyps. $V$ is zooxanthellae cell volume in $\mu$m$^3$. [a]Baird et al. (2016a),[c]Redfield et al. (1963) and Kirk (1994),[d]Finkel (2001),[e]Ribes and Atkinson (2007); Wyatt et al. (2010),[f,g]Gustafsson et al. (2013, 2014).





| | Symbol | Value |
|---|---|---|
| *Parameters* | | |
| Maximum growth rate of zooxanthellae | $\mu_{CS}^{max}$ | 1 d$^{-1}$ |
| Rate coefficient of xanthophyll switching | $\tau_{xan}$ | 1/600 s$^{-1}$ |
| [a]Atomic ratio of Chl a to RCII in *Symbiodinium* | $A_{\mathrm{RCII}}$ | 500 mol Chl mol RCII$^{-1}$ |
| [a]Stoichiometric ratio of RCII units to photons | $m_{\mathrm{RCII}}$ | 0.1 mol RCII mol photon$^{-1}$ |
| Maximum rate of zooxanthellae expulsion | $\gamma$ | 1 d$^{-1}$ |
| Oxygen half-saturation for aerobic respiration | $K_{OA}$ | 500 mg O m$^{-3}$ |
| Molar mass of Chl a | $M_{Chla}$ | 893.49 g mol$^{-1}$ |
| [b]Ratio of Chl a to xanthophyll | $\Theta_{chla2xan}$ | 0.2448 mg Chl mg X$^{-1}$ |
| [b]Ratio of Chl a to Chl c | $\Theta_{chla2chlc}$ | 0.1273 mg Chl-a mg Chl-c$^{-1}$ |
| [b]Ratio of Chl a to peridinin | $\Theta_{chla2per}$ | 0.4733 mg Chl mg$^{-1}$ |
| [b]Ratio of Chl a to $\beta$-carotene | $\Theta_{chla2caro}$ | 0.0446 mg Chl mg$^{-1}$ |
| [c]Lower limit of ROS bleaching | $[\mathrm{ROS}_{threshold}]$ | $5 \times 10^{-4}$ mg O cell$^{-1}$ |

**Table 32.** Constants and parameter values used in the coral bleaching model. [a]In Suggett et al. (2009). [b] ratio of constant terms in multivariate analysis in Hochberg et al. (2006). [c]Fitted parameter based on the existence of non-bleaching threshold (Suggett et al., 2009), and a comparison of observed bleaching and model output in the $\sim$1 km model.

$$g = k_{day}(\Omega_a - 1)(R_C^*)^2 + k_{night}(\Omega_a - 1) \tag{192}$$

where $g$ is the rate of net calcification, $k_{day}$ and $k_{night}$ are defined in Table 31 with habitat-specific values (Anthony et al., 2011; Mongin and Baird, 2014). The fluxes are scaled by the effective projected area of the community, $A_{eff}$. The power of 2 for $R_C^*$ ensures that generally light replete symbionts provide the host with sufficient energy for calcification.

### 4.4.3 Dissolution of shelf carbonate sands

In addition to the dissolution of carbonate sands on a growing coral reef, which is captured in the net dissolution quantified above, the marine carbonates on the continental shelf dissolve (Eyre et al., 2018). Like above, the dissolution of marine carbonates is approximated as a source of DIC and alkalinity but does not affect the properties (mass, porosity etc.) of the underlying sediments.

We assume carbonate dissolution from the sediment bed is proportional to the fraction of the total surface sediment is composed of either sand or mud carbonates. Other components, whose fraction do not release DIC and alkalinity, including carbonate gravel and non-carbonate mineralogies. Thus the change in $DIC$ and $A_T$ in the bottom water column layer is given





| Calcification | $Ca^{2+} + 2HCO_3^-$ | $\longrightarrow$ | $CaCO_3 + CO_2 + H_2O$ | (196) |
|---|---|---|---|---|
| | $\dfrac{\partial A_T}{\partial t}$ | $=$ | $-2gA_{eff}/h_{wc}$ | (197) |
| | $\dfrac{\partial DIC}{\partial t}$ | $=$ | $-12gA_{eff}/h_{wc}$ | (198) |
| | $g$ | $=$ | $k_{day}(\Omega_a - 1)(R_C^*)^2 + k_{night}(\Omega_a - 1)$ | (199) |
| | $\Omega_a$ | $=$ | $\dfrac{[CO_3^{2-}][Ca^{2+}]}{K_{sp}}$ | (200) |
| Dissolution | $CaCO_3 + CO_2 + H_2O$ | $\longrightarrow$ | $Ca^{2+} + 2HCO_3^-$ | (201) |
| | $\dfrac{\partial A_T}{\partial t}$ | $=$ | $2d_{CaCO_3}\left(\dfrac{Mud_{CaCO_3} + Sand_{CaCO_3}}{M}\right)/h_{wc}$ | (202) |
| | $\dfrac{\partial DIC}{\partial t}$ | $=$ | $12d_{CaCO_3}\left(\dfrac{Mud_{CaCO_3} + Sand_{CaCO_3}}{M}\right)/h_{wc}$ | (203) |
| | $d_{CaCO_3}$ | $=$ | $-11.51\Omega_a + 33.683$ | (204) |

Table 33. Equations for coral polyp calcification and dissolution. The concentration of carbonate ions, $[CO_3^{2-}]$, is determined from equilibrium carbon chemistry as a function of $A_T$, $DIC$, temperature and salinity, and the concentration of calcium ions, $[Ca^{2+}]$, is a mean oceanic value. 12 g C mol C$^{-1}$. Other constants and parameters are defined in Table 31.

by:

$$\frac{\partial DIC}{\partial t} = -12d_{CaCO_3}\left(\frac{Mud_{CaCO_3} + Sand_{CaCO_3}}{M}\right)/h_{wc} \tag{193}$$

$$\frac{\partial A_T}{\partial t} = -2d_{CaCO_3}\left(\frac{Mud_{CaCO_3} + Sand_{CaCO_3}}{M}\right)/h_{wc} \tag{194}$$

5  where $M$ is the total mass of surface layer inorganic sediments, $d_{CaCO_3}$ is the dissolution rate of CaCO$_3$, and is the reverse reaction to calcification and $h_{wc}$ is the thickness of the water column layer. The dissolution rate, $d_{CaCO_3}$ [mmol m$^{-2}$ d$^{-1}$] is assumed to be a function of $\Omega_a$ (Eyre et al., 2018):

$$d_{CaCO_3} = -11.51\Omega_a + 33.683 \tag{195}$$





| Name | Nom. size $\mu$m | Sinking vel. m d$^{-1}$ | Organic | Origin | Phosphorus adsorption | Colour |
|---|---|---|---|---|---|---|
| Gravel CaCO$_3$ | $10^4$ | 60,480 | N | I | N | W |
| Gravel non-CaCO$_3$ | $10^4$ | 60,480 | N | I | N | B |
| Sand CaCO$_3$ | $10^2$ | 172.8 | N | I | N | W |
| Sand non-CaCO$_3$ | $10^2$ | 172.8 | N | I | N | B |
| Mud CaCO$_3$ | 30 | 17.2 | N | I | Y | W |
| Mud non-CaCO$_3$ | 30 | 17.2 | N | I | Y | B |
| FineSed | 30 | 17.2 | N | C | Y | B |
| Dust | 1 | 1 | N | C | Y | B |
| $D_{Atk}$ | - | 10 | Y | OM | N | B |
| $D_{Red}$ | - | 10 | Y | OM | N | B |
| $D_C, D_N, D_P$ | - | 100 | Y | OM | N | B |

**Table 34.** Characteristics of the particulate classes. Y - Yes, N - No, I - initial condition, C - catchment, OM - remineralistion from organic matter, B - brown, W - white (Condie et al., 2009; Margvelashvili, 2009).

## 5 Processes in the sediments

### 5.1 Brief summary of processes in the sediments

The EMS model contains a multi-layered sediment compartment with time and space-varying vertical layers, and the same horizontal grid as the water column and epibenthic models. All state variables that exist in the water column layers have

5 an equivalent in the sediment layers (and are specified by `<variable name>_sed`). The dissolved tracers are given as a concentration in the porewater, while the particulate tracers are given as a concentration per unit volume (see Sec. 7.5.2).

The sediment model contains inorganic particles of different size (Dust, Mud, Sand and Gravel) and different mineralogies (carbonate and non-carbonate). The critical shear stress for resuspension, and the sinking rates, are generally larger for large particles, while and mineralogy only affects the optical properties. The size-class Dust comes only in a non-carbonate mineral-

10 ogy, and the Mud-carbonate class contains a category of FineSed-mineral that has the same physical and optical properties as Mud-mineral, except that it is initialised with a zero value and only enters the domain from rivers.

The organic matter classes are discussed in the Sec. 6.4. The inorganic and organic particulate classes are summarised in Table 34.





### 5.2 Sediment optical model

#### 5.2.1 Light absorption by benthic microalgae

The calculation of light absorption by benthic microalgae assumes they are the only attenuating component in a layer (biofilm) that lies on top layer of sediment, with a perfectly absorbing layer below and no scattering by any other components in the layer.

Thus no light penetrates through to the second sediment layer where benthic microalgae also reside. Thus the downwelling irradiance at wavelength $\lambda$ at the bottom of a layer, $E_{d,\lambda,bot}$, is given by:

$$E_{d,bot,\lambda} = E_{d,top,\lambda}e^{-n\alpha_\lambda h} \tag{205}$$

where $E_{d,top,\lambda}$ is the downwelling irradiance at wavelength $\lambda$ at the top of the layer and $\alpha_\lambda$ is the absorption cross-section of the cell at wavelength $\lambda$, and $n$ is the concentration of cells in the layer. The layer thickness used here, $h$, is the thickness of

the top sediment layer, so as to convert the concentration of cells in that layer, $n$, into the areal concentration of cells in the biofilm, $nh$.

Given no scattering in the cell, and that the vertical attenuation coefficient is independent of azimuth angle, the scalar irradiance that the benthic microalgae are exposed to in the surface biofilm is given by:

$$E_{o,\lambda} = (E_{d,top,\lambda} - E_{d,bot,\lambda})/(n\alpha_\lambda h) \tag{206}$$

The photons captured by each cell, and the microalgae process, follow the same equations as for the water column (Sec. 3.3.3).

#### 5.2.2 Bottom reflectance of macrophytes, benthic microalgae and sediment types

In the water column optical (Sec 3.2.3), the calculation of remote-sensing reflectance required the contribution to water-leaving irradiance from bottom reflection. In order to calculate the importance of bottom reflectance, the integrated weighting of the water column must be calculated (Sec. 3.2.3), with the remaining being ascribed to the bottom. Thus, the weighting of the

bottom reflectance as a component of surface reflectance is given by:

$$w_{\lambda,bot} = 1 - \frac{1}{z_{bot}} \int_0^{z_{bot}} \exp\left(-2K_{\lambda,z'}\right) dz' \tag{207}$$

where $K_\lambda$ is the attenuation coefficient at wavelength $\lambda$ described above, the factor of 2 accounts for the pathlength of both downwelling and upwelling light.

The bottom reflectance between 400 and 800 nm of $\sim 100$ substrates (including turtles and giant clams!) have been measured

on Heron Island using an Ocean Optics 2000 (Roelfsema and Phinn, 2012; Leiper et al., 2012). The data for selected substrates are shown in Fig 3 of Baird et al. (2016b). When the bottom is composed of mixed communities, the surface reflectance is weighted by the fraction of the end members visible from above, with the assumption that the substrates are layered from top to bottom by macroalgae, seagrass (*Zostera* then *Halophila*), corals (zooxanthellae then skeleton), benthic microalgae, and then sediments. Since the sediment is sorted in the simulation by the sediment process, the sediments are assumed to be well





mixed in surface sediment layer. Implicit in this formulation is that the scattering of one substrate type (i.e. benthic microalgae) does not contribute to the relectance of another (i.e. sand). In terms of an individual photon, it implies that if it first intercepts substrate A, then it is only scattered and/or absorbed by A.

*Calculation of bottom fraction.*

The fraction of the bottom taken up by a benthic plant of biomass $B$ is $A_{eff} = 1 - \exp(-\Omega_B B)$, with $\exp(-\Omega_B B)$ uncovered. Thus the fraction of the bottom covered by macroalgae, seagrass (Zostera then shallow and deep Halophila) and corals polyps is given by:

$$f_{MA} = 1 - \exp(-\Omega_{MA} MA) \tag{208}$$

$$f_{SG} = (1 - f_{MA})(1 - \exp(-\Omega_{SG} SG)) \tag{209}$$

$$f_{SGH} = (1 - f_{MA} - f_{SG})(1 - \exp(-\Omega_{SGH} SGH)) \tag{210}$$

$$f_{SGD} = (1 - f_{MA} - f_{SG} - f_{SGH})(1 - \exp(-\Omega_{SGD} SGD)) \tag{211}$$

$$f_{polyps} = (1 - f_{MA} - f_{SG} - f_{SGH} - f_{SGD})(1 - \exp(-\Omega_{CH} CH)) \tag{212}$$

Of the fraction of the bottom taken up by the polyps, $f_{polyps}$, zooxanthellae are first exposed. Assuming the zooxanthellae are horizontally homogeneous, the fraction taken up by the zooxanthellae is given by:

$$f_{zoo} = \min[f_{polyps}, \frac{\pi}{2\sqrt{3}} n\pi r_{zoo}^2] \tag{213}$$

where $\pi r^2$ is the projected area of the cell, $n$ is the number of cells, and $\pi/(2\sqrt{3}) \sim 0.9069$ accounts for the maximum packaging of spheres. Thus the zooxanthellae can take up all the polyp area. The fraction, if any, of the exposed polyp area remaining is assumed to be coral skeleton:

$$f_{skel} = f_{polyps} - \min[f_{polyps}, \frac{\pi}{2\sqrt{3}} n\pi r_{zoo}^2] \tag{214}$$

The benthic microalgae overlay the sediments. Following the zooxanthellae calculation above, the fraction taken up by benthic
microalgae is given by:

$$f_{MPB} = \min[(1 - f_{MA} - f_{SG} - f_{SGH} - f_{SGD} - f_{polyps}), \frac{\pi}{2\sqrt{3}} n\pi r_{MPB}^2] \tag{215}$$





Finally, the sediment fractions are assigned relative to their density in the surface layer, assuming the finer fractions overlay gravel:

$$M = Mud_{\mathrm{CaCO_3}} + Sand_{\mathrm{CaCO_3}} + Mud_{\mathrm{non-CaCO_3}} + Sand_{\mathrm{non-CaCO_3}} + FineSed + Dust \tag{216}$$

$$f_{CaCO_3} = (1 - f_{MA} - f_{SG} - f_{SGH} - f_{SGD} - f_{polyps} - f_{MPB})\left(\frac{Mud_{\mathrm{CaCO_3}} + Sand_{\mathrm{CaCO_3}}}{M}\right) \tag{217}$$

$$f_{non-CaCO_3} = (1 - f_{MA} - f_{SG} - f_{SGH} - f_{SGD} - f_{polyps} - f_{MPB})\left(\frac{Mud_{\mathrm{non-CaCO_3}} + Sand_{\mathrm{non-CaCO_3}}}{M}\right) \tag{218}$$

$$f_{FineSed} = (1 - f_{MA} - f_{SG} - f_{SGH} - f_{SGD} - f_{polyps} - f_{MPB})\left(\frac{FineSed + Dust}{M}\right) \tag{219}$$

with the porewaters not being considered optically-active. Now that the fraction of each bottom type has been calculated, the fraction of backscattering to absorption plus backscattering for the benthic surface as seen just below the surface, $u_{bot,\lambda}$, is given by:

$$
\begin{aligned}
u_{bot,\lambda} \quad = \quad & w_{\lambda,bot}(f_{MA}\rho_{MA,\lambda} \\
& + f_{SG}\rho_{SG,\lambda} \\
& + f_{SGH}\rho_{SGH,\lambda} \\
& + f_{SGD}\rho_{SGD,\lambda} \\
& + f_{zoo}\frac{b_{zoo,\lambda}}{a_{zoo,\lambda} + b_{zoo,\lambda}} \\
& + f_{skel}\rho_{skel,\lambda} \\
& + f_{MPB}\frac{b_{MPB,\lambda}}{a_{MPB,\lambda} + b_{MPB,\lambda}} \\
& + f_{\mathrm{CaCO_3}}\rho_{\mathrm{CaCO_3},\lambda} + f_{non-\mathrm{CaCO_3}}\rho_{non-\mathrm{CaCO_3},\lambda} + f_{FineSed}\rho_{non-\mathrm{CaCO_3},\lambda})
\end{aligned}
\tag{220}
$$

where the absorption and backscattering are calculated as given in Sec. 3.2.1, and $\rho$ is the measured bottom refectance of each end member (Dekker et al., 2011; Hamilton, 2001; Reichstetter et al., 2015).

For the values of surface reflectance for sand and mud from Heron Island (Roelfsema and Phinn, 2012; Leiper et al., 2012), and microalgal optical properties calculated as per Sec. 3.2, a ternary plot can be used to visualise the changes in true colour with sediment composition (Fig 13 of Baird et al. (2016b)).

It is important to note that while the backscattering of light from the bottom is considered in the model for the purposes of calculating reflectance (and therefore comparing with observations), it is not included in the calculation of water column scalar irradiance, which would require a radiative transfer model (Mishchenko et al., 2002).

10000





| Variable | Symbol | Units |
|---|---|---|
| Ammonia concentration | $[NH_4]$ | mg N m$^{-3}$ |
| Sediment Dissolved Inorganic Carbon (DIC) | $DIC$ | mg C m$^{-3}$ |
| Sediment Dissolved Inorganic Phosphorus (DIP) | $P$ | mg P m$^{-3}$ |
| Sediment Particulate Inorganic Phosphorus (PIP) | $PIP$ | mg P m$^{-3}$ |
| Sediment Immobolised Particulate Inorganic Phosphorus (PIPI) | $PIPI$ | mg P m$^{-3}$ |
| Sediment Non-Algal Particulates (NAP) | $NAP$ | kg m$^{-3}$ |
| Sediment dissolved oxygen concentration | $[O_2]$ | mg O m$^{-3}$ |

**Table 35.** State and derived variables for the sediment inorganic chemistry model.

| Description | Symbol | Units |
|---|---|---|
| Maximum rate of nitrification in the water column | $\tau_{nit,wc}$ | 0.1 d$^{-1}$ |
| Maximum rate of nitrification in the sediment | $\tau_{nit,sed}$ | 20 d$^{-1}$ |
| Oxygen half-saturation constant for nitrification | $K_{O_2,nit}$ | 500 mg O m$^{-3}$ |
| Maximum rate of denitrification | $\tau_{denit}$ | 0.8 d$^{-1}$ |
| Oxygen half-saturation constant for de-nitrification | $K_{O_2,denit}$ | 10000 mg O m$^{-3}$ |
| Rate of P adsorbed/desorbed equilibrium | $\tau_{Pabs}$ | 0.04 d$^{-1}$ |
| Isothermic const. P adsorption for NAP | $k_{Pads,wc}$ | 300 kg NAP$^{-1}$ |
| Oxygen half-saturation for P adsorption | $K_{O_2,abs}$ | 2000 mg O m$^{-3}$ |
| Rate of P immobilisation | $\tau_{Pimm}$ | 0.0012 d$^{-1}$ |

**Table 36.** Constants and parameter values used in the sediment inorganic chemistry.

## 5.3 Sediment chemistry

### 5.3.1 Sediment nitrification - denitrification

Nitrification in the sediment is similar to the water-column, but with a sigmoid rather than hyperbolic relationship at low oxygen, for numerical reasons. Denitrification occurs only in the sediment.

### 5.3.2 Sediment phosphorus absorption - desorption

Sediment phosphorus absorption - desorption is similar to water column.

There is an additional pool of immobilised particulate inorganic phosphorus, $PIPI$, which accumulates in the model over time as $PIP$ becomes immobilised, and represents permanent sequestration.





$$\text{Nitrification}: \text{NH}_4^+ + 2\text{O}_2 \quad \longrightarrow \quad \text{NO}_3^- + \text{H}_2\text{O} + 2\text{H}^+ \tag{221}$$

$$\text{De} - \text{nitrification}: \text{NO}_3^- + \frac{1}{2}\text{O}_2 \quad \longrightarrow \quad \frac{1}{2}\text{N}_{2(\text{g})} + 2\text{O}_2 \tag{222}$$

$$\tag{223}$$

$$\frac{\partial [\text{NH}_4]}{\partial t} = -\tau_{nit,wc}[\text{NH}_4]\frac{[\text{O}_2]^2}{K_{\text{O}_2,nit}^2 + [\text{O}_2]^2} \tag{224}$$

$$\frac{\partial [\text{O}_2]}{\partial t} = -2\frac{32}{14}\tau_{nit,wc}[\text{NH}_4]\frac{[\text{O}_2]^2}{K_{\text{O}_2,nit}^2 + [\text{O}_2]^2} + 2\frac{32}{14}\tau_{denit}[\text{NO}_3]\frac{K_{\text{O}_2,denit}}{K_{\text{O}_2,denit} + [\text{O}_2]} \tag{225}$$

$$\frac{\partial [\text{NO}_3]}{\partial t} = \tau_{nit,wc}[\text{NH}_4]\frac{[\text{O}_2]^2}{K_{\text{O}_2,nit}^2 + [\text{O}_2]^2} - \tau_{denit}[\text{NO}_3]\frac{K_{\text{O}_2,denit}}{K_{\text{O}_2,denit} + [\text{O}_2]} \tag{226}$$

$$\frac{\partial P}{\partial t} = \left(\tau_{Pabs}\left(\frac{PIP}{k_{Pads,sed}NAP} - \frac{[\text{O}_2]P}{K_{\text{O}_2,\text{abs}} + [\text{O}_2]}\right)\right)/\phi \tag{227}$$

$$\frac{\partial PIP}{\partial t} = -\tau_{Pabs}\left(\frac{PIP}{k_{Pads,wc}NAP} - \frac{[\text{O}_2]P}{K_{\text{O}_2,\text{abs}} + [\text{O}_2]}\right) - \tau_{Pimm}PIP \tag{228}$$

$$\frac{\partial PIPI}{\partial t} = \tau_{Pimm}PIP \tag{229}$$

**Table 37.** Equations for the sediment inorganic chemistry.

## 6 Common water / epibenthic / sediment processes

### 6.1 Preferential uptake of ammonia

The model contains two forms of dissolved inorganic nitrogen (DIN), dissolved ammonia ($\text{NH}_4$) and dissolved nitrate ($\text{NO}_3$):

$$N = [\text{NH}_4] + [\text{NO}_3] \tag{230}$$

5   where $N$ is the concentration of DIN, $[\text{NH}_4]$ is the concentration of dissolved ammonia and $[\text{NO}_3]$ is the concentration of nitrate. In the model, the ammonia component of the DIN pool is assumed to be taken up first by all primary producers, followed by the nitrate, with the caveat that the uptake of ammonia cannot exceed the diffusion limit for ammonia. The underlying principle of this assumption is that photosynthetic organisms can entirely preference ammonia, but that the uptake of ammonia is still limited by diffusion to the organism's surface.

10   As the nitrogen uptake formulation varies for the different autotrophs, the formulation of the preference of ammonia also varies. The diffusion coefficient of ammonia and nitrate are only 3 % different, so for simplicity we have used the nitrate diffusion coefficient for both.




Thus, for microalgae with internal reserves of nitrogen, the partitioning of nitrogen uptake is given by:

$$\frac{\partial N}{\partial t} = -\psi D_N N (1 - R_N^*)(B/m_N) \tag{231}$$

$$\frac{\partial [\text{NH}_4]}{\partial t} = -\min\left[\psi D_N N(1-R_N^*), \psi D_N[\text{NH}_4]\right](B/m_N) \tag{232}$$

$$\frac{\partial [\text{NO}_3]}{\partial t} = -\left(\psi D_N N(1-R_N^*) - \min\left[\psi D_N N(1-R_N^*), \psi D_N[\text{NH}_4]\right]\right)(B/m_N) \tag{233}$$

For macroalgae, which also have diffusion limits to uptake, but are not represented with internal reserves of nitrogen, the terms are:

$$\frac{\partial N}{\partial t} = -\mu_{MA} MA \tag{234}$$

$$\frac{\partial [\text{NH}_4]}{\partial t} = -\min\left[SA_{eff}[\text{NH}_4], \mu_{MA} MA\right] \tag{235}$$

$$\frac{\partial [\text{NO}_3]}{\partial t} = -\left(\mu_{MA} MA - \min\left[SA_{eff}[\text{NH}_4], \mu_{MA} MA\right]\right) \tag{236}$$

Zooxanthellae is a combination of the two cases above, because in the model they contain reserves like microalgae, but the uptake rate is across a 2D surface like macroalage.

In the case of nutrient uptake by seagrass, which has a saturating nitrogen uptake functional form, the terms are:

$$\frac{\partial N_s}{\partial t} = -\mu_{SG} SG \tag{237}$$

$$\frac{\partial [\text{NH}_4]_s}{\partial t} = -\min\left[\mu_{SG} SG, \frac{\mu_{SG}^{max}[\text{NH}_4]_s SG}{K_N + [\text{NH}_4]_s}\right] \tag{238}$$

$$\frac{\partial [\text{NO}_3]_s}{\partial t} = -\left(\mu_{SG} SG - \min\left[\mu_{SG} SG, \frac{\mu_{SG}^{max}[\text{NH}_4]_s SG}{K_N + [\text{NH}_4]_s}\right]\right) \tag{239}$$

where $K_N$ is a function of the ratio of above ground to below ground biomass described in Baird et al. (2016a).

One feature worth noting is that the above formulation for preferential ammonia uptake requires no additional parameters, which is different to other classically applied formulations (Fasham et al., 1990) that require a new parameter, potentially for each autotroph. This simple use of the geometric constraint has an important role in reducing model complexity.

## 6.2 Oxygen release during nitrate uptake

For all autotrophs, the uptake of a nitrate ion results in the retention of the one nitrogen atom in their reserves or structural material, and the release of the three oxygen atoms into the water column or porewaters.

$$\frac{\partial [\text{O}]}{\partial t} = -\frac{48}{14}\frac{\partial [\text{NO}_3]}{\partial t} \tag{240}$$

The oxygen that is part of the structural material is assumed to have been taken up through photosynthesis.





### 6.3 Temperature dependence of ecological rates

Physiological rate parameters (maximum growth rates, mortality rates, remineralisation rates) have a temperature dependence that is determined from:

$$r_T = r_{Tref} Q_{10}^{(T-T_{ref})/10} \tag{241}$$

where $r_T$ is the physiological rate parameter (e.g. $\mu$, $\zeta$ etc.) at temperature $T$, $T_{ref}$ is the reference temperature (nominally 20°C for GBR), $r_T$ the physiological rate parameter at temperature $T_{ref}$, $Q_{10}$ is the Q10 temperature coefficient and represents the rate of change of a biological rate as a result of increasing temperature by 10°C.

Note that while physiological rates may be temperature-dependent, the ecological processes they are included in may not. For example, for extremely light-limited growth, all autotrophs capture light at a rate independent of temperature. With the

10 reserves of nutrients replete, the steady-state realised growth rate, $\mu$, becomes the rate of photon capture, $k$. This can be shown algebracially: $\mu = \mu^{max} R_C^* = k(1 - R^*)$, where $R^*$ is the reserves of carbon. Rearranging, $R^* = k/(\mu^{max} + k)$. At $k << \mu^{max}$, $R^* = k/\mu^{max}$, thus $\mu = \mu^{max} k/\mu^{max} = k$. This corresponds with observations of no temperature dependence of photosynthesis at low light levels (Kirk, 1994).

Similar arguments show that extremely nutrient limited autorophs will have the same temperature dependence to that of

15 the diffusion coefficient. Thus, the autotroph growth model has a temperature-dependence that adjust appropriately to the physiological condition of the autotroph, and is a combination of constant, exponential, and polynomial expressions.

Physiological rates in the model that are not temperature dependent are: mass transfer rate constant for particulate grazing by corals, $S_{Part}$; net coral calcification $g$; maximum chlorophyll synthesis, $k_{Chl}^{max}$; and rate of translocation between leaves and roots in seagrass, $\tau_{tran}$.

### 20 6.4 Detritus remineralisation

The non-living components of C, N, and P cycles include the particulate labile and refractory pools, and a dissolved pool (Fig. 5). The labile detritus has a pool at the Redfield ratio, $D_{Red}$, and at the Atkinson ratio, $D_{Atk}$, resulting from dead organic matter at these ratios. The labile detritus from both pools then breaks down into refractory detritus and dissolved organic matter. The refractory detritus and dissolved organic matter pools are quantified by individual elements (C, N, P), in order to account

for the mixed source of labile detritus. Finally, a component of the breakdown of each of these pools is returned to dissolved inorganic components. The variables, parameters and equations can be found in Tables 38, 40 & 39 respectively.

As the refractory and dissolved components are separated into C, N and P components, this introduces the possibility to have P components break down quicker than C and N. This is specified as the breakdown rate of P relative to N, $\Phi_{RD_P}$ and $\Phi_{DOM_P}$ respectively for refractory and dissolved detritus respectively.





| Variable | Symbol | Units |
|---|---|---|
| Ammonia concentration | $[NH_4]$ | mg N m$^{-3}$ |
| Dissolved Inorganic Carbon (DIC) | $DIC$ | mg C m$^{-3}$ |
| Dissolved Inorganic Phosphorus (DIP) | $P$ | mg P m$^{-3}$ |
| Dissolved oxygen concentration | $[O_2]$ | mg O m$^{-3}$ |
| Labile detritus at Redfield ratio | $D_{Red}$ | mg N m$^{-3}$ |
| Labile detritus at Atkinson ratio | $D_{Atk}$ | mg N m$^{-3}$ |
| Refractory Detritus C | $D_C$ | mg C m$^{-3}$ |
| Refractory Detritus N | $D_N$ | mg N m$^{-3}$ |
| Refractory Detritus P | $D_P$ | mg P m$^{-3}$ |
| Dissolved Organic C | $O_C$ | mg C m$^{-3}$ |
| Dissolved Organic N | $O_N$ | mg N m$^{-3}$ |
| Dissolved Organic P | $O_P$ | mg P m$^{-3}$ |
| Chemical Oxygen Demand (COD) | $COD$ | mg O m$^{-3}$ |

**Table 38.** State and derived variables for the detritus remineralisation model in both the sediment and water column.

### 6.4.1 Anaerobic and anoxic respiration

The processes of remineralisation, phytoplankton mortality and zooplankton grazing return carbon dioxide to the water column. In oxic conditions, these processes consume oxygen in a ratio of $DIC : \frac{32}{12}[O_2]$. At low oxygen concentrations, the oxygen consumed is reduced:

$$\frac{\partial [O_2]}{\partial t} = -\frac{\partial DIC}{\partial t}\frac{32}{12}\frac{[O_2]^2}{K_{OA}^2 + [O_2]^2} \tag{255}$$

where $K_{OA}$ = 256 mg O m$^{-3}$ is the half-saturation constant for anoxic respiration (Boudreau, 1996). A sigmoid saturation term is used because it is more numerically stable as the oxygen concentration approaches 0. The anoxic component of remineralisation results in an increased chemical oxygen demand (COD):

$$\frac{\partial COD}{\partial t} = \frac{\partial DIC}{\partial t}\frac{32}{12}\left(1 - \frac{[O_2]^2}{K_{OA}^2 + [O_2]^2}\right) \tag{256}$$

COD is a dissolved tracer, with the same units as oxygen.

When oxygen and COD co-exist they react to reduce both, following:

$$\frac{\partial [O_2]}{\partial t} = -\tau_{COD}\min[COD, 8000]\frac{[O_2]}{8000} \tag{257}$$

$$\frac{\partial COD}{\partial t} = -\tau_{COD}\min[COD, 8000]\frac{[O_2]}{8000} \tag{258}$$





$$\frac{\partial D_{Red}}{\partial t} = -r_{Red}D_{Red} \tag{242}$$

$$\frac{\partial D_{Atk}}{\partial t} = -r_{Atk}D_{Atk} \tag{243}$$

$$\frac{\partial D_C}{\partial t} = \frac{106}{16}\frac{12}{14}\zeta_{Red}r_{Red}D_{Red} + \frac{550}{30}\frac{12}{14}\zeta_{Atk}r_{Atk}D_{Atk} - r_RD_C \tag{244}$$

$$\frac{\partial D_N}{\partial t} = \zeta_{Red}r_{Red}D_{Red} + \zeta_{Atk}r_{Atk}D_{Atk} - r_RD_N \tag{245}$$

$$\frac{\partial D_P}{\partial t} = \frac{1}{16}\frac{31}{14}\zeta_{Red}r_{Red}D_{Red} + \frac{1}{30}\frac{31}{14}\zeta_{Atk}r_{Atk}D_{Atk} - \Phi_{RD_P}r_RD_P \tag{246}$$

$$\frac{\partial O_C}{\partial t} = \frac{106}{16}\frac{12}{14}\vartheta_{Red}r_{Red}D_{Red} + \frac{550}{30}\frac{12}{14}\vartheta_{Atk}r_{Atk}D_{Atk} + \vartheta_{Ref}r_RD_C - r_OO_C \tag{247}$$

$$\frac{\partial O_N}{\partial t} = \vartheta_{Red}r_{Red}D_{Red} + \vartheta_{Atk}r_{Atk}D_{Atk} + \vartheta_{Ref}r_RD_N - r_OO_N \tag{248}$$

$$\frac{\partial O_P}{\partial t} = \frac{1}{16}\frac{31}{14}\vartheta_{Red}r_{Red}D_{Red} + \frac{1}{30}\frac{31}{14}\vartheta_{Atk}r_{Atk}D_{Atk} + \vartheta_{Ref}\Phi_{RD_P}r_RD_P - \Phi_{DOM_P}r_OO_P \tag{249}$$

$$\frac{\partial [\text{NH}_4]}{\partial t} = r_{Red}D_{Red}(1-\zeta_{Red}-\vartheta_{Red}) \\ + r_{Atk}D_{Atk}(1-\zeta_{Atk}-\vartheta_{Atk}) + r_RD_N(1-\vartheta_{Ref}) + r_OO_N \tag{250}$$

$$\frac{\partial DIC}{\partial t} = \frac{106}{16}\frac{12}{14}r_{Red}D_{Red}(1-\zeta_{Red}-\vartheta_{Red}) \\ + \frac{550}{30}\frac{12}{14}r_{Atk}D_{Atk}(1-\zeta_{Atk}-\vartheta_{Atk}) + r_RD_C(1-\vartheta_{Ref}) + r_OO_C \tag{251}$$

$$\frac{\partial P}{\partial t} = \frac{1}{16}\frac{31}{14}r_{Red}D_{Red}(1-\zeta_{Red}-\vartheta_{Red}) \\ + \frac{1}{30}\frac{31}{14}r_{Atk}D_{Atk}(1-\zeta_{Atk}-\vartheta_{Atk}) + \Phi_{RD_P}r_RD_P(1-\vartheta_{Ref}) + \Phi_{DOM_P}r_OO_P \tag{252}$$

$$\frac{\partial [\text{O}_2]}{\partial t} = -\frac{32}{12}\frac{\partial DIC}{\partial t}\frac{[\text{O}_2]^2}{K_{OA}^2 + [\text{O}_2]^2} \tag{253}$$

$$\frac{\partial [COD]}{\partial t} = \frac{32}{12}\frac{\partial DIC}{\partial t}\left(1-\frac{[\text{O}_2]^2}{K_{OA}^2 + [\text{O}_2]^2}\right) \tag{254}$$

**Table 39.** Equations for detritus remineralisation in the water column and sediment.





| Description | Symbol | Red | Atk | Refractory | Dissolved |
|---|---|---|---|---|---|
| Detritus breakdown rate (d$^{-1}$) | $r_{Red,Atk,R,O}$ | 0.04 | 0.01 | 0.001 | 0.0001 |
| Fraction of detritus to refractory | $\zeta_{Red,Atk}$ | 0.19 | 0.19 | - | - |
| Fraction of detritus to DOM | $\vartheta_{Red,Atk,R}$ | 0.1 | 0.1 | 0.05 | |
| Breakdown rate of P relative to N | $\Phi_{R,O}$ | N/A | N/A | 2 | 2 |

**Table 40.** Constants and parameter values used in the water column detritus remineralisation model. Red = Redfield ratio (C:N:P = 106:16:1); Atk = Atkinson ratio (C:N:P = 550:30:1); Ref = Refractory. See L∅nborg et al. (2017).

| | Labile Det., $D_{Red}$ | Refractory Det., $D$ | Dissolved Organic, $O$ |
|---|---|---|---|
| Redfield | 25 | - | - |
| Carbon | - | 27 | 767 |
| Nitrogen | - | 4.75 | 135 |
| Phosphorus | - | 0.66 | 18.7 |

**Table 41.** Steady-state detrital and dissolved organic C, N and P concentrations for primary production equal to 2 mg N m$^{-1}$

where 8000 mg O m$^{-3}$ is approximately the saturation concentration of oxygen in seawater, and $\tau_{COD}$ is the timescale of this reduction. The term $\min[COD, 8000]$ is required because $COD$ represents the end stage of anoxic reduction and can become large for long simulations. Even with this limitation, if $\tau_{COD} = 1$ hr$^{-1}$, the processes in Eqs. 257 and 258 proceed faster than most of the other porewater processes.

# 7 Numerical integration

## 7.1 Splitting of physical and ecological integrations

The numerical solution of the time-dependent advection-diffusion-reaction equations for each of the ecological tracers is implemented through sequential solving of the partial differential equations (PDEs) for advection and diffusion, and the ordinary differential equations for reactions. This technique, called operator splitting, is common in geophysical science (Hundsdorfer and Verwer, 2003).

The time-step of the splitting is typically 15 min - 1 hour (Table 42). Under the sequential operator splitting technique used, first the advection-diffusion processes are solved for the period of the time-step. The value of the tracers at the end of this PDE integration, and the initial time, are then used as initial conditions for the ODE integration. After the ODE integration has run for same time period, the value of the tracers is update, and time is considered to have moved forward just one time-step. The integration continues to operate sequentially for the whole model simulation.

The PDE solutions are described in the physical model description available at:





www.emg.cmar.csiro.au/www/en/emg/software/EMS/hydrodynamics.html.

| Description | Values |
| --- | --- |
| Timestep of hydrodynamic model | 90 s (GBR4), 20 s (GBR1) |
| $^a$Timestep of ODE ecological model | 3600 s (GBR4), 1800 s (GBR1) |
| Timestep of optical and carbon chemistry models | 3600 s |
| Optical model resolution in PAR | $\sim 20$ nm |
| ODE integrator | 5th order Dormand-Prince |
| ODE tolerance | $10^{-5}$ mg N m$^{-3}$ |
| Maximum number of ODE steps in ecology | 2000 |
| Maximum number of iterations in carbon chemistry | 100 |
| Accuracy of carbon chemistry calculations | $[H^+] = 10^{-12}$ mol |

**Table 42.** Integration details. Optical wavelengths (nm): 290 310 330 350 370 390 410 430 440 450 470 490 510 530 550 570 590 610 630 650 670 690 710 800.$^a$Since the integrator is 5th order, the ecological derivatives are evaluated at least every approximately $3600/5 = 900$ s, and more regularly for stiff equations.

## 7.2 Diffusive exchange of dissolved tracers across sediment-water interface

Due to the thin surface sediment layer, and the potentially large epibenthic drawndown of porewater dissolved tracers, the exchange of dissolved tracers between the bottom water column layer and the top sediment layer is solved in the same numerical operation as the ecological tracers (other transport processes occurring between ecological timesteps). The flux, $J$, is given by:

$$J = k(C_s - C) \tag{259}$$

where $C$ and $C_s$ are the concentration in water column and sediment respectively, $k = 4.6 \times 10^{-7}$ m s$^{-1}$ is the transfer coefficient. In the model parameterisation, $k = D/h$ where $D = 3 \times 10^{-9}$ m$^2$ s$^{-1}$ is the diffusion coefficient and $h = 0.0065$ mm is the thickness of the diffusive layer.

While in reality $k$ would vary with water column and sediment hydrodynamics as influenced by community type etc, these complexities has not been considered. In addition to the diffusive flux between the sediment and water column, particulate deposition entrains water column water into the sediments, and particulate resuspension releases porewaters into the water column. Sediment model details can be found at: https://research.csiro.au/cem/software/ems/ems-documentation/.

## 7.3 Optical integration

The inherent and apparent optical properties are calculated between the physical and ecological integrations. The spectral resolution of 25 wavebands has been chosen to resolve the absorption peaks associated with Chl $a$, and to span the optical





wavelengths. As IOPs can be calculated at any wavelength given the model state, IOPs and AOPs at observed wavelengths are recalculated after the integration.

Additionally, the wavelengths integrated have been chosen such that the lower end of one waveband and the top end of another fall on 400 and 700 nm respectively, allowing precise calculation of photosynthetically available radiation (PAR).

## 7.4 Adaptive solution of ecological processes

A 5th-order Dormand-Prince ordinary differential equation integrator (Dormand. and Prince, 1980) with adaptive step control is used to integrate the local rates of changes due to ecological processes. This requires 7 function evaluations for the first step and 6 for each step after. A tolerance of $1 \times 10^{-5}$ mg N m$^{-3}$ is required for the integration step to be accepted.

For an $n_{wc}$-layer water column and $n_{sed}$-layer sediment, the integrator sequentially solves the top $n_{wc} - 1$ water column layers; the $n$th water column layer, epibenthic and top sediment layer together; and then the $n_{sed} - 1$ to bottom sediment layers.

## 7.5 Additional integration details

### 7.5.1 Approximation of stoichiometric coefficients

In this model description we have chosen to explicitly include atomic mass as two significant figure values, so that the conversion are more readable in the equations than if they had all been rendered as mathematical symbols. Nonetheless these values are more precisely given in the numerical code (Table 43).

It is worth remembering that the atomic masses are approximations assuming the ratio of isotopes found in the Periodic Table (Atkins, 1994), based on the natural isotopic abundance of the Earth. So, for example, $^{14}$N and $^{15}$N have atomic masses of 14.00307 and 15.00011 respectively, with $^{14}$N making up 99.64 % of the abundance on Earth. Thus the value 14.01 comes from $14.00307 \times 99.64 + 15.00011 \times 0.36 = 14.0067$. The isotopic discrimination in the food web of 3 ppt per trophic level would increase the mean atomic mass by $(15.00011 - 14.00307) \times 0.003 = 0.003$ per trophic level. Perhaps more importantly, if the model had state variables for $^{14}$N and $^{15}$N, then the equations would change to contain coefficients of 14 for the $^{14}$N isotope equations, and 15 for the $^{15}$N isotope equations, that would be applied in the numerical code using 14.00 and 15.00 respectively.

| Element | Value in symbolic equations | Value in code |
|---|---|---|
| Nitrogen, N | 14 | 14.01 |
| Carbon, C | 12 | 12.01 |
| Oxygen, O$_2$ | 32 | 32.00 |
| Phosphorus, P | 31 | 30.97 |

**Table 43.** Atomic mass of the C, N, P and O$_2$, both in the model description where two significant figures are used for brevity, and in the numerical code, where precision is more important.





### 7.5.2 Mass conservation in water column and sediment porewaters

The model checks the conservation of Total C, $TC$, Total N, $TN$, Total P, $TP$, and oxygen, $[O_2]$, within each grid cell at each time step using the following conservation laws. To establish mass conservation, the sum of the change in mass (of N, P, C and O) with time and the mass of sinks / sources (such as sea-air fluxes, denitrification) must equate to zero.

The total mass and conservation equations are same for the water column and porewaters, with the caveats that (1) air-sea fluxes only affect surface layers of the water, (2) denitrification only occurs in the sediment, and (3) the porosity, $\phi$, of the water column is 1. In the sediment, the concentration of particulates is given in per unit volume of space, while the concentration of dissolved tracers is given in per unit volume of porewater. The concentration of dissolved tracer, $X$, per unit space is given by $\phi X$.

Thus the total carbon in a unit volume of space, and its conservation, are given by:

$$TC = \phi\left(DIC + O_C\right) + \left(\frac{550}{30}\frac{12}{14}D_{Atk} + D_C + \frac{106}{16}\frac{12}{14}\left(D_{red} + \sum B(1 + R_C^*) + \sum Z\right)\right) \qquad (260)$$

$$\frac{\partial TC}{\partial t} + \underbrace{k_{\mathrm{CO_2}}\left([\mathrm{CO_2}] - [\mathrm{CO_2}]_{atm}\right)/h}_{sea-air\ flux} = 0 \qquad (261)$$

The total nitrogen in a unit volume of space, and its conservation, are given by:

$$TN = \phi\left([\mathrm{NO_3}] + [\mathrm{NH_4}] + O_N\right) + \left(D_{Atk} + D_{red} + D_N + \sum B(1 + R_N^*) + \sum Z\right) \qquad (262)$$

$$\frac{\partial TN}{\partial t} + \left(\text{denitrification} - \text{nitrogen fixation}\right)/\phi - \text{dust input}/h = 0 \qquad (263)$$

The total phosphorus in a unit volume of space, and its conservation, are given by:

$$TP = \phi\left(DIP + O_P\right) + PIP + PIPI + \frac{1}{30}\frac{31}{14}D_{Atk} + D_P + \frac{1}{16}\frac{31}{14}\left(D_{red} + \sum B(1 + R_P^*) + \sum Z\right) \qquad (264)$$

$$\frac{\partial TP}{\partial t} - \text{dust input}/h = 0 \qquad (265)$$

The concept of oxygen conservation in the model is more subtle than that of C, N and P due to the mass of oxygen in the water molecules themselves not being considered. When photosynthesis occurs, C is transferred from the dissolved phase to reserves within the cell. With both dissolved and particulate pools considered, mass conservation of C is straightforward. In contrast

to C, during photosynthesis oxygen is drawn from the water molecules (i.e. $H_2O$), whose mass is not being considered, and released into the water column. Conversely, when organic matter is broken down oxygen is consumed from the water column and released as $H_2O$.





In order to obtain a mass conservation for oxygen, the concept of Biological Oxygen Demand (BOD) is used. Often BOD represents the biological demand for oxygen in say a 5 day incubation, $BOD_5$. Here, for the purposes of mass conservation checks, we use $BOD_\infty$, the oxygen demand over an infinite time for breakdown. This represents the total oxygen removed from the water molecules for organic matter creation.

Anaerobic respiration reduces $BOD_\infty$ without reducing $O_2$, but instead creating reduced-oxygen species. This is accounted for in the oxygen balance by the prognostic tracer Chemical Oxygen Demand (COD). In other biogeochemical modelling studies this is represented by a negative oxygen concentration.

Thus at any time point the biogeochemical model will conserve the oxygen concentration minus $BOD_\infty$ minus COD, plus or minus any sources and sinks such as sea-air fluxes. The total oxygen minus $BOD_\infty$ minus COD in a unit volume of water,
and its conservation, is given by:

$$[O_2] + \frac{48}{14}[NO_3] - BOD_\infty - COD =$$

$$\phi \left( [O_2] + \frac{48}{14}[NO_3] - COD + \frac{32}{12}O_C \right) - \left( \frac{550}{30}\frac{32}{14}D_{Atk} + \frac{32}{12}D_C + \frac{106}{16}\frac{32}{14}\left( D_{red} + \sum B_N(1 + R_C^*) \right) \right) \qquad (266)$$

$$\frac{\partial([O_2] + \frac{48}{14}[NO_3] - BOD_\infty - COD)}{\partial t} + \mathcal{R} - \overbrace{\frac{k_{O_2}([O_2]_{sat} - [O_2])}{h}}^{sea-air\ flux} - \underbrace{2\frac{106}{16}\frac{32}{14}\tau_{nit,wc}[NH_4]\frac{[O_2]}{K_{nit,O} + [O_2]}}_{nitrification} = 0 \qquad (267)$$

where $\mathcal{R}$ is respiration of organic matter.

In addition to dissolved oxygen, BOD and COD, nitrate ($NO_3$) appears in the oxygen mass balance. This is necessary because the N associated with nitrate uptake is not taken into the autotrophs, but rather released into the water column or porewater. Other entities that contain oxygen in the ocean include the water molecule ($H_2O$) and the phosphorus ion ($PO_4$).
In the case of water, this oxygen reservoir is considered very large, with the small flux associated with its change balanced by BOD. In the case of $PO_4$, this is a small reservoir. As oxygen remains bound to P through the entire processes of uptake into reserves and incorporated into structural material and then release, it is not necessary to include it in the oxygen balance for the purposes of ensuring consistency. Nonetheless, strictly the water column and porewater oxygen reservoirs could include a term $+\frac{64}{31}[PO_4]$, and the BOD would have similar quantities for reserves and structural material.

### 7.5.3 Mass conservation in the epibenthic

Mass conservation in the epibenthos requires consideration of fluxes between the water column, porewaters and the epibenthic organisms (macroalgae, seagrass and coral hosts and symbionts).





The total carbon in the epibenthos, and its conservation, is given by:

$$TC = \frac{550}{30}\frac{12}{14}\left(MA + SG_A + SG_B\right) + \frac{106}{16}\frac{12}{14}\left(CS\left(1 + R_C^*\right) + CH\right) \tag{268}$$

$$\left.\frac{\partial TC}{\partial t}\right|_{epi} + h_{wc}\left.\frac{\partial TC}{\partial t}\right|_{wc} + h_{sed}\left.\frac{\partial TC}{\partial t}\right|_{sed} + \underbrace{12\left(gA_{eff} - d_{\mathrm{CaCO_3}}\right)}_{coral\ calcification\ -\ dissolution} = 0 \tag{269}$$

where $h_{wc}$ and $h_{sed}$ are the thickness of the bottom water column and top sediment layers, $R_C^*$ is the normalised internal reserves of carbon in zooxanthallae, $12g$ is the rate coral calcification per unit area of coral, $A_{eff}$ is the area of the bottom covered by coral per m$^{-2}$, and the diffusion terms between porewaters and the water column cancel, so do not appear in the equations. Note the units of mass of $CS$ needs to be in g N, and some configurations may have multiple seagrass and macroalgae species.

Similarly for nitrogen, phosphorus and oxygen in the epibenthos:

$$TN = MA + SG_A + SG_B + CS\left(1 + R_N^*\right) + CH \tag{270}$$

$$\left.\frac{\partial TN}{\partial t}\right|_{epi} + h_{wc}\left.\frac{\partial TN}{\partial t}\right|_{wc} + h_{sed}\left.\frac{\partial TN}{\partial t}\right|_{sed} = 0 \tag{271}$$

$$TP = \frac{1}{30}\frac{31}{14}\left(MA + SG_A + SG_B\right) + \frac{1}{16}\frac{31}{14}\left(CS\left(1 + R_P^*\right) + CH\right) \tag{272}$$

$$\left.\frac{\partial TP}{\partial t}\right|_{epi} + h_{wc}\left.\frac{\partial TP}{\partial t}\right|_{wc} + h_{sed}\left.\frac{\partial TP}{\partial t}\right|_{sed} = 0 \tag{273}$$

$$BOD_\infty = \frac{550}{30}\frac{32}{14}\left(MA + SG_A + SG_B\right) + \frac{106}{16}\frac{32}{14}\left(CS\left(1 + R_C^*\right) + CH\right) \tag{274}$$

$$-\left.\frac{\partial BOD_\infty}{\partial t}\right|_{epi} + h_{wc}\left.\frac{\partial([O_2] - BOD_\infty)}{\partial t}\right|_{wc} + h_{sed}\left.\frac{\partial([O_2] - BOD_\infty)}{\partial t}\right|_{sed} = 0 \tag{275}$$

where there is no dissolved oxygen in the epibenthos.

### 7.5.4 Wetting and drying

When a water column becomes dry (the sea level drops below the seabed depth) ecological processes are turned off.



### 7.5.5 Unconditional stability

In addition to the above standard numerical techniques, a number of innovations are used to ensure model solutions are reached. Should an integration step fail in a grid cell, no increment of the state variables occurs, and the model continues with a warning flag registered (as `Ecology Error`). Generally the problem does not reoccur due to the transport of tracers alleviating the
stiff point in phase space of the model.

## 8   Model evaluation

The EMS BGC model has been deployed in a range of environments around Australia, and with each deployment skill assessment has been undertaken (for a history of these applications see Sec. 11). More recently, the EMS BGC has been thoroughly assessed against remotely-sensed and in situ observations on the Great Barrier Reef, as part of the eReefs project (Schiller
et al., 2014). The assessment of version B1p0 of the eReefs marine model configuration of the EMS included a 497 page report documenting a range of model configurations (4 km, 1 km and relocatable fine resolution versions) (Herzfeld et al., 2016). The optical and carbon chemistry outputs were assessed in Baird et al. (2016b) and Mongin et al. (2016b) respectively.

A more recent assessment of the biogeochemical model (vB2p0) compared simulations against a range of in situ observations that included 24 water quality moorings, 2 nutrient sampling programs (with a total of 18 stations) and time-series of taxon-
specific plankton abundance. In addition to providing a range of skill metrics, the assessment included analysis of seasonal plankton dynamics (Skerratt et al., 2019).

The techniques and observations used in Skerratt et al. (2019) have been compared to the version described in this paper (vB3p0) (see Supplementary Material). This includes observations of Chl a, dissolved inorganic carbon, nitrogen, phosphorus and ammonia, dissolved organic nitrogen and phosphorus, alkalinity, pH, aragonite saturation, mass of suspended sediments
and turbidity and Secchi depth.

In the following section we provide highlights of this assessment, with a focus on water chlorophyll dynamics.

### 8.1   Chlorophyll dynamics in a Great Barrier Reef (GBR) configuration

The most accurate measurements of water column chlorophyll concentrations in the GBR are obtained using high-performance liquid chromatography (HPLC) and chlorophyll extractions from water column samples. Inspection of time-series at Pelorus Island (Fig. 14) shows large variability in both the observations and the simulations, driven by inter-annual trends with 2011-2013 experiencing much greater river loads than 2014-2016, intra-annual trends driven by greater loads of nutrients during the wet season (Jan - May) than the remainder of the year, as well as monthly variability related to tidal movements and predator-
prey oscillations. Even given this variability, comparison of the instantaneous state of the extracted chlorophyll concentrations against vB3p0 was able to achieve an rms of 0.25 mg m$^{-3}$, and a bias of -0.14 mg m$^{-3}$.





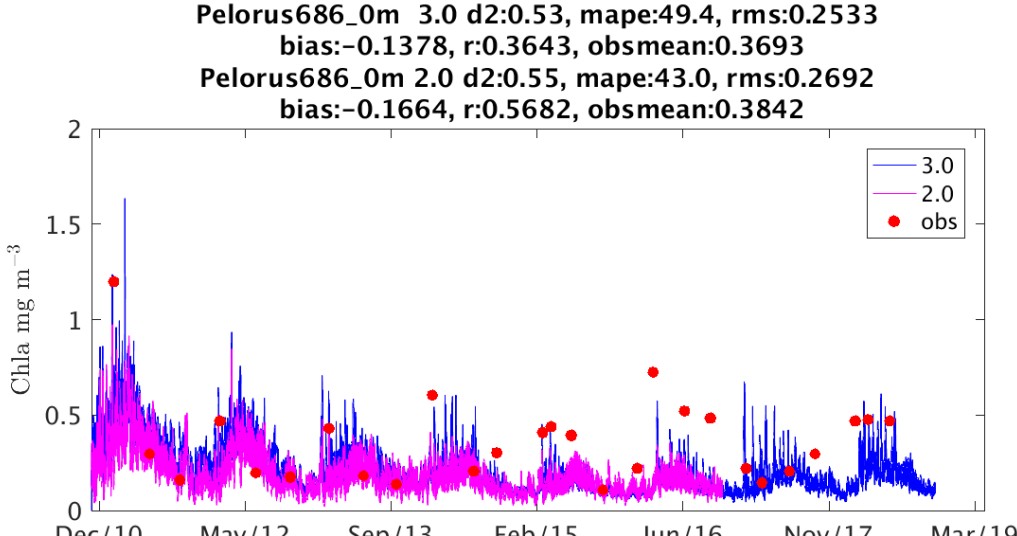

**Figure 14.** Observed surface chlorophyll concentration (red dots) at Pelorus Island Marine Monitoring Program site (146°29' E, 18°33' S) with a comparison to configurations vB2p0 (pink line) and vB3p0 (blue line). Statistics listed include the Willmott d2 metric (Willmott et al., 1985), mean absolute percent error (mape) and root mean square (rms) error.

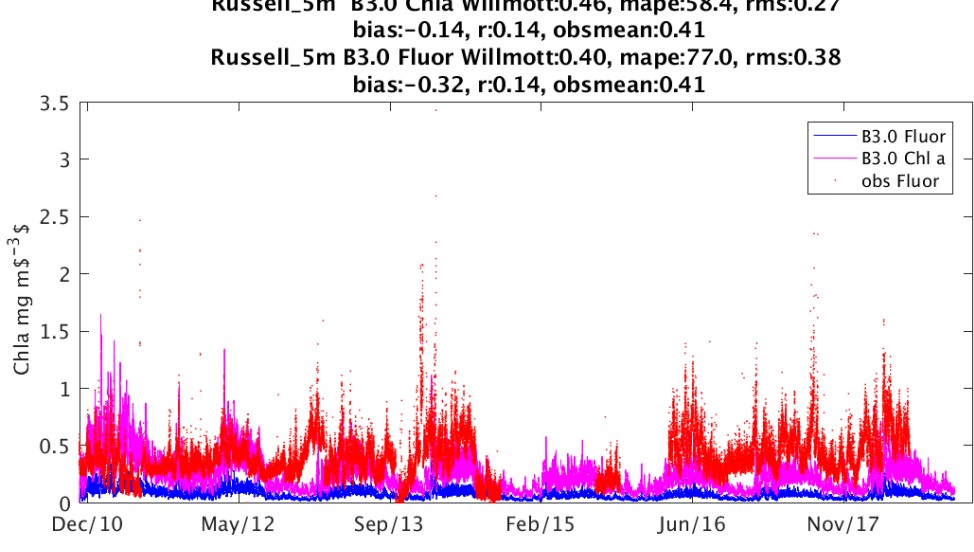

**Figure 15.** Observed chlorophyll fluorescence (red dots) at 5 m depth at Russell Island Marine Monitoring Program site (146°5' E, 17°14' S) with a comparison to configuration vB3p0 (pink line). The blue line is a trial product simulated fluorescence. For more information see Fig. 14.



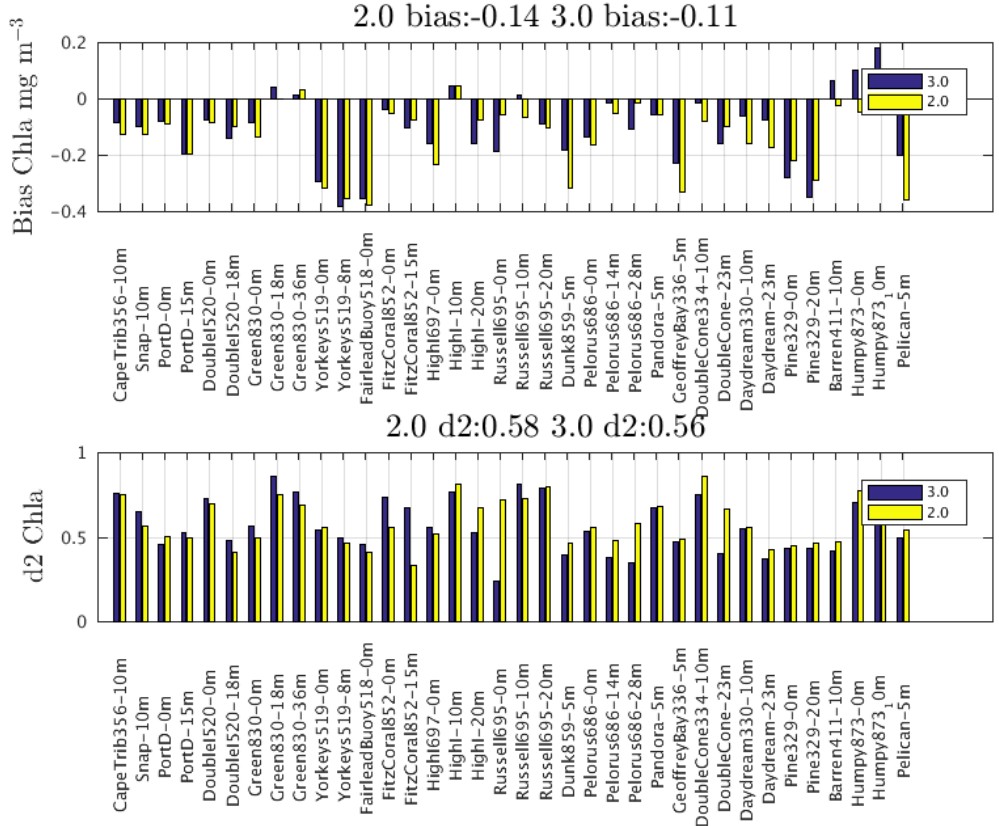

**Figure 16.** Skill metrics for the comparison of chlorophyll extracts at the Long Term Monitoring sites against observations for model version 3p0 and 2p0. For more information see Fig. 14 and for site locations p161-165 of the Supplementary Material.

Near the water sample sites, moored flourometers provide a greater temporal resolution of chlorophyll dynamics (Fig. 15). The observed time-series show high daily variability, which is also seen in the vB3p0 simulations. The florescent signal is generally considered to be less accurate than the chlorophyll extractions, with jumps seen between deployments. Nonetheless at this site the skill scores at Russell Island (Fig. 15) were similar to that of chlorophyll extractions at Pelorus Island.

5    The above model comparisons were undertaken at 14 sites along the GBR inshore waters. The summary of the bias and d2 metrics for extracted chlorophyll is given in Fig. 16. In general there was a small negative bias in simulated chlorophyll, which was reduced in v3p0. There were only small differences between the model formulation of v2p0 and v3p0, with the greatest difference being a reduced denitrification rate that slight increased the chlorophyll concentrations.

10   The outputs of all hindcasts in the eReefs project can be downloaded from:

`http://dapds00.nci.org.au/thredds/catalogs/fx3/catalog.html`





## 9 Code availability

The model web page is:

`https://research.csiro.au/cem/software/ems/`

The webpage links to an extensive User Guide for the entire EMS package, which contains any information that is generic

across the hydrodynamic, sediment, transport and ecological models, such as input/output formats. A smaller Biogeochemical User Guide documents details relevant only to the biogeochemical and optical models (such as how to specify wavelengths for the optical model), and a Biogeochemical Developer's Guide describes how to add additional processes to the code.

The entire Environmental Modelling Suite (EMS) C code is available on `GitHub`:

`https://github.com/csiro-coasts/EMS/`

The paper describes the BGC library within EMS. The version is labelled as vB3p0 to distinguish it from earlier versions of the ecological library used in the eReefs project and others. At the `GitHub` site, vB3p0 is referred to as ecology v1.1.1, is contained within EMS release v1.1 in the `GitHub` archive, and can be accessed at:

`https://github.com/csiro-coasts/EMS/releases/tag/v1.1.1`

### 9.1 Code architecture

This paper is a scientific description of the EMS ecological library (`/EMS/model/lib/ecology`). The ecological library consists primarily of a set of routines describing individual processes. The model chooses which processes it will include based on a configure file, an example of which appears below. The model equations are primarily derivatives of the ecological state variables, and have been split in this paper into separate processes (such as a phytoplankton growth), thus aligning with the code (such as `phytoplankton_spectral_grow_wc.c`). This object-based approach allows individual processes to be

included / excluded in a configuration file without re-writing the model code.

Within a process file, the routine containing the ecological derivatives is `<process_name>_calc`, and within that routine the ecological derivatives are stored within the array `y1`. Each element in the array `y1` stores the derivatives of a state variable. The index to the array for each state variable is determined within each process initialisation routine, `<process_name>_init`, and stored in the processes' workspace `ws`. In the case of nitrate, for example, the derivative held in `y1` will be the sum of the

derivatives calculated in multiple processes (such as each autotrophic growth process, nitrification, denitrification, and each grazing and mortality process). The array of derivatives is then used by the model's adaptive integrator to update the model state, as held in the array `y`.

Some components of the ecological model are updated only once every time step without the derivatives being calculated. These include the optical and carbon chemistry model state variables. In these cases, the state variables, which are stored in the

array `y`, are updated directly and this is done in either the routine `<process_name>_precalc` or `<process_name>_postcalc`.

The list of processes that this paper describes are for version B3p0, which is invoked with a configuration file listing the processes in each of the domains `water`, `sediment` and `epibenthic`:

`water`





```
    {
    tfactor
    viscosity
    moldiff
 values_common
    remineralization
    microphytobenthos_spectral_grow_wc
    phytoplankton_spectral_grow_wc(small)
    phytoplankton_spectral_grow_wc(large)
trichodesmium_mortality_wc
    trichodesmium_spectral_grow_wc
    phytoplankton_spectral_mortality_wc(small)
    phytoplankton_spectral_mortality_wc(large)
    zooplankton_mortality_wc(small)
zooplankton_mortality_wc(large)
    zooplankton_large_carnivore_spectral_grow_wc
    zooplankton_small_spectral_grow_wc
    nitrification_wc
    p_adsorption_wc
carbon_chemistry_wc
    gas_exchange_wc(carbon,oxygen)
    light_spectral_wc(H,HPLC)
    massbalance_wc
    }
epibenthos
    {
    tfactor_epi()
    values_common_epi()
    macroalgae_spectral_grow_epi()
seagrass_spectral_grow_epi(Zostera)
    seagrass_spectral_grow_epi(Halophila)
    seagrass_spectral_grow_epi(Deep)
    coral_spectral_grow_bleach_epi()
    coral_spectral_carb_epi(H)
macroalgae_mortality_epi()
    seagrass_spectral_mortality_proto_epi(Zostera)
    seagrass_spectral_mortality_proto_epi(Halophila)
    seagrass_spectral_mortality_proto_epi(Deep)
```





```
     massbalance_epi()
     light_spectral_uq_epi(H)
     diffusion_epi()
     }
 sediment
     {
     tfactor
     viscosity
     moldiff
values_common
     remineralization
     light_spectral_sed(HPLC)
     microphytobenthos_spectral_grow_sed
     carbon_chemistry_wc()
microphytobenthos_spectral_mortality_sed
     phytoplankton_spectral_mortality_sed(small)
     phytoplankton_spectral_mortality_sed(large)
     zooplankton_mortality_sed(small)
     zooplankton_mortality_sed(large)
trichodesmium_mortality_sed
     nitrification_denitrification_sed
     p_adsorption_sed
     massbalance_sed()
     }
```

or alternatively with a call in the configuration file: `PROCESSFNAME B3p0`.

    Other processes in the `process_library` can be validly called, but their scientific description is not given in this paper. The header in the source code for each process file gives detail about it use within the code, such as any arguments that it requires (for example `seagrass_spectral_grow_epi` requires the seagrass type as an argument).

## 10   Relocatable Coast and Ocean Model (RECOM)

A web based interface, RECOM, has been developed to automate the process of downscaling the EMS model using an existing hindcast as boundary conditions (`https://research.csiro.au/ereefs/models/models-about/recom/`, including the RECOM User Manual). For the purposes of learning how to apply the EMS software available, RECOM provides the user with the ability to generate a complete test case of a domain situated along the northeast Australian coastline. Once a RECOM simulation has been generated using the web interface, the entire simulation including source code, forcing and initial

condition files, model configuration files and the model output can be downloaded. This allows the user to repeat the model





simulation on their own computing system, and modify code, forcing, and output frequency as required. The technical details of RECOM are detailed in Baird et al. (2018), and in the RECOM User Manual.

## 11   Discussion

The EMS BGC model development has been a function of the historical applications of the model across a rage of ecosystems,
so it is worth giving a brief history of the model development.

### 11.1   History of the development of the EMS biogeochemical model

The EMS biogeochemical model was first developed as a nitrogen-based model for determining the assimilative capacity for sewerage discharged in to Port Philip Bay (Fig. 4), the embayment of the city of Melbourne (Harris et al., 1996). This study saw a focus on sediment processes such as denitrication, and demonstrated the ability of bay-wide denitrification to prevent
change in the ecological state of the bay exposed to sewerage treatment plant loads (Murray and Parslow, 1997; Murray and Parlsow, 1999). The basic structure of the model, and in particular the split of pelagic, epibenthic and sediment zones were in place for this project. This zonation generated the ability to resolve processes in shallow water systems, and in particular to consider benthic flora in detail.

The next major study involved simulating a range of estuarine morphologies (salt wedge, tidal, lagoon, residence times) and
forcings (river flow seasonality, nutrient inputs etc.) that were representative of Australia's 1000+ estuaries (Baird et al., 2003). At this point carbon and phosphorus were included in the model, and the process of including physical limits to ecological processes begun (e.g. diffusion limitation of nutrient uptake and encounter rate limitation of grazing).

Following studies in the phosphorus-limited Gippsland Lakes and macro-tidal Ord River system led to the refinement of the phosphorus absorption / desorption processes. Further studies of the biogeochemical - sediment interactions in the sub-tropical
Fitzroy River (Robson et al., 2006) and investigation of the impacts of a tropical cyclone (Condie et al., 2009), saw a stronger link to remote observations. At this time the use of offline transport schemes were also implemented (such as the Moreton Bay model), allowing for faster model integration by an order of magnitude (Gillibrand and Herzfeld, 2016).

The next major change in the BGC model involved implementing variable C:N:P ratios of microalgae through the introduction of reserves of energy, nitrogen and phosphorus (Wild-Allen et al., 2010), allowing for more accurate prediction of the
elemental budgets and impacts of natural and anthropogenic forcing of the Derwent River estuary, southeast Tasmania. This study was followed up by a number of studies developing scenarios to inform management strategies of the region (Wild-Allen et al., 2011, 2013; Skerratt et al., 2013; Hadley et al., 2015a, b).

From 2010 onwards, EMS has been applied to consider the impacts of catchment loads on the Great Barrier Reef. The focus on water clarity led to the development of a spectrally-resolved optical model, and the introduction of simulated true colour
(Fig. 3). The eReefs project was the first EMS application to consider corals, resulting in the introduction of the host-symbiont coral system and equilibrium carbon chemistry (Mongin and Baird, 2014; Mongin et al., 2016b, a). Additionally, the calculation





of model outputs that match remote-sensing observations allowed the model to be run in a data assimilating system, where the observation-model mis-match was based on remote-sensing reflectance (Jones et al., 2016).

The most recent application of the EMS BGC model has been for investigating the environmental impact of aquaculture in Los Lagos, Chile. For the Los Lagos application, new processes for fish farms, dinoflagellates and benthic filter feeders

were added, although these additions aren't described in this document. As a demonstration of the ability to add and remove processes, the Los Lagos application was run with the same EMS C executable file as the Great Barrier Reef application - just with the configuration files altered.

## 11.2 Future developments

The EMS has been developed to address specific scientific questions in Australia's coastal environment. As a result, the set of

10 processes the EMS considers varies from those typically applied by other groups developing marine BGC models. Processes which have not been considered, but often are considered in marine BGC models, include iron and silicate limitation (which are not common on the Australian continental shelf or estuaries), photoinhibition of microalgae, explicit bacterial biomass. Each of these will be considered as the need arises.

A deliberate decision in the development of the EMS BGC model was made to avoid higher trophic level processes, such fish

dynamics and reproduction of long-lived species. This decision was made because: (1) including these longer time-scale, often highly non-linear, processes reduces the ability of development to concentrate on BGC processes; and (2) it was recognised that CSIRO has developed a widely-used ecosystem model (Atlantis, `https://research.csiro.au/atlantis/`, Fulton et al. (2014)), and that coupling the EMS with Atlantis takes advantage of complimentary strengths of the two modelling systems.

A recent capacity introduced to EMS is the development of a relocatable capability (RECOM, Sec. 10), allowing model configurations (grid, river and meteorological forcing, ecological processes, boundary conditions) to be automatically generated. This capability will be a good test of the portability of the BGC model, and in particular the use of geometric description of physical limits to ecological processes.

Future enhancements in the EMS BGC model for tropical systems are likely to continue to pursue those components at

25 risk from human impacts, such as dissolution of marine carbonates affecting sediment substrate and herbicide interactions with photosystems. We also expect to continue to refine the optical model, and in particular the relationship between particle size distribution and mass-specific scattering and absorption properties. In temperate systems, current and near-future deployments of the EMS code in Australia will be focussed on coastal system characterisation for aquaculture, carbon sequestration and management decision support for the Blue Economy. Ongoing research includes improved methods for model validation

against observations and translation of model outputs into knowledge that informs stakeholder decisions.

## 11.3 Concluding thoughts

The BGC model in the CSIRO EMS has developed unique parameterisation when compared to other marine biogeochemical models applied elsewhere due in part to a unique set of scientific challenges of the Australian coastline. It has proved to be



useful in many applications, most notably the Great Barrier Reef where extensive observational datasets has allowed new process model development and detailed model skill assessment [(Baird et al., 2016b, a; Mongin et al., 2016b; Skerratt et al., 2019) and eReefs.info]. This document provides easy access to some of the novel process formulations that have been important in this success, as well as a complete description of the entire modelling system, which can be downloaded from

5   GitHub.



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

*Competing interests.* The authors declare no competing interests.

### Acknowledgements

Many scientists and projects have contributed resources and knowhow to the development of this model over 20+ years. For this dedication we are very grateful.

Those who have contributed to the numerical code include (CSIRO unless stated):

Mike Herzfeld, Philip Gillibrand, John Andrewartha, Farhan Rizwi, Jenny Skerratt, Mathieu Mongin, Mark Baird, Karen Wild-Allen, John Parlsow, Emlyn Jones, Nugzar Margvelashvili, Pavel Sakov (BOM, who introduced the process structure), Jason Waring, Stephen Walker, Uwe Rosebrock, Brett Wallace, Ian Webster, Barbara Robson, Scott Hadley (University of Tasmania), Malin Gustafsson (University of Technology Sydney, UTS), Matthew Adams (University of Queensland, UQ).

We also thank Britta Schaffelke for her commitment to observations that are used in model evaluation detailed in the Sup-
plementary Material, and Cedric Robillot for his leadership of the eReefs Project.

Collaborating scientists include:

Bronte Tilbrook, Andy Steven, Thomas Schroeder, Nagur Cherukuru, Peter Ralph (UTS), Russ Babcock, Kadija Oubelkheir, Bojana Manojlovic (UTS), Stephen Woodcock (UTS), Stuart Phinn (UQ), Chris Roelfsema (UQ), Miles Furnas (AIMS), David McKinnon (AIMS), David Blondeau-Patissier (Charles Darwin University), Michelle Devlin (James Cook University),
Eduardo da Silva (JCU), Julie Duchalais, Jerome Brebion, Leonie Geoffroy, Yair Suari, Cloe Viavant, Lesley Clementson (pigment absorption coefficients), Dariusz Stramski (inorganic absorption and scattering coefficients), Erin Kenna, Line Bay (AIMS), Neal Cantin (AIMS), Luke Morris (AIMS), Daniel Harrison (USYD), Karlie MacDonald.

Funding bodies: CSIRO Wealth from Oceans Flagship, Gas Industry Social & Environmental Research Alliance (GISERA), CSIRO Coastal Carbon Cluster, Derwent Estuary Program, INFORM2, eReefs, Great Barrier Reef Foundation, Australian





Climate Change Science Program, University of Technology Sydney, Department of Energy and Environment, Integrated Marine Observing System (IMOS), National Environment Science Program (NESP TWQ Hub).





## Appendix A: State (prognostic) variables

The below tables list the ecologically-relevant physical variables (Table A1), 10 dissolved (Table A2), 2 zooplankton (Table A3), 20 microalgal (Table A4), 7 non-living inorganic particulate (Table A5), 7 non-living organic particulate (Table A6) and 7 epibenthic plant (Table A7) and 5 coral polyp (Table A8) and 4 reaction centre (Table A9) state variables. All state variables that exist in the water column layers have an equivalent in the sediment layers (and are specified by `<variable name>_sed`). The dissolved tracers are given as a concentration in the porewater, while the particulate tracers are given as a concentration per unit volume (see Sec. 7.5.2).

| Name | Symbol | Units | Description |
|---|---|---|---|
| Temperature (temp) | $T$ | °C | Water temperature |
| Salinity (salt) | $S$ | PSU | Water salinity |
| Sea level elevation (eta) | $\eta$ | m | Sea level elevation relative to mean sea level |
| Porosity (porosity) | $\phi$ | - | Fraction of the volume made up of water |
| Bottom shear stress (us-trcw_skin) | $\tau$ | N m$^{-2}$ | Shear stress due to currents and waves |
| Sand ripple height (PHYS-RIPH) | - | m | Physical dimension used for estimating bottom roughness due to ripples according to Grant and Madsen (1982) |
| Solar zenith (Zenith) | $\theta_{air}$ | radians | Solar zenith angle is the angle between the zenith and the centre of the Sun's disc, taking a value of zero with the sun directly above, and $\pi/2$ when at, or below, the horizon. |

**Table A1.** Long name (and variable name) in model output files, symbol and units used in this document, and a description of ecologically-relevant physical state and diagnostic variables.





| Name | Symbol | Units | Description |
|---|---|---|---|
| Total alkalinity (alk) | $A_T$ | $\mathrm{mol\ kg^{-1}}$ | Concentration of ions that can be converted to uncharged species by a strong acid. The model assumes $A_T = [\mathrm{HCO_3^-}] + [\mathrm{CO_3^{2-}}]$, often referred to as carbonate alkalinity. Alkalinity and DIC together quantify the equilibrium state of the seawater carbon chemistry. |
| Nitrate (NO3) | $[\mathrm{NO_3^-}]$ | $\mathrm{mg\ N\ m^{-3}}$ | Concentration of nitrate. In the absence of nitrite $[\mathrm{NO_2^-}]$ in the model, nitrate represents $[\mathrm{NO_3^-}] + [\mathrm{NO_2^-}]$. |
| Ammonia (NH4) | $[\mathrm{NH_4^-}]$ | $\mathrm{mg\ N\ m^{-3}}$ | Concentration of ammonia. |
| Dissolved Inorganic Phosphorus (DIP) | $P$ | $\mathrm{mg\ P\ m^{-3}}$ | Concentration of dissolved inorganic phosphorus, also referred to as orthophosphate or soluble reactive phosphorus, SRP, composed chiefly of $\mathrm{HPO_4^{2-}}$ ions, with a small percentage present as $\mathrm{PO_4^{3-}}$. |
| Dissolved inorganic carbon (DIC) | $DIC$ | $\mathrm{mg\ C\ m^{-3}}$ | Concentration of dissolved inorganic carbon, composed chiefly at seawater pH of $\mathrm{HCO_3^-}$, with a small percentage present as $\mathrm{CO_3^{2-}}$. |
| Dissolved oxygen (Oxygen) | $[\mathrm{O_2}]$ | $\mathrm{mg\ O\ m^{-3}}$ | Concentration of oxygen. |
| Chemical Oxygen Demand (COD) | $COD$ | $\mathrm{mg\ O\ m^{-3}}$ | Concentration of products of anoxic respiration in oxygen units. This represents products such as hydrogen sulfide, $\mathrm{H_2S}$, that are produced during anoxic respiration and which, upon reoxidation of the water, will consume oxygen. |
| Dissolved Organic Carbon (DOR_C) | $O_C$ | $\mathrm{mg\ C\ m^{-3}}$ | The concentration of carbon in dissolved organic compounds. |
| Dissolved Organic Nitrogen (DOR_N) | $O_N$ | $\mathrm{mg\ N\ m^{-3}}$ | The concentration of nitrogen in dissolved organic compounds. |
| Dissolved Organic Phosphorus (DOR_P) | $O_P$ | $\mathrm{mg\ P\ m^{-3}}$ | The concentration of phosphorus in dissolved organic compounds. |

**Table A2.** Long name (and variable name) in model output files, symbol and units used in this document, and a description of all dissolved state variables. When the concentration of an ion is given, the chemical formulae appears in [ ] brackets.

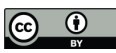



| Name | Symbol & Units | Description |
|---|---|---|
| Phytoplankton N (*_N) | $B$ [mg N m$^{-3}$] | Total structural biomass of nitrogen of the phytoplankton population. All microalgae have a C:N:P ratio of the structural material of 106:16:1. Thus the mass of phosphorus in the structural material of a population with a biomass $B$ is given by: $\frac{1}{16}\frac{31}{14}B$ and the mass of carbon by: $\frac{106}{16}\frac{12}{14}B$. The number of cells is given by $B/m_N$. |
| Phytoplankton N reserves (*_NR) | $BR_N^*$ [mg N m$^{-3}$] | Total non-structural biomass of nitrogen of the phytoplankton population. Phytoplankton N reserves divided by Phytoplankton N is a number between 0 and 1 and represents the factor by which phytoplankton growth is inhibited due to the internal reserves of nitrogen. |
| Phytoplankton P reserves (*_PR) | $\frac{1}{16}\frac{31}{14}BR_P^*$ [mg P m$^{-3}$] | Total non-structural biomass of phosphorus of the phytoplankton population. Phytoplankton P reserves divided by (Phytoplankton N $\times\frac{1}{16}\frac{31}{14}$) is a number between 0 and 1 and represent the factor by which phytoplankton growth is inhibited due to the internal reserves of phosphorus. |
| Phytoplankton I reserves (*_I) | $1060\frac{1}{16}\frac{1}{14}BR_I^*$ [mmol photon m$^{-3}$] | Total non-structural biomass of fixed carbon of the phytoplankton population, quantified in photons. Phytoplankton I reserves divided by (Phytoplankton N $\times\frac{1060}{16}\frac{1}{14}$) is a number between 0 and 1 and represent the factor by which phytoplankton growth is inhibited due to the internal reserves of energy (or fixed carbon). |
| Phytoplankton chlorophyll (*_Chl) | $nc_iV$ [mg m$^{-3}$] | Concentration of the chlorophyll $a$ pigment of the population. The four phytoplankton classes have two pigments, a chlorophyll $a$-based pigment and an accessory pigment. As the pigment concentration adjusts to optimise photosynthesis, including the presence of the accessory pigment, the intracellular content, $c_iV$, represents only the chlorophyll $a$-based pigment. As the model does not distinguish between mono-vinyl and di-vinyl forms of chlorophyll, this $c_i$ represents either form, depending on the phytoplankton type. |

**Table A3.** Long name (and variable name) in model output files, symbol and units used in this document, and description of all microalgae state variables in the model. The model contains four categories of phytoplankton (category shown as * in left column, and given in brackets in following list): small (PhyS), $r < 2\ \mu m$ phytoplankton; large (PhyL): $r > 2\mu m$ phytoplankton; $Trichodesmium$ (Tricho): nitrogen fixing phytoplankton; and benthic microalgae (MPB): fast-sinking diatoms that are suspended primarily in the top layer of sediment porewaters. The elemental ratio of phytoplankton including both structural material and reserves is given by: C:N:P $= 106(1+R_C^*):16(1+R_N^*):(1+R_P^*)$. In the model description (this document) we have more correctly described fixed carbon as carbon reserves, while in the model outputs they are represented quantified as energy reserves. The relationship is 1 mg C of carbon reserves is equal to $(1060/106)/12.01$ mmol photons of energy reserves.





| Name | Symbol | Units | Description |
|---|---|---|---|
| Zooplankton N (ZooS_N, ZooL_N) | $Z$ | [mg N m$^{-3}$] | Total biomass of nitrogen in animals. With only small and large zooplankton categories resolved, small zooplankton represents the biomass of unicellular fast growing animals (protozoans) and large zooplankton represents the biomass of all other animals (metazoans). All zooplankton have a C:N:P ratio of 106:16:1. Thus the mass of phosphorus of a population with a biomass $Z$ is given by: $\frac{1}{16}\frac{31}{14}Z$ and the mass of carbon by: $\frac{106}{16}\frac{12}{14}Z$. |

**Table A4.** Long name (and variable name) in model output files, symbol, unit used in this document, and description of zooplankton state variables in the model.





| Name | Symbol | Units | Description |
|---|---|---|---|
| Fine Sediment (FineSed) | $FineSed$ | [kg m$^{-3}$] | Identical to Mud-mineral, except that it is initialised to zero in the model domain, and enters only from the catchments. |
| Dust (Dust) | $Dust$ | [kg m$^{-3}$] | Very small sized, re-suspending particles with a sinking velocity of 1 m d$^{-1}$ and mass-specific optical properties based on observations in Gladstone Harbour. |
| Mud-mineral (Mud-mineral) | $Mud_{non-CaCO_3}$ | [kg m$^{-3}$] | Small sized, re-suspending particles with a sinking velocity of 17 m d$^{-1}$, and mass-specific optical properties based on observations in Gladstone Harbour. |
| Mud-carbonate (Mud-carbonate) | $Mud_{CaCO_3}$ | [kg m$^{-3}$] | Small sized, re-suspending particles with a sinking velocity of 17 m d$^{-1}$, and mass-specific optical properties based on observations of suspended carbonates at Lucinda Jetty. |
| Sand-mineral (Sand-mineral) | $Sand_{non-CaCO_3}$ | [kg m$^{-3}$] | Medium sized, re-suspending particles with a sinking velocity of 173 m d$^{-1}$ and mass-specific optical properties based on observations in Gladstone Harbour. |
| Sand-carbonate (Sand-carbonate) | $Sand_{CaCO_3}$ | [kg m$^{-3}$] | Medium sized, re-suspending particles with a sinking velocity of 173 m d$^{-1}$ and mass-specific optical properties based on observations of suspended carbonates at Lucinda Jetty. |
| Gravel-mineral (Gravel-mineral) | $Gravel_{non-CaCO_3}$ | [kg m$^{-3}$] | Large, non-resuspending particles. |
| Gravel-carbonate (Gravel-carbonate) | $Gravel_{CaCO_3}$ | [kg m$^{-3}$] | Large, non-resuspending particles. |

**Table A5.** Long name (and variable name) in model output files, symbol, unit used in this document, and description of all inorganic particulate state variables in the model.





| Name | Symbol | Units | | Description |
|------|--------|-------|---|-------------|
| Particulate Inorganic Phosphorus (PIP) | $PIP$ | [mg m$^{-3}$] | P | Phosphorus ions that are absorbed onto particles. It is considered a particulate with the same properties as Mud. |
| Immobilised Particulate Inorganic Phosphorus (PIPI) | $PIPI$ | [mg m$^{-3}$] | P | Phosphorus that is permanently removed from the system through burial of PIP. |
| Labile Detritus Nitrogen Plank (DetPL_N) | $D_{Red}$ | [mg m$^{-3}$] | N | Concentration of N in labile (quickly broken down) organic matter with a C:N:P ratio of 106:16:1 derived from living microalgae, zooplankton, coral host tissue and zooxan-thellae with the same C:N:P ratio. Thus the mass of phosphorus in $D_{Red}$ is given by: $\frac{1}{16}\frac{31}{14}D_{Red}$ and the mass of carbon by: $\frac{106}{16}\frac{12}{14}D_{Red}$. |
| Labile Detritus Nitrogen Benthic (DetBL_N) | $D_{Atk}$ | [mg m$^{-3}$] | N | Concentration of N in labile (quickly broken down) organic matter with a C:N:P ratio of 550:30:1 derived from living seagrass and macroalgae with the same C:N:P ratio. Thus the mass of phosphorus in $D_{Atk}$ is given by: $\frac{1}{30}\frac{31}{14}D_{Atk}$ and the mass of carbon by: $\frac{550}{30}\frac{12}{14}D_{Atk}$. |
| Refractory Detritus Carbon (DetR_C) | $D_C$ | [mg m$^{-3}$] | C | Concentration of carbon as particulate refractory (slowly breaking down) material. It is sourced only from the breakdown of labile detritus, and from rivers. |
| Refractory Detritus Nitrogen (DetR_N) | $D_N$ | [mg m$^{-3}$] | N | Concentration of nitrogen as particulate refractory (slowly breaking down) material. It is sourced only from the breakdown of labile detritus, and from rivers. |
| Refractory Detritus Phosphorus (DetR_P) | $D_P$ | [mg m$^{-3}$] | P | Concentration of phosphorus as particulate refractory (slowly breaking down) material. It is sourced only from the breakdown of labile detritus, and from rivers. |

**Table A6.** Long name (and variable name) in model output files, symbol, unit used in this document, and description of all particulate detrital state variables in the model.





| Name | Symbol | Description |
|---|---|---|
| Macroalgae (MA) | $MA$ [g N m$^{-2}$] | Concentration of nitrogen biomass per m$^2$ of macrolagae. Macroalgae (or seaweed) grows above all other benthic plants (corals, seagrasses, benthic microalgae). It is parameterised as a non-calcifying leafy algae, with a C:N:P ratio of 550:30:1, and a formulation for calculating the percentage of the bottom covered as $1 - \exp(-\Omega_{MA}\, MA)$. In the model, in the absence of both calcifying macroalgae (particularly *Halimeda*) and unicellular epiphytes, macroalgae represents the biomass of all seaweeds and epiphytes. |
| Seagrass (SG) | $SG$ [g N m$^{-2}$] | Concentration of nitrogen biomass per m$^2$ of a seagrass form parameterised to be similar to *Zostera*. This form captures light after it has passed through macroalgae and before it passes through *Halophila*. This form is better adapted to high light, low nutrient conditions than *Halophila* as a result of a deeper root structure and being able to shade it. See macroalgae for elemental ratio and bottom cover. |
| Halophila (SGH) | $SGH$ [g N m$^{-2}$] | Concentration of nitrogen biomass per m$^2$ of a seagrass form parameterised to be similar to *Halophila ovalis*. This form captures light after it has passed through the *Zostera* seagrass form. The *Halophila ovalis* form is batter adapted to low light conditions than *Zostera*, having a faster growth rate and lower minimum light requirement. See macroalgae for elemental ratio and bottom cover. |
| Deep seagrass (SGD) | $SGD$ [g N m$^{-2}$] | Concentration of nitrogen biomass per m$^2$ of a seagrass form parameterised to be similar to *Halophila deciphens*. This form captures light after it has passed through the *Zostera* and *Halophila ovalis* seagrass form. |
| *root N | $*ROOT\_N$ [g N m$^{-2}$] | Concentration of nitrogen biomass per m$^2$ in the roots of seagrass type * (SG, SGH or SGD). While this biomass in reality exists in multiple depths in the sediments, and in the model accesses multiple layers for nutrient uptake, it is quantified in the epibenthic compartment. |

**Table A7.** Long name (and variable name) in model output files, units, symbol used in this document, and description of macroalgae and seagrass state variables in the model. The order in the above table corresponds to their vertical position, and therefore the order in which they access light. Benthic microalgae, being suspended in porewaters, is consider as a particulate in Table A5.





| Name | Symbol | Description |
|---|---|---|
| Coral host N (CH_N) | $CH$ [g N m$^{-2}$] | Concentration of nitrogen biomass per m$^2$ of coral host tissue in the entire grid cell. Unlike other epibenthic variables, corals area is assumed to exist in communities that are potentially smaller than the grid size. The fraction of the grid cell covered by corals is given by $A_{CH}$. Thus the biomass in the occupied region is given by $CH/A_{CH}$. The percent coverage of the coral of the bottom for the whole cell is given by $A_{CH}$ $(1 - \exp(-\Omega_{CH}\ CH/A_{CH}))$. With only one type of coral resolved, $CH$ represents the biomass of all symbiotic corals. Since the model contains no other benthic filter-feeders, $CH$ best represent the sum of the biomass of all symbiotic filter-feeding organisms such as corals, sponges, clams etc. C:N:P is 106:16:1. |
| Coral symbiont N (CS_N) | $CS$ [mg N m$^{-2}$] | Concentration of nitrogen biomass per m$^2$ of coral symbiont cells, or zooxanthellae. To determine the density of cells, use $n = CS/m_N$. The percentage of the bottom covered is given by $\frac{\pi}{2\sqrt{3}} n \pi r_{zoo}^2$ where $\pi r^2$ is the projected area of the cell, $n$ is the number of cells, and $\pi/(2\sqrt{3}) \sim 0.9069$ accounts for the maximum packaging of spheres. C:N:P is 106:16:1. |
| Coral symbiont chl (CS_Chl) | $n c_i V$ [mg chl m$^{-2}$] | Concentration of chlorophyll biomass per m$^2$ of coral symbiont cells. |
| Coral symbiont diadi-noxanthin (CS_Xp) | $n x_p V$ [mg pig m$^{-2}$] | Concentration of the photosynthetic xanthophyll cycle pigment per m$^2$ of coral symbiont cells. |
| Coral symbiont diatox-anthin (CS_Xh) | $n x_h V$ [mg pig m$^{-2}$] | Concentration of heat dissipating xanthophyll cycle pigment biomass per m$^2$ of coral symbiont cells. |

**Table A8.** Long name (and variable name) in model output files, units, symbol used in this document, and description of coral state variables in the model.





| Name | Symbol | Description |
|---|---|---|
| Symbiont oxidised RC (CS_Qox) | $Q_{ox}$ [mg N m$^{-2}$] | Concentration of symbiont reaction centres in an oxidised state per m$^2$, residing in the symbiont, of in the entire grid cell. |
| Symbiont reduced RC (CS_Qred) | $Q_{red}$ [mg N m$^{-2}$] | Concentration of symbiont reaction centres in a reduced state per m$^2$, residing in the symbiont, of in the entire grid cell. |
| Symbiont inhibited RC (CS_Qin) | $Q_{in}$ [mg N m$^{-2}$] | Concentration of symbiont reaction centres in an inhibited state per m$^2$, residing in the symbiont, of in the entire grid cell. |
| Symbiont reactive oxygen (CS_RO) | $[ROS]$ [mg N m$^{-2}$] | Concentration of reactive oxygen per m$^2$, residing in the symbiont, of in the entire grid cell. |

**Table A9.** Long name (and variable name) in model output files, units, symbol used in this document, and description of coral reaction state variables in the model.





## Appendix B: Parameter values used in eReefs biogeochemical model (B3p0).

The below five tables of parameters are specified for each run, and can be automatically generated by the EMS software after a simulation from the parameter file. At model initialisation the model produces a file ecology_setup.txt that contains a list of all the parameter values used, both those specified in the parameter file, and those using model defaults.

5    For a more information see Robson et al. (2018).

| Description | Name in code | Symbol | Value | Units |
|---|---|---|---|---|
| Reference temperature | Tref | $T_{ref}$ | 2.000000e+01 | Deg C |
| Temperature coefficient for rate parameters | Q10 | $Q10$ | 2.000000e+00 | none |
| Nominal rate of TKE dissipation in water column | TKEeps | $\epsilon$ | 1.000000e-06 | $m^2\ s^{-3}$ |
| Atmospheric CO2 | xco2_in_air | $pCO_2$ | 3.964800e+02 | ppmv |
| Concentration of dissolved N2 | N2 | $[N_2]_{gas}$ | 2.000000e+03 | $mg\ N\ m^{-3}$ |
| DOC-specific absorption of CDOM 443 nm | acdom443star | $k_{CDOM,443}$ | 1.300000e-04 | $m^2\ mg\ C^{-1}$ |

**Table B1.** Environmental parameters in eReefs biogeochemical model (B3p0).





| Description | Name in code | Symbol | Value | Units |
|---|---|---|---|---|
| Chl-specific scattering coef. for microalgae | bphy | $b_{phy}$ | 2.000000e-01 | $m^{-1}(mg\ Chla\ m^{-3})^{-1}$ |
| Nominal N:Chl a ratio in phytoplankton by weight | NtoCHL | $R_{N:Chl}$ | 7.000000e+00 | $g\ N(g\ Chla)^{-1}$ |
| Maximum growth rate of PL at Tref | PLumax | $\mu_{PL}^{max}$ | 1.400000e+00 | $d^{-1}$ |
| Radius of the large phytoplankton cells | PLrad | $r_{PL}$ | 4.000000e-06 | m |
| Natural (linear) mortality rate, large phytoplankton | PhyL_mL | $m_{L,PL}$ | 1.000000e-01 | $d^{-1}$ |
| Natural (linear) mortality rate in sed., large phyto. | PhyL_mL_sed | $m_{L,PL,sed}$ | 1.000000e+01 | $d^{-1}$ |
| Maximum growth rate of PS at Tref | PSumax | $\mu_{PL}^{max}$ | 1.600000e+00 | $d^{-1}$ |
| Radius of the small phytoplankton cells | PSrad | $r_{PS}$ | 1.000000e-06 | m |
| Natural (linear) mortality rate, small phyto. | PhyS_mL | $m_{L,PS}$ | 1.000000e-01 | $d^{-1}$ |
| Natural (linear) mortality rate in sed., small phyto. | PhyS_mL_sed | $m_{L,PS,sed}$ | 1.000000e+00 | $d^{-1}$ |
| Maximum growth rate of MB at Tref | MBumax | $\mu_{MPB}^{max}$ | 8.390000e-01 | $d^{-1}$ |
| Radius of the MPB cells | MBrad | $r_{MPB}$ | 1.000000e-05 | m |
| Natural (quadratic) mortality rate, MPB (in sed) | MPB_mQ | $m_{Q,MPB}$ | 1.000000e-04 | $d^{-1}(mg\ N\ m^{-3})^{-1}$ |
| Ratio of xanthophyll to chl a of PS | PSxan2chl | $\Theta_{xan2chl,PS}$ | 5.100000e-01 | $mg\ mg^{-1}$ |
| Ratio of xanthophyll to chl a of PL | PLxan2chl | $\Theta_{xan2chl,PL}$ | 8.100000e-01 | $mg\ mg^{-1}$ |
| Ratio of xanthophyll to chl a of MPB | MBxan2chl | $\Theta_{xan2chl,MPB}$ | 8.100000e-01 | $mg\ mg^{-1}$ |
| Maximum growth rate of Trichodesmium at Tref | Tricho_umax | $\mu_{MPB}^{max}$ | 2.000000e-01 | $d^{-1}$ |
| Radius of Trichodesmium colonies | Tricho_rad | $r_{MPB}$ | 5.000000e-06 | m |
| Sherwood number for the Tricho dimensionless | Tricho_Sh | $Sh_{Tricho}$ | 1.000000e+00 | none |
| Linear mortality for Tricho in sediment | Tricho_mL | $m_{L,Tricho}$ | 1.000000e-01 | $d^{-1}$ |
| Quadratic mortality for Tricho due to phages in wc | Tricho_mQ | $m_{Q,Tricho}$ | 1.000000e-01 | $d^{-1}(mg\ N\ m^{-3})^{-1}$ |
| Critical Tricho above which quadratic mortality applies | Tricho_crit | $Tricho_{crit}$ | 2.000000e-04 | $mg\ N\ m^{-3}$ |
| Minimum density of Trichodesmium | p_min | $\rho_{min,Tricho}$ | 9.000000e+02 | $kg\ m^{-3}$ |
| Maximum density of Trichodesmium | p_max | $\rho_{max,Tricho}$ | 1.050000e+03 | $kg\ m^{-3}$ |
| DIN conc below which Tricho N fixes | DINcrit | $DIN_{crit}$ | 1.000000e+01 | $mg\ N\ m^{-3}$ |
| Ratio of xanthophyll to chl a of Trichodesmium | Trichoxan2chl | $\Theta_{xan2chl,Tricho}$ | 5.000000e-01 | $mg\ mg^{-1}$ |
| Minimum carbon to chlorophyll a ratio | C2Chlmin | $\theta_{min}$ | 2.000000e+01 | wt/wt |

**Table B2.** Phytoplankton parameters in eReefs biogeochemical model (B3p0).


| Description | Name in code | Symbol | Value | Units |
|---|---|---|---|---|
| Maximum growth rate of ZS at Tref | ZSumax | $\mu^{ZS}_{max}$ | 4.000000e+00 | $d^{-1}$ |
| Radius of the small zooplankton cells | ZSrad | $r_{ZS}$ | 5.000000e-06 | m |
| Swimming velocity for small zooplankton | ZSswim | $U_{ZS}$ | 2.000000e-04 | $m\ s^{-1}$ |
| Grazing technique of small zooplankton | ZSmeth | | rect | none |
| Maximum growth rate of ZL at Tref | ZLumax | $\mu^{ZL}_{max}$ | 1.330000e+00 | $d^{-1}$ |
| Radius of the large zooplankton cells | ZLrad | $r_{ZL}$ | 3.200000e-04 | m |
| Swimming velocity for large zooplankton | ZLswim | $U_{ZL}$ | 3.000000e-03 | $m\ s^{-1}$ |
| Grazing technique of large zooplankton | ZLmeth | | rect | none |
| Growth efficiency, large zooplankton | ZL_E | $E_{ZL}$ | 4.260000e-01 | none |
| Growth efficiency, small zooplankton | ZS_E | $E_{ZS}$ | 4.620000e-01 | none |
| Natural (quadratic) mortality rate, large zooplankton | ZL_mQ | $m_{Q,ZL}$ | 1.200000e-02 | $d^{-1}(mg\ N\ m^{-3})^{-1}$ |
| Natural (quadratic) mortality rate, small zooplankton | ZS_mQ | $m_{Q,ZS}$ | 2.000000e-02 | $d^{-1}(mg\ N\ m^{-3})^{-1}$ |
| Fraction of growth inefficiency lost to detritus, large zoo. | ZL_FDG | $\gamma_{ZL}$ | 5.000000e-01 | none |
| Fraction of mortality lost to detritus, large zoo. | ZL_FDM | $N/A$ | 1.000000e+00 | none |
| Fraction of growth inefficiency lost to detritus, small zoo. | ZS_FDG | $\gamma_{ZS}$ | 5.000000e-01 | none |
| Fraction of mortality lost to detritus, small zooplankton | ZS_FDM | $N/A$ | 1.000000e+00 | none |

**Table B3.** Zooplankton parameters in eReefs biogeochemical model (B3p0).





| Description | Name in code | Symbol | Value | Units |
|---|---|---|---|---|
| Fraction of labile detritus converted to refractory detritus | F_LD_RD | $\zeta_{Red}$ | 1.900000e-01 | none |
| Fraction of labile detritus converted to DOM | F_LD_DOM | $\vartheta_{Red}$ | 1.000000e-01 | none |
| fraction of refractory detritus that breaks down to DOM | F_RD_DOM | $\vartheta_{Ref}$ | 5.000000e-02 | none |
| Breakdown rate of labile detritus at 106:16:1 | r_DetPL | $r_{Red}$ | 4.000000e-02 | $d^{-1}$ |
| Breakdown rate of labile detritus at 550:30:1 | r_DetBL | $r_{Atk}$ | 1.000000e-03 | $d^{-1}$ |
| Breakdown rate of refractory detritus | r_RD | $r_R$ | 1.000000e-03 | $d^{-1}$ |
| Breakdown rate of dissolved organic matter | r_DOM | $r_O$ | 1.000000e-04 | $d^{-1}$ |
| Respiration as a fraction of umax | Plank_resp | $\phi$ | 2.500000e-02 | none |
| Oxygen half-saturation for aerobic respiration | KO_aer | $K_{OA}$ | 2.560000e+02 | mg O m$^{-3}$ |
| Maximum nitrification rate in water column | r_nit_wc | $\tau_{nit,wc}$ | 1.000000e-01 | $d^{-1}$ |
| Maximum nitrification rate in water sediment | r_nit_sed | $\tau_{nit,sed}$ | 2.000000e+01 | $d^{-1}$ |
| Oxygen half-saturation for nitrification | KO_nit | $K_{O_2,nit}$ | 5.000000e+02 | mg O m$^{-3}$ |
| Rate at which P reaches adsorbed/desorbed equilibrium | Pads_r | $\tau_{Pabs}$ | 4.000000e-02 | $d^{-1}$ |
| Freundlich Isothermic Const P adsorption to TSS in wc | Pads_Kwc | $k_{Pads,wc}$ | 3.000000e+01 | mg P kg TSS$^{-1}$ |
| Freundlich Isothermic Const P adsorption to TSS in sed | Pads_Ksed | $k_{Pads,sed}$ | 7.400000e+01 | mg P kg TSS$^{-1}$ |
| Oxygen half-saturation for P adsorption | Pads_KO | $K_{O_2,abs}$ | 2.000000e+03 | mg O m$^{-3}$ |
| Exponent for Freundlich Isotherm | Pads_exp | $N/A$ | 1.000000e+00 | none |
| Maximum denitrification rate | r_den | $\tau_{denit}$ | 8.000000e-01 | $d^{-1}$ |
| Oxygen half-saturation constant for denitrification | KO_den | $K_{O_2,denit}$ | 1.000000e+04 | mg O m$^{-3}$ |
| Rate of conversion of PIP to immobilised PIP | r_immob_PIP | $\tau_{Pimm}$ | 1.200000e-03 | $d^{-1}$ |

**Table B4.** Detritus parameters in eReefs biogeochemical model (B3p0).



| Description | Name in code | Symbol | Value | Units |
| --- | --- | --- | --- | --- |
| Sediment-water diffusion coefficient | EpiDiffCoeff | $D$ | 3.000000e-07 | m² s⁻¹ |
| Thickness of diffusive layer | EpiDiffDz | $h$ | 6.500000e-03 | m |
| Maximum growth rate of MA at Tref | MAumax | $\mu_{MA}^{max}$ | 1.000000e+00 | d⁻¹ |
| Natural (linear) mortality rate, macroalgae | MA_mL | $\zeta_{MA}$ | 1.000000e-02 | d⁻¹ |
| Nitrogen-specific leaf area of macroalgae | MAleafden | $\Omega_{MA}$ | 1.000000e+00 | m² g N⁻¹ |
| Respiration as a fraction of umax | Benth_resp | $\phi$ | 2.500000e-02 | none |
| net dissolution rate of sediment without coral | dissCaCO3_sed | $d_{sand}$ | 1.000000e-03 | mmol C m⁻² s⁻¹ |
| Grid scale to reef scale ratio | CHarea | $A_{CH}$ | 1.000000e-01 | m² m⁻² |
| Nitrogen-specific host area of coral polyp | CHpolypden | $\Omega_{CH}$ | 2.000000e+00 | m² g N⁻¹ |
| Max. growth rate of Coral at Tref | CHumax | $\mu_{CH}^{max}$ | 5.000000e-02 | d⁻¹ |
| Max. growth rate of zooxanthellae at Tref | CSumax | $\mu_{CS}^{max}$ | 4.000000e-01 | d⁻¹ |
| Radius of the zooxanthellae | CSrad | $r_{CS}$ | 5.000000e-06 | m |
| Quadratic mortality rate of coral polyp | CHmort | $\zeta_{CH}$ | 1.000000e-02 | (g N m⁻²)⁻¹ d⁻¹ |
| Linear mortality rate of zooxanthellae | CSmort | $\zeta_{CS}$ | 4.000000e-02 | d⁻¹ |
| Fraction of coral host death translocated. | CHremin | $f_{remin}$ | 5.000000e-01 | - |
| Rate coefficent for particle uptake by corals | Splank | $S_{part}$ | 3.000000e+00 | m d⁻¹ |
| Maximum daytime coral calcification | k_day_coral | $k_{day}$ | 1.320000e-02 | mmol C m⁻² s⁻¹ |
| Maximum nightime coral calcification | k_night_coral | $k_{night}$ | 6.900000e-03 | mmol C m⁻² s⁻¹ |
| Carbonate sediment dissolution rate on shelf | dissCaCO3_shelf | $d_{shelf}$ | 1.000000e-04 | mmol C m⁻² s⁻¹ |
| Age tracer growth rate per day | ageing_decay | $n/a$ | 1.000000e+00 | d d⁻¹ |
| Age tracer decay rate per day outside source | anti_ageing_decay | $\Phi$ | 1.000000e-01 | d⁻¹ |
| Bleaching ROS threshold | ROSthreshold | $\phi_{ROS}$ | 5.000000e-04 | mg O cell⁻¹ |
| Xanthophyll switching rate coefficient | Xanth_tau | $\tau_{xan}$ | 8.333333e-04 | s⁻¹ |
| Ratio of RCII to Chlorophyll a | chla2rcii | $A_{RCII}$ | 2.238413e-06 | mol RCII g Chl⁻¹ |
| Stoichiometric ratio of RCII units to photons | photon2rcii | $m_{RCII}$ | 1.000000e-07 | mol RCII mol photon⁻¹ |
| Maximum zooxanthellae expulsion rate | ROSmaxrate | $\gamma$ | 1.000000e+00 | d⁻¹ |
| Scaling of DetP to DOP, relative to N | r_RD_NtoP | $r_{RD_N toP}$ | 2.000000e+00 | - |
| Scaling of DOM to DIP, relative to N | r_DOM_NtoP | $r_{DOM_N toP}$ | 1.500000e+00 | - |
| Radius of Trichodesmium colonies | Tricho_colrad | $r_{Trichocolony}$ | 5.000000e-06 | m |

**Table B5.** Benthic parameters in eReefs biogeochemical model (B3p0), excluding seagrass




| Description | Name in code | Symbol | Value | Units |
|---|---|---|---|---|
| Maximum growth rate of SG at Tref | SGumax | $\mu_{SG}^{max}$ | 4.000000e-01 | $d^{-1}$ |
| Half-saturation of SG N uptake in SED | SG_KN | $K_{SG,N}$ | 4.200000e+02 | mg N m$^{-3}$ |
| Half-saturation of SG P uptake in SED | SG_KP | $K_{SG,P}$ | 9.600000e+01 | mg P m$^{-3}$ |
| Natural (linear) mortality rate aboveground seagrass | SG_mL | $\zeta_{SG_A}$ | 3.000000e-02 | $d^{-1}$ |
| Natural (linear) mortality rate belowground seagrass | SGROOT_mL | $\zeta_{SG_B}$ | 4.000000e-03 | $d^{-1}$ |
| Fraction (target) of SG biomass below-ground | SGfrac | $f_{below,SG}$ | 7.500000e-01 | - |
| Time scale for seagrass translocation | SGtransrate | $\tau_{tran,SG}$ | 3.330000e-02 | $d^{-1}$ |
| Nitrogen-specific leaf area of seagrass | SGleafden | $\Omega_{SG}$ | 1.500000e+00 | m$^2$ g N$^{-1}$ |
| Seagrass seed biomass as fraction of 63 % cover | SGseedfrac | $f_{seed,SG}$ | 1.000000e-02 | - |
| Sine of nadir Zostera canopy bending angle | SGorient | $\sin\beta_{blade,SG}$ | 5.000000e-01 | - |
| Compensation irradiance for Zostera | SGmlr | $E_{comp,SG}$ | 4.500000e+00 | mol m$^{-2}$ |
| Maximum depth for Zostera roots | SGrootdepth | $z_{root,SG}$ | -1.500000e-01 | m |
| Critical shear stress for SG loss | SG_tau_critical | $\tau_{SG,shear}$ | 1.000000e+00 | N m$^{-2}$ |
| Time-scale for critical shear stress for SG loss | SG_tau_time | $\tau_{SG,time}$ | 4.320000e+04 | s |
| Maximum growth rate of SGH at Tref | SGHumax | $\mu_{SGH}^{max}$ | 4.000000e-01 | $d^{-1}$ |
| Half-saturation of SGH N uptake in SED | SGH_KN | $K_{SGH,N}$ | 4.200000e+02 | mg N m$^{-3}$ |
| Half-saturation of SGH P uptake in SED | SGH_KP | $K_{SGH,P}$ | 9.600000e+01 | mg P m$^{-3}$ |
| Nitrogen-specific leaf area of SGH | SGHleafden | $\Omega_{SGH}$ | 1.900000e+00 | m$^2$ g N$^{-1}$ |
| Natural (linear) mortality rate, aboveground SGH | SGH_mL | $\zeta_{SGH_A}$ | 6.000000e-02 | $d^{-1}$ |
| Natural (linear) mortality rate, belowground SGH | SGHROOT_mL | $\zeta_{SGH_B}$ | 4.000000e-03 | $d^{-1}$ |
| Fraction (target) of SGH biomass below-ground | SGHfrac | $f_{below,SGH}$ | 5.000000e-01 | - |
| Time scale for Halophila translocation | SGHtransrate | $\tau_{tran,SGH}$ | 3.330000e-02 | $d^{-1}$ |
| Halophila seed biomass as fraction of 63 % cover | SGHseedfrac | $f_{seed,SGH}$ | 1.000000e-02 | - |
| Sine of nadir Halophila canopy bending angle | SGHorient | $\sin\beta_{blade,SGH}$ | 1.000000e+00 | - |
| Compensation irradiance for Halophila | SGHmlr | $E_{comp,SGH}$ | 2.000000e+00 | mol m$^{-2}$ |
| Maximum depth for Halophila roots | SGHrootdepth | $z_{root,SGH}$ | -8.000000e-02 | m |
| Critical shear stress for SGH loss | SGH_tau_critical | $\tau_{SGH,shear}$ | 1.000000e+00 | N m$^{-2}$ |
| Time-scale for critical shear stress for SGH loss | SGH_tau_time | $\tau_{SGH,time}$ | 4.320000e+04 | s |

**Table B6.** Seagrass parameters in eReefs biogeochemical model (B3p0).

...





| Description | Name in code | Symbol | Value | Units |
|---|---|---|---|---|
| Maximum growth rate of SGD at Tref | SGDumax | $\mu_{SGD}^{max}$ | 4.000000e-01 | $d^{-1}$ |
| Half-saturation of SGD N uptake in SED | SGD_KN | $K_{SGD,N}$ | 4.200000e+02 | $mg\ N\ m^{-3}$ |
| Half-saturation of SGD P uptake in SED | SGD_KP | $K_{SGD,P}$ | 9.600000e+01 | $mg\ P\ m^{-3}$ |
| Nitrogen-specific leaf area of SGD | SGDleafden | $\Omega_{SGD}$ | 1.900000e+00 | $m^2\ g\ N^{-1}$ |
| Natural (linear) mortality rate, aboveground SGD | SGD_mL | $\zeta_{SGD_A}$ | 6.000000e-02 | $d^{-1}$ |
| Natural (linear) mortality rate, belowground SGD | SGDROOT_mL | $\zeta_{SGD_B}$ | 4.000000e-03 | $d^{-1}$ |
| Fraction (target) of SGD biomass below-ground | SGDfrac | $f_{below,SGD}$ | 2.500000e-01 | - |
| Time scale for deep SG translocation | SGDtransrate | $\tau_{tran,SGD}$ | 3.330000e-02 | $d^{-1}$ |
| Deep SG seed biomass as fraction of 63 % cover | SGDseedfrac | $f_{seed,SGD}$ | 1.000000e-02 | - |
| Sine of nadir deep SG canopy bending angle | SGDorient | $\sin\beta_{blade,SGD}$ | 1.000000e+00 | - |
| Compensation irradiance for deep SG | SGDmlr | $E_{comp,SGD}$ | 1.500000e+00 | $mol\ m^{-2}$ |
| Maximum depth for deep SG roots | SGDrootdepth | $z_{root,SGD}$ | -5.000000e-02 | m |
| Critical shear stress for deep SG loss | SGD_tau_critical | $\tau_{SGD,shear}$ | 1.000000e+00 | $N\ m^{-2}$ |
| Time-scale for shear stress for deep SG loss | SGD_tau_time | $\tau_{SGD,time}$ | 4.320000e+04 | s |

**Table B7.** Deep seagrass parameters in eReefs biogeochemical model (B3p0).