# Peer review of "CSIRO Environmental Modelling Suite (EMS): Scientific description of the optical and biogeochemical models (vB3p0)."

_Geoscientific Model Development, 2019_

## Short Comment (SC1) · 14 Aug 2019

This comment addresses the compliance of the manuscript with GMD policy on code and data availability. The issues raised here must be satisfactorily addressed before a revised manuscript can be accepted for publication.

Thank you for publishing the full source of your model. The one technical issue is that GitHub, while an excellent development platform, is not a suitable persistent archive for citable code. Indeed, GitHub themselves tell you this and provide instructions as to how to obtain a citable archive of a particular release of a GitHub project using their

interface to the Zenodo archive[1]. Please provide a proper persistent archive of the exact version of the code you are documenting (the GitHub Zenodo integration makes this very easy).

Please also ensure that any files required to run the verification experiments are also archived (it's possible this data is already on the thredds server you currently cite, in which case simply making this explicit would suffice).

[1]https://guides.github.com/activities/citable-code/

---

## Referee Comment (RC1) · Anonymous Referee #1 · 6 Dec 2019

The manuscript provides a compilation of many individuals' past coding efforts to develop the EMS. The model, consisting of biogeochemical, optical, and sedimentary components, is within the scope of GMD and scientifically relevant. Collecting the mathematical descriptions of the major model components into a single document linked to the model code and User Guide might improve convenience for EMS users. The authors provide sufficient documentation to reproduce their results. The language is clear and the presentation is well-structured, though a spellchecker should be run on the document as there are a number of typos.

Most if not all of the material has been previously published in the peer-reviewed literature. Hence, the material cannot be considered novel, nor does the manuscript represent a substantial advance in modelling. Because this manuscript represents a collection of previously published work, little effort is spent explaining why parameterisations are the way they are. While methods and assumptions may be valid, they are not always clearly outlined. References to the primary literature are given, but the manuscript cannot be understood by a non-expert reader as a stand-alone document. Likewise, the model is given a perfunctory evaluation that includes no discussion of the biases. More detailed assessments are cited, but the reader of the present manuscript is left with no real understanding of why the model performs well (and what biases may be due to) in the examples provided.

It is difficult to make recommendations that could improve this manuscript, in its present form, with respect to the principal review criteria because added detail with respect to model formulation and more complete model assessments have already been published. Reproducing earlier work at length is not feasible. My recommendation is to refocus the manuscript to a summary of the equations (as already done), followed by a meta-analysis of model performance across past applications. A thorough discussion of systematic biases across ecosystems could represent a major advance for the model. Included in this meta-analysis should be a description of how the many "not attributed" parameters (in the supplement) are tuned. Perhaps the authors could even go further, and address those biases by presenting an improved model.

More specific comments can be found in the Supplement.

Please also note the supplement to this comment:
https://www.geosci-model-dev-discuss.net/gmd-2019-115/gmd-2019-115-RC1-supplement.pdf
* * *
[Figure]

**Supplement:**

Review of "CSIRO Environmental Modelling Suite (EMS): Scientific description of the optical and biogeochemical models (vB3p0).

The manuscript provides a compilation of many individuals' past coding efforts to develop the EMS. The model, consisting of biogeochemical, optical, and sedimentary components, is within the scope of GMD and scientifically relevant. Collecting the mathematical descriptions of the major model components into a single document linked to the model code and User Guide might improve convenience for EMS users. The authors provide sufficient documentation to reproduce their results. The language is clear and the presentation is well-structured, though a spellchecker should be run on the document as there are a number of typos.

Most if not all of the material has been previously published in the peer-reviewed literature. Hence, the material cannot be considered novel, nor does the manuscript represent a substantial advance in modelling. Because this manuscript represents a collection of previously published work, little effort is spent explaining why parameterisations are the way they are. While methods and assumptions may be valid, they are not always clearly outlined. References to the primary literature are given, but the manuscript cannot be understood by a non-expert reader as a stand-alone document. Likewise, the model is given a perfunctory evaluation that includes no discussion of the biases. More detailed assessments are cited, but the reader of the present manuscript is left with no real understanding of why the model performs well (and what biases may be due to) in the examples provided.

It is difficult to make recommendations that could improve this manuscript, in its present form, with respect to the principal review criteria because added detail with respect to model formulation and more complete model assessments have already been published. Reproducing earlier work at length is not feasible. My recommendation is to refocus the manuscript to a summary of the equations (as already done), followed by a meta-analysis of model performance across past applications. A thorough discussion of systematic biases across ecosystems could represent a major advance for the model. Included in this meta-analysis should be a description of how the many "not attributed" parameters (in the supplement) are tuned. Perhaps the authors could even go further, and address those biases by presenting an improved model.

Specific comments:

P3 line 15  replace "which" with "that"

P3 line 20 "is unlimited"

P4 line 16 replace "to" with "by"

P4 line 22 "to geometric"

P5 line 4 replace "with" with "and"

P6 lines 2-5 Please delete those 2 sentences.

P6 line 14 "dramatic"

P8 Fig 4. IPIP is not the same abbreviation used in the text or Figure 5. What do the stars mean? What does the dashed line mean around meteorology?

P9 line 1 replace "recalculated" with "diagnosed", replace "optimise" with "reduce"

P9 line 2 delete "and accuracy", replace "recalculated at" with "diagnosed for"

P10 line 16/Eqn 1 wc vs wsink- not consistent

P12 line 30 "calculate"

Figs 7&8 should be referenced earlier than they are

P13 line 5- P15 line 4 Are the 2 schemes assessed?

P15 line 13 "include"

P18 Fig. 9 caption "calculate u"

P22 line 10 "as good"

P23 line 18 "functional groups"

P25 lines 5-7 Please break this sentence in two.

P27 Please define HPLC the first time it is used

P32 line 3 delete "is"

P32 line 6 replace "is" with "are"

P35 line 19 "additional"

P37 line 13 "inefficient"

P38 Table 14 caption "lost to the water"

P41 line 31 This sentence is an outlier and could be moved to earlier in the text

P43 line 10 "we will"

P44 line 18 "sit"

P44 line 19 replace "the" with "be"

P56 line 11 "that is"

P58 line 9 "while mineralogy"

P62 line 6 "to the water"

P65 line 15 "adjusts"

P69 line 3 "drawdown"

P69 line 12 "have not been"

P71 line 5 "are the same"

P72 line 5 "by instead"

P76 line 8 "slightly"

P79 line 27 "its use"

P79 line 32 remove "available"

P81 line 14 "such as"

P82 line 1 "have allowed"

---

## Referee Comment (RC2) · Marcello Vichi (Referee) · 16 Jan 2020

This manuscript offers a description of the Environment Modelling Suite (EMS) developed over several years in Australia by CSIRO. It is based on a set of existing publications, as well as some on-line documentation. Despite the limited amount of original material included in this manuscript, I see the value of its publication, because it would offer a single point entry for new users interested in approaching such a complex suite. It would also be a supporting reference document for further scientific applications in other regions.

**1 Main comments**

I have some major concerns, that I would like the authors to address before resubmitting the manuscript

1. It is a very lengthy manuscript, that does not have distinct points of entry. In its current version, it needs to be read sequentially in order to appreciate the various components. I would suggest the authors to separate the pelagic from the benthic component, especially because the community of scientists is rather different, and this would also allow to clarify some aspects of the coupling with the transport processes that are a bit overlooked. This would help the referees to focus more on the original components of the model because they would be closer to their expertise, and would provide a more informed assessment.

2. In the current state, it looks more of a hybrid combination between a user manual, a scientific model description, a technical report and a summary of model applications. I would suggest the authors to better clarify their aims and decide which approach is the main one. In particular, it is not clear how the authors decided to include a full description of certain aspects, while for others they refer the reader to published literature in toto. Three unique features are listed in the Introduction (pag 6), but it is not clear how innovative they are with respect to other published works (e.g. Dutkiewicz et al., 2015).

3. I am concerned about the lack of discussion on the model science and what differentiates it from the other available open-source models. There is a cursory introduction on how the model differs from other approaches, although they are lumped into being Fasham-like, which is actually inaccurate, since several models consider stoichiometry and the internal storage of nurient and energy (for instance Lancelot et al., 1993,2000; Baretta et al., 1995; Vichi et al., 2007). Aumont et al., 2015 and Butenschoen et al 2016 are indeed referenced, but to remark the need for thorough description of models). Rather strikingly, only by looking at Table 6 the (skilled) readers understands that

the main currency of the model is nitrogen and that biomass is measured in N concentration units. I am not suggesting to have an additional lengthy discussion on the type of biogeochemical models, but just to make the reader aware of what are the peculiarities of this approach with respect to others. The scaling or geometric constraints are indeed a special feature of this modelling approach, although it should be clarified that models with multiple functional types also includes implicit size considerations in the value of the parameters . The concept that geometric description is a mean to reduce parameter uncertainties (pag 4, L1-3) would also require further clarification, especially because this model implements only two size classes with generic functions, which means it has a limited range of applications in the coastal region. If the Si or Fe cycles would need to be resolved, or specific harmful algae, then additional parameters would be forcibly needed.

**2 Detailed comments**

P6L14-15 There should be some description on what Fig. 3 shows. Since it is used in the introduction, I would expect some more context. However, the authors suggest me to go and read Baird et al. 2016 to understand the figure (and what is GBR4 at this stage?) Sec 1.1: This is the most important section when engaging with a manuscript of this size. However, it offers only a quick list of the upcoming sections, something akin to what is offered in shorter manuscripts. The authors state that descriptions are sorted by processes, but I would argue with this statements, since some processes are spread across various sections and there are several cross-references that interrupt the flow of the description. I would also strongly advocate against the offered solution to the reader (L26-28): to combine all the various process terms to obtain the complete differential equation. This is in my opinion what makes the manuscript more difficult to read, since there is no full appreciation of the dynamics of each state variable. This approach was also followed by Vichi et al. (2007), but in that case, a specific notation

was introduced and the full dynamics were presented.

Sec 2. I have a few problems with the organization of this section. The EMS biogeo-chemical model has not been introduced yet (while the next Fig. 4 and 5 are about the biogeochemical model and not the optics), but the reader is offered an initial descrip-tion with no references of the science of IOP and AOP. I would suggest to invert the order and first illustrate the model structure, then highlighting the details of the optical model. Otherwise, the authors can opt for a shorter manuscript that would focus on the optical model only if they consider this the most innovative component. At Pag. 9 L19 microalgae are mentioned, but there is no mathematical equivalence neither an explanation of what small and large means. I recognize that there are tables later in the model where size classes are provided, but a set of ranges should be given from the beginning, especially because of the emphasis on geometric constraints.

P10L12 The authors should state what kind of approximation they make when consid-ering dissolved and particulate concentrations of pelagic variables affected by transport processes. The basic approximation of fluid dynamics is the continuum hypothesis, which should also be considered for biogeochemistry (e.g. O'Brien and Wroblewski, 1973; Vichi et al., 2007). I understand that this is an aspect that was overlooked in the early works (Nihoul, 1975; Fasham et al, 1990, etc), but it is nowadays essential given the increasing resolutions of hydrodynamic models.

P10L17 I would suggest the authors to give information on whether the model has been coupled with other hydrodynamic models.

P10L22 This is one of the many cases where the authors start an explanation and then drop it abruptly referring to published papers (see my main comment 2 above) Sec 3.2. What is the difference with Sec. 2.1 and why two separate sections are needed?

P23L18 Please define a "function group"

P23L23 I think that the concept of internal reserves is an essential one to understand

the equations. Nevertheless, the authors refer to Fig. 3 in another paper (see main comment 2)

P24 Table 4 caption: the authors should explain what they mean by cells here. Mean population characteristics? This is not introduced anywhere in the main text.

P24L7-8 and L10 The variable R* is introduced and used without any context. I would suggest to show an equation defining R* instead of a figure.

P25L7 Menten

Sec 3.3.3 I am struggling with this section. There is very little structure in the description. I can follow the flow because of my experience in numerical models, but to my understanding this manuscript should introduce the model to a larger audience and expand its usage beyond the group of developers (main comment 1 above). I think the authors need to make a decision on what is the narrative approach they want to have, either from the point of view of the optics or from the biogechemistry and ecosystem viewpoint. I understand that from the point of view of an optical model, the absorption cross section is independent of the physiology and definition of phytoplankton. However, treating separately eq 6 and the variable rho does make the reading more difficult. At the risk of being pedantic, I think one should first present the dynamics in eq 37 and then illustrate the various terms. I'm particularly thinking about a student user who would like to learn the model and who may not have a full understanding of the underlying physiology.

P27L20 and L28 Table 6 is referenced before Table 5, which is the one listing the state variables whose dynamics is listed in Table 6. I would strongly suggest that a full list of the state variables is given at the very beginning when the biogeochemical model is introduced. In L20, Tab 5 presents state and diagnostic variables, not equations.

P28L4 Please refer to the eq numbers and not just the table.

P30L2-3 This is an important information that should be given in the introduction, and

briefly expanded upon to clarify how this models is positioned in the context of the existing theories and methodologies. Sec 3.6. The mathematical formulation and equations are very little detailed for this component. Table 14 containing the dynamics is just referenced, and there is not a single equation describing the biomass evolution. It is also not clear from the beginning that the zooplankton variable has no variable stoichiometry (or internal reserves).

P35L13 This is a coarse over-generalization which does not pay much attention to the model development occurred in the past 30 years. Models that use preference factors and a food matrix do not have this issue (Gentleman et al., 2003). Sec 3.6.1 Grazing is actually not illustrated in this section. Table 15 is just mentioned but the specific terms not described. It is not much clear what is the food web accessed by zooplankton, apart from the first generic sentence at the beginning of sec. 3.2. How the fluxes between the state variables are actually computed is not clear. Table 12 What is the difference between variable $m_n$ in Table 7 and variable $m_B$? Sec. 3.8 I suggest the authors to make specific reference to the numbered equations and not to the Table containing them (also check the typo at line 9 same page "zooplankton plankton")

Sec 3.9 I would see the section on non-grazing plankton mortality to be more pertinent to plankton dynamics, and less to zooplankton grazing. Is zooplankton mentioned at L6 because this is a loss term for all the plankton? It is rather confusing, and proper structuring would be helpful. Sec 3.10 I guess the author means gas exchange at the surface of the ocean here.

P41L3 The variable is $u_{10}$. This is the cubic function.

P41L18 positive

P41L31 This seems a fragment with no connection with the previous paragraph.

P42L22 Please clarify what "vertical order" means.

P43L18 Please refer where the diffusivity values have been taken. I could not find a

table with the values.

Eq113 and other. I would suggest the authors to use named constant for the stoichiometric coefficients, to allow identifying which conversion is actually being done. Also, the authors should briefly illustrate the rationale for the use of this multiple minimum function, which I guess is linked to the maintenance of the constant stoichiometry in this functional group biomass.

P49L1 This is another example of the main problem I have with this manuscript (main comment 2). The same can be found at L29 w=in the case of coral processes. The authors seem to have cherry-picked what should be described and what should be left for the reader to scavenge through the literature. Please explain if there is an underlying rationale or a unified criterion.

P49L6-7 These are microalgae, so I wonder why the authors decided not to use the same dynamics described earlier.

P57 How is $M$ computed?

P58L5 With the use of coding style, the manuscript turns towards the user manual. This is the first time that code is used in the document. I am not against it if properly explained, but makes the manuscript less coherent (see main point 2).

P58L7-8 Make reference to Table 34 where the variables are listed.

Sec 5.2.1 I am a bit confused here, because light is not an environmental variable that controls the dynamics in the sediment model. What is the difference with Sec. 4.2.2? I understand that the optical model is a major part of this manuscript, but then I would separate the biogeochemistry from the optics in two different manuscripts and make reference when needed instead of mixing the descriptions (see main point 1). Sec 5.3.1 and 5.3.2 There is no reference to equation numbers in these short sections. Sec. 6 I understand that section 6 collects all processes that are common to the pelagic, epibenthic and benthic processes. Therefore, I would expect to find here

cross-references to the other sections where they have been described, as well as the inverse (when the dynamics are first illustrated, inform the reader that a certain term is considered a common process and found in Sec. 6).

P68L9-11 I agree with this sentence, but this would deserve some more discussion. According to the Introduction, the model is designed to be generic, and the combination of physical and biogeochemical processes may lead to stiffness (this is cursory mentioned somewhere in the text or in the captions). The authors can refer to Butenschön et al., 2012 for an illustration of the problem. I would also ask the authors to clarify what do they mean with the "time step of the splitting"? Different time steps can be used for the various steps, but I am not familiar with a splitting time step. (Please explain what GBR1-4 mean in the table)

P68L13 The choice of a 5th order ODE should be justified, especially in the case of empirically-derived parameterizations. Please also say here which scheme is used and that it includes adaptive stepping. This information is given further below somewhere.

P69L2-4 This justification raises some concerns. The method is explicit, which would actually lead to instabilities that would require a time step shortening. Is this what the authors mean?

P69L16 Please explain the term "between". Is this implicit or explicit? According to Table 42, it uses the same time step as the ecology, but not clear what light environment is used for the ecology.

P70L9-10 This sentence is not clear. I understand that the manuscript is not about the coupling between the physics and the ecology, but this sentence would require more context. Please refer to the table where the number of levels in the described applications are listed.

Sec 8 I would argue that this manuscript does not offer any model evaluation beyond what has been already published. Is the assessment done here a technical check that

version vB3p0 produces the same results as vB2p0 described in Skerratt et al. (2019)?

Sec 9 and 10. I am not sure these sections help the manuscript concept, and they confirm my impression of the lack of coherence in the original idea of the presentation (point 1 above). I would suggest the authors to reconsider their structure and to move some sections to the appendix or to on line material.

Sec 11 I have some concerns about the content of this discussion. I would expect Sec 11.1 in the introduction. I cannot really see any discussion here as I have indicated in my main point 3 above.

**Bibliography**

Baretta, J.W., Ebenhöh, W., Ruardij, P., 1995. The European Regional Seas Ecosystem Model, a complex marine ecosystem model. Journal of Sea Research 33, 233–246.

Butenschön, M., Zavatarelli, M., Vichi, M., 2012. Sensitivity of a marine coupled physical biogeochemical model to time resolution, integration scheme and time splitting method. Ocean Modelling 52–53, 36—53.

Gentleman, W., Leising, A., Frost, B., Strom, S., Murray, J., 2003. Functional responses for zooplankton feeding on multiple resources: a review of assumptions and biological dynamics. Deep-Sea Research Part II 50, 2847–2875.

Lancelot, C., Hannon, E., Becquevort, S., Veth, C., Baar, H.J.W.D., 2000. Modeling phytoplankton blooms and carbon export production in the Southern Ocean: dominant controls by light and iron in the Atlantic sector in Austral spring 1992. Deep-Sea Research Part II 47, 1621–1662.

Lancelot, C., Mathot, S., Veth, C., de Baar, H., 1993. Factors controlling phytoplankton

ice-edge blooms in the marginal ice-zone of the northwestern Weddell Sea during sea ice retreat 1988: field observations and mathematical modelling. Polar Biology 13, 377–387.

Nihoul, J.C.J. (Ed.), 1975. Modelling of Marine Systems, Elsevier Oceanography Series. Elsevier.

O'Brien, J.J., Wroblewski, J.S., 1973. On advection in phytoplankton models. J. Theor. Biology 38, 197–202.

Vichi, M., Pinardi, N., Masina, S., 2007. A generalized model of pelagic biogeochemistry for the global ocean ecosystem. Part I: theory. Journal of Marine Systems 64, 89–109. https://doi.org/10.1016/j.jmarsys.2006.03.006

---

## Author Comment (AC1) · 24 Mar 2020

Dear Dr. Hargraves,

Please find below comments by the executive editor (David Ham) and two reviewer's comments in blue, and my response in black. I am very thankful to the reviewers, and in particular for the in-depth review provided by the second reviewer (Dr. Marcello Vichi), that has much improved the manuscript.

Yours sincerely,

Mark Baird

**Executive Editor comment.**

This comment addresses the compliance of the manuscript with GMD policy on code and data availability. The issues raised here must be satisfactorily addressed before a revised manuscript can be accepted for publication. Thank you for publishing the full source of your model. The one technical issue is that GitHub, while an excellent development platform, is not a suitable persistent archive for citable code. Indeed, GitHub themselves tell you this and provide instructions as to how to obtain a citable archive of a particular release of a GitHub project using their interface to the Zenodo archive1 . Please provide a proper persistent archive of the exact version of the code you are documenting (the GitHub Zenodo integration makes this very easy).

We will retain the GitHub archive, but have also added an entry in the permanent CSIRO software collection which has the citation:

CSIRO (2019): EMS Release v1.1.1. v1. CSIRO. Software Collection. https://doi.org/10.25919/5e701c5c2d9c9.

[email advice from Dr. Ham suggested this would meet the journal's requirements].

Please also ensure that any files required to run the verification experiments are also archived (it's possible this data is already on the thredds server you currently cite, in which case simply making this explicit would suffice).

Section 13 (Section 11 in the original manuscript) provides the means to download a complete model system with all forcing and configuration files.

**Reply Reviewer #1.**

*The manuscript provides a compilation of many individuals' past coding efforts to develop the EMS. The model, consisting of biogeochemical, optical, and sedimentary components, is within the scope of GMD and scientifically relevant. Collecting the mathematical descriptions of the major model components into a single document linked to the model code and User Guide might improve convenience for EMS users. The authors provide sufficient documentation to reproduce their results. The language is clear and the presentation is well-structured, though a spellchecker should be run on the document as there are a number of typos.*

Thank you.

*Most, if not all, of the material has been previously published in the peer-reviewed literature. Hence, the material cannot be considered novel, nor does the manuscript represent a substantial advance in modelling.*

As pointed out by the reviewer, a number of individual process parameterisations in the model have been published before. When these published papers contain complete sections that are exactly the same as version B3p0, such as the seagrass and coral model parameterisations, we have not re-derived the model equations, but just included tables of parameter values and equations.

Many of the formulations have not been published before, or have been updated from much earlier work. But most importantly, some aspect of the model approach only become apparent, and reproducible, when the entire model is described together.

To summarise the unpublished components (numbering uses original manuscript):

**Previously unpublished formulations.**

- Preferential ammonia uptake that considers the physical limit to ammonia uptake due to diffusion to the uptake surface.
- Oxygen balance that includes chemical oxygen demand as well as the oxygen content of nitrate. This requires new equations for the differing implications for oxygen of anerobic versus aerobic remineralisation of organic matter.
- The zooplankton grazing on model microalgae that contain the definition of internal reserves used in EMS
- Phosphorus immobilisation.
- Gas exchange and equilibrium carbon chemistry – a combination of other studies, but also adapted to the EMS model structure.

**Updated formulations building on previous work.**

- Optical model. An earlier manuscript (Baird et al., 2016) described a new optical model that included some IOPs, but this work much expands the use of mineralogical-based inorganic particulate IOPs, as well as an extensive library of pigment-specific absorption coefficients.
- Further expansion of optical implications of benthic habitat, and, in particular, the role of symbionts (Eq. 212-214).
- Aragonite saturation state -depended coral calcification and dissolution (Table 33).
- Revised calcification rate calculation based on the physiological state of symbionts (Eq. 192)
- Revised phosphorus absorption-desorption model (Section 3.5.3).
- Revised nitrogen-fixing *Trichodesmium* formulation (Section 3.4).
- Differential breakdown rates of detrital and organic phosphorus when compared to nitrogen.
- Improved nitrification / denitrification model with new sigmoidal formulation for stability at the transition between oxic and anoxic conditions.

**Novel aspects apparent in whole model description.**

- Derivation of planktonic rates using the individual rather than the population.
- Systematic use of geometric descriptions of ecological rates.

- Process-based scientific description so that individual formulations can be easily extracted for use by other modellers.
- The details of mass balance equations of C, N, P, and O that can be dissolved, contained within particles, and that are distributed within 3D volumes of water column and porewaters, and as 2D masses on the sediment-water column interface.

Finally, readers might also find some aspects of the model presentation worth considering. For example, we used integer values for stoichiometric ratios in the equations (i.e. $\frac{16}{106}\frac{12}{14}$), consistently throughout the manuscript. I think this improves readability over a large number of stoichiometric coefficients.

Because this manuscript represents a collection of previously published work, little effort is spent explaining why parameterisations are the way they are. While methods and assumptions may be valid, they are not always clearly outlined.

I don't think this is a fair comment. The reason the manuscript is so long is because there are descriptions of why certain formulations were chosen. For example, arguably the most important functional form in the whole model, relating benthic cover to biomass, has a large section dedicated to its explanation (beginning of section 6). Another example of a basic explanation is Fig. 12, the schematic of the growth from reserves.

Nonetheless, the 2nd reviewer wanted a more thorough introduction of internal reserves, and this is now given.

References to the primary literature are given, but the manuscript cannot be understood by a non-expert reader as a stand-alone document.

Following the suggestion of the other reviewer, three new sections, "Structure of the model description", "Presentation of process equations" and "Model stoichiometry", have been added at the beginning. These hopefully provide more background for the non-expert reader.

Likewise, the model is given a perfunctory evaluation that includes no discussion of the biases. More detailed assessments are cited, but the reader of the present manuscript is left with no real understanding of why the model performs well (and what biases may be due to) in the examples provided.

I agree the model evaluation is brief. A thorough model evaluation would add much to the length of the manuscript, and repeat that already undertaken in other work, particularly Skerratt et al., 2019.

A thorough discussion of systematic biases across ecosystems could represent a major advance for the model. Included in this meta-analysis should be a description of how the many "not attributed" parameters (in the supplement) are tuned. Perhaps the authors could even go further, and address those biases by presenting an improved model.

An analysis of how a single model performs across many environments is an interesting task, and versions of this have been undertaken by, for example, studying how differing global biogeochemical models perform in different regions of the world. But the coastal environment, with its multiple habitat types make this virtually impossible.

**Reviewer #2**

This is an extremely in-depth review, and I greatly appreciate all of the comments by the reviewer. When submitting to this journal I was hoping for such a review to help us improve our work. Thank you so much.

This manuscript offers a description of the Environment Modelling Suite (EMS) developed over several years in Australia by CSIRO. It is based on a set of existing publications, as well as some on-line documentation. Despite the limited amount of original material included in this manuscript, I see the value of its publication, because it would offer a single point entry for new users interested in approaching such a complex suite. It would also be a supporting reference document for further scientific applications in other regions.

Thank you.

1 Main comments

I have some major concerns, that I would like the authors to address before resubmitting the manuscript

1. It is a very lengthy manuscript, that does not have distinct points of entry.

I apologise for the length. To reduce its size I have placed in the supplementary material components that lead to the calculation of remote-sensing reflectance, and thus true colour. These are essentially diagnostic variables, and therefore not necessary for the core model. I have also removed "Model User" components.

> In its current version, it needs to be read sequentially in order to appreciate the various components. I would suggest the authors to separate the pelagic from the benthic component, especially because the community of scientists is rather different, and this would also allow to clarify some aspects of the coupling with the transport processes that are a bit overlooked. This would help the referees to focus more on the original components of the model because they would be closer to their expertise, and would provide a more informed assessment.

It is true that different scientific communities often exclusively model pelagic or benthic processes. For example, benthic modellers often consider habitats and diversity rather than nutrients. However the closest model to EMS, the ERSEM model, also describes pelagic and benthic processes in the same GMD paper:

Butenschön et al., ERSEM 15.06: a generic model for marine biogeochemistry and the ecosystem dynamics of the lower trophic levels Geosci. Model Dev., 9, 1293–1339, 2016

Also, I would argue that the key innovation of this model, as stated on page 4 line 13 of the original submission, is:

"the most important consequence of using geometric constraints in the BGC model is the representation of benthic flora as two dimensional surfaces, while plankton are represented as three dimensional suspended objects"

This innovation is most apparent when the pelagic and benthic models are described together.

2. In the current state, it looks more of a hybrid combination between a user manual, a scientific model description, a technical report and a summary of model applications. I

would suggest the authors to better clarify their aims and decide which approach is the main one. In particular, it is not clear how the authors decided to include a full description of certain aspects, while for others they refer the reader to published literature in toto.

The primary aim of the paper is a scientific model description. The two sections that are referenced off in toto are the coral and seagrass sections, although the equations are retained. The rationale for this is that these two sections are described virtually exactly in the journal articles as in the software version (B3p0), and the journal they are published in, Ecological Modelling, allows full scientific descriptions.

I have moved two sections that looked 'code'-like to the supplementary material.

3. Three unique features are listed in the Introduction (pag 6), but it is not clear how innovative they are with respect to other published works (e.g. Dutkiewicz et al., 2015).

The manuscript splits the unique aspects of the model into two lists, one set derived with geometric origins (which I would argue is innovative too), and a second more generic list that this comment refers to (p6). As the reviewer points, the second list involves new equations, but some of them are approximations when others such as Dutkiewicz have equivalent, and perhaps better solutions. I have added a brief reference to the approximate nature of these features.

4. I am concerned about the lack of discussion on the model science and what differentiates it from the other available open-source models. There is a cursory introduction on how the model differs from other approaches, although they are lumped into being Fasham-like, which is actually inaccurate, since several models consider stoichiometry and the internal storage of nutrient and energy (for instance Lancelot et al., 1993,2000; Baretta et al., 1995; Vichi et al., 2007). Aumont et al., 2015 and Butenschoen et al 2016 are indeed referenced, but to remark the need for thorough description of models).

New section in the Discussion added: "Comparison with other marine biogeochemical models".

Rather strikingly, only by looking at Table 6 the (skilled) readers understands that the main currency of the model is nitrogen and that biomass is measured in N concentration units. I am not suggesting to have an additional lengthy discussion on the type of biogeochemical models, but just to make the reader aware of what are the peculiarities of this approach with respect to others.

Thank you. Upon re-reading, I agree the section on microalgae growth does not give enough context to the reader. Further that when it does come in, it follows a particular line of equation development. I have added a significant contextual start to this section before pursing the particular model I have chosen.

The scaling or geometric constraints are indeed a special feature of this modelling approach, although it should be clarified that models with multiple functional types also includes implicit size considerations in the value of the parameters.

In the modelling of plankton, geometric and size-based properties are similar concepts, and I have included a reference to size-based trait modelling. However geometric-constraints is broader than just size, as illustrated in the next section (on 2D leaves vs 3D microalgae).

The concept that geometric description is a means to reduce parameter uncertainties (pag 4, L1-3) would also require further clarification, especially because this model implements only two size classes with generic functions, which means it has a limited range of applications in the coastal region.

Yes. Thank you. An example makes a more powerful point. The best illustration is in the relationship between benthic cover and seagrass nitrogen. I have added a few sentences when I get to this example on the next page of the manuscript.

If the Si or Fe cycles would need to be resolved, or specific harmful algae, then additional parameters would be forcibly needed.

Yes, some new stoichiometric coefficients for the new elements. The EMS ecology library does contain a harmful algal bloom category, but because it is a process that can be excluded, and wasn't used in the simulations shown in the results, we have decided not to include a description here.

2 Detailed comments

P6L14-15 There should be some description on what Fig. 3 shows. Since it is used in the introduction, I would expect some more context. However, the authors suggest me to go and read Baird et al. 2016 to understand the figure (and what is GBR4 at this stage?)

I have removed this figure as true colour is a diagnostic variables and not essential.

Sec 1.1: This is the most important section when engaging with a manuscript of this size. However, it offers only a quick list of the upcoming sections, something akin to what is offered in shorter manuscripts. The authors state that descriptions are sorted by processes, but I would argue with this statements, since some processes are spread across various sections and there are several cross-references that interrupt the flow of the description.

The biogeochemical processes are distinct, as demonstrated by the fact that each table of equations conserves mass. But the reviewer might be getting at the fact that light absorption appears in two places, because it is part of the optical model and the ecological process of photosynthesis. Following suggestions elsewhere in this review, I have combined the optical model components, and now make this structure clearer from the beginning.

I would also strongly advocate against the offered solution to the reader (L26-28): to combine all the various process terms to obtain the complete differential equation. This is in my opinion what makes the manuscript more difficult to read, since there is no full appreciation of the dynamics of each state variable. This approach was also followed by Vichi et al. (2007), but in that case, a specific notation was introduced and the full dynamics were presented.

The reviewer is supporting our approach of presenting partial differential equations for each processes. Looking at Vichi et al., (2007) this paper has an excellent introduction to how the model presented. I have copied this approach, adding three new subsections, "Structure of the model description", "Presentation of process equations" and "Model stoichiometry".

Sec 2. I have a few problems with the organization of this section. The EMS biogeochemical model has not been introduced yet (while the next Fig. 4 and 5 are about the biogeochemical model and not the optics), but the reader is offered an initial description with no references of the science of

IOP and AOP. I would suggest to invert the order and first illustrate the model structure, then highlighting the details of the optical model.

I mostly agree with these comment, and have re-organised the manuscript by:

1. Extracting all components of the optical model that involve calculating the light field into a separate section titled "Optical model", with subsections "Pelagic optical model", "Epibenthic optical model" and "Sediment optical model".
2. Point 1 allows for a section to describe the IOP / AOP terminology that is very familiar to optical modellers, but is only beginning to be adopted by biogeochemical modellers.
3. Removed all calculations of remote-sensing reflectance. This is a diagnostic variables, and for sake of brevity description of all diagnostic variables has been removed from the manuscript.

I have put the optical model before the BGC model because the EMS suite can be used with an optical model but no biogeochemical processes, but there can be no biogeochemical model (at least with autotrophs) without an optical model.

Otherwise, the authors can opt for a shorter manuscript that would focus on the optical model only if they consider this the most innovative component.

No, I am emphasising the BGC model. In fact, as mentioned later, I have removed the components describing the calculation of remote-sensing reflectance.

At Pag. 9 L19 microalgae are mentioned, but there is no mathematical equivalence neither an explanation of what small and large means. I recognize that there are tables later in the model where size classes are provided, but a set of ranges should be given from the beginning, especially because of the emphasis on geometric constraints.

OK. Included comment on size and function role of different plankton.

P10L12 The authors should state what kind of approximation they make when considering dissolved and particulate concentrations of pelagic variables affected by transport processes. The basic approximation of fluid dynamics is the continuum hypothesis, which should also be considered for biogeochemistry (e.g. O'Brien and Wroblewski, 1973; Vichi et al., 2007). I understand that this is an aspect that was overlooked in the early works (Nihoul, 1975; Fasham et al, 1990, etc), but it is nowadays essential given the increasing resolutions of hydrodynamic models.

Text add:

The advection-diffusion terms of Eq. 1, based on the continuum hypothesis for a fluid (Vichi et al., 2007), are solved by either an in-line advection scheme with the baroclinc timestep of the hydrodynamic model, or an offline transport scheme using a potentially much longer timestep (Gillibrand and Herzfeld, 2016). Options in EMS include mass conservative Lagrangian and flux-form schemes described in Herzfeld (2006) and Gillibrand and Herzfeld (2016).

P10L17 I would suggest the authors to give information on whether the model has been coupled with other hydrodynamic models.

Included reference to application of the plankton model coupled to the Princeton Ocean Model.

P10L22 This is one of the many cases where the authors start an explanation and then drop it abruptly referring to published papers (see my main comment 2 above)

Yes, but this part is well described elsewhere, and the manuscript is already too long.

Sec 3.2. What is the difference with Sec. 2.1 and why two separate sections are needed?

Agreed. Removed.

P23L18 Please define a "function group"

I meant functional groups. Thank you.

P23L23 I think that the concept of internal reserves is an essential one to understand the equations. Nevertheless, the authors refer to Fig. 3 in another paper (see main comment 2)

My mistake. A version of this figure is actually in this paper. Cross-reference fixed.

P24 Table 4 caption: the authors should explain what they mean by cells here. Mean population characteristics? This is not introduced anywhere in the main text.

I have removed 'cells' in the Table caption. However, the concept of 'cell' vs. 'population' is important in the model formation. This even gets a mention in the abstract "A second focus has been on, where possible, the use of geometric derivations of physical limits to constrain ecological rates, which generally requires population-based rates to be derived from initially considering the size and shape of individuals."

Since the most important implications of cell vs. population are in microalgae, I have included a new section 5.1.5 in the revised manuscript.

P24L7-8 and L10 The variable $R^*$ is introduced and used without any context. I would suggest to show an equation defining $R^*$ instead of a figure.

Agreed. Beginning of "Microlagae growth" now includes a section on reserves.

P25L7 Menten

Spelling fixed.

Sec 3.3.3 I am struggling with this section. There is very little structure in the description. I can follow the flow because of my experience in numerical models, but to my understanding this manuscript should introduce the model to a larger audience and expand its usage beyond the group of developers (main comment 1 above). I think the authors need to make a decision on what is the narrative approach they want to have, either from the point of view of the optics or from the biogechemistry and ecosystem viewpoint.

The focus is biogeochemistry. Optics component has been scaled back to that required to calculate light field and therefore photosynthesis.

I understand that from the point of view of an optical model, the absorption cross section is independent of the physiology and definition of phytoplankton. However, treating separately eq 6 and the variable rho does make the reading more difficult. At the risk of being pedantic, I think one should first present the dynamics in eq 37 and then illustrate the various terms. I'm particularly thinking about a student user who would like to learn the model and who may not have a full understanding of the underlying physiology.

To be helpful, I cross-reference chlorophyll synthesis as influencing intracellular concentrations in the optical model, and the role of absorption cross section in determining chlorophyll synthesis term in the BGC model. I have left the order of the manuscript the same.

P27L20 and L28 Table 6 is referenced before Table 5, which is the one listing the state variables whose dynamics is listed in Table 6.

Rearranged Table citations.

I would strongly suggest that a full list of the state variables is given at the very beginning when the biogeochemical model is introduced.

There is a full list in the Appendix. The philosophy of introducing one process at time is important to model framework, and I think also the scientific description (i.e. microalgae nutrient uptake does not depend directly on whether a seagrass community is there or not).

In P27L20, Tab 5 presents state and diagnostic variables, not equations.

Fixed.

P28L4 Please refer to the eq numbers and not just the table.

Fixed.

P30L2-3 This is an important information that should be given in the introduction, and briefly expanded upon to clarify how this models is positioned in the context of the existing theories and methodologies.

Yes, this has moved to the top of the microalgae model section. I haven't moved it into the introduction, because the introduction concentrates on the components of the model, rather than the equations.

Sec3.6. The mathematical formulation and equations are very little detailed for this component.

I have added a section referring to other saturating zooplankton grazing terms. Also, to help negotiate the equations I have cross-referenced some of the equations individually from the text.

Table 14 containing the dynamics is just referenced, and there is not a single equation describing the biomass evolution.

Eq. 76 in the original manuscript is biomass evolution.

It is also not clear from the beginning that the zooplankton variable has no variable stoichiometry (or internal reserves).

OK. Made this clear at the top of the zooplankton equation description.

P35L13 This is a coarse over-generalization which does not pay much attention to the model development occurred in the past 30 years. Models that use preference factors and a food matrix do not have this issue (Gentleman et al., 2003).

Yes. I looked up a range of papers, and my preconceptions were out of date. I have removed a few sentences. Thank you.

Sec 3.6.1 Grazing is actually not illustrated in this section.

Improved the model schematic in Section 2.2 to better resolve grazing interactions, showing microzooplankton eating small phytoplankton, large zooplankton eating large phytoplankton, *Trichodesmium* and microphytobenthos, and large zooplankton eating small zooplankton.

Table 15 is just mentioned but the specific terms not described. It is not much clear what is the foodweb accessed by zooplankton, apart from the first generic sentence at the beginning of sec. 3.2. How the fluxes between the state variables are actually computed is not clear.

Yes, I agree, the diets of the zooplankton we not mentioned in Sec 3.6. Fixed.

Table 12 What is the difference between variable mn in Table 7 and variable mB?

No difference. I have revised all use of m and its subscripts with the following change:

Both be $m_{B,N}$. Looked for other terms such as $m_P$ that I made $m_{B,N}$.

Sec. 3.8 I suggest the authors to make specific reference to the numbered equations and not to the Table containing them

I have done this in some cases.

(also check the typo at line 9 same page "zooplankton plankton")

Typo fixed.

Sec 3.9 I would see the section on non-grazing plankton mortality to be more pertinent to plankton dynamics, and less to zooplankton grazing. Is zooplankton mentioned at L6 because this is a loss term for all the plankton? It is rather confusing, and proper structuring would be helpful.

I have replaced 'phytoplankton mortality' with 'plankton mortality', which is what was meant.

Sec 3.10 I guess the author means gas exchange at the surface of the ocean here.

Title changes to: "Air-sea gas exchange"

P41L3 The variable is u10. This is the cubic function.

Thank you. Removed power of three.

P41L18 positive

Replaced.

P41L31 This seems a fragment with no connection with the previous paragraph.

Thank you. Fragment removed.

P42L22 Please clarify what "vertical order" means.

Replaced with "from top to bottom"

P43L18 Please refer where the diffusivity values have been taken. I could not find a table with the values.

Thank you. I refer back to the Table with the first use of diffusivity coefficients, and now also reference the original source, Li and Gregory (1974).

Eq113 and other. I would suggest the authors to use named constant for the stoichiometric coefficients, to allow identifying which conversion is actually being done.

In this manuscript I have made a deliberate decision to represent stoichiometric coefficients with combinations of integers. The reviewer's suggestion, to replace these with coefficients, would follow the approach generally used by biogeochemical modellers. However, I wonder if the use of integers is better. It is, after all, the way that it is done in physical chemistry. Also, there would be three stoichiometric coefficients for each autotroph, requiring unique subscripts, challenging the readability of the equations. The slight correction required for isotopes is small (Section 7.4.1 in original manuscript) and has been accounted for in the model itself.

Also, the authors should briefly illustrate the rationale for the use of this multiple minimum function, which I guess is linked to the maintenance of the constant stoichiometry in this functional group biomass.

OK. Added:

We have used the commonly applied multiple minimum function (Liebig, 1840), although it is noted that others use the multiple of limitation terms (Fasham, 1993). The microalgae model described above uses dynamical reserves to determine the growth rate. The growth approximated using dynamical reserves closer approximates a multiple minimum function than a multiple of minimum terms, so it was deemed more appropriate to use a multiple minimum function for macroalgae and seagrass for which internal reserves were not resolved.

P49L1 This is another example of the main problem I have with this manuscript (main comment 2). The same can be found at L29 w=in the case of coral processes. The authors seem to have cherry-picked what should be described and what should be left for the reader to scavenge through the literature. Please explain if there is an underlying rationale or a unified criterion.

Two papers exactly describe a subset of the equations in vB3p0. For coral processes:

Baird, M. E., M. Mongin, F. Rizwi, L. K. Bay, N. E. Cantin, M. Soja-Wozniak and J. Skerratt (2018) A mechanistic model of coral bleaching due to temperature-mediated light-driven reactive oxygen build-up in zooxanthellae. Ecol. Model 386: 20-37.

And for seagrass processes:

Baird, M. E., M. P. Adams, R. C. Babcock, K. Oubelkheir, M. Mongin, K. A. Wild-Allen, J. Skerratt, B. J. Robson, K. Petrou, P. J. Ralph, K. R. O'Brien, A. B. Carter, J. C. Jarvis, M. A. Rasheed (2016) A biophysical representation of seagrass growth for application in a complex shallow-water biogeochemical model *Ecol. Mod.* 325: 13-27.

The sections of these in the original submission failed this journal's plagiarism software. It was decided because these were the most detailed, and isolatable, sections of the model (and which are not implemented in all cases) they could be referenced to.

Some of the earlier papers (Baird et al., 2013 for example) include more fragmented sets of equations, but we consider it too difficult to bring them together in a coherent model description.

P49L6-7 These are microalgae, so I wonder why the authors decided not to use the same dynamics described earlier.

Good point. The dynamics are similar but we added a few extra processes to zooxanthellae, the most important being photoadaptation and photoinhibition, to resolve coral bleaching. While

photoadaptation and photoinhibition could easily be added to the other microalgae, we would need to add 5 new state variables for each microalgae, or about twenty 3D new state variables that require advecting by the transport model, a large computational cost. In contrast, because the zooxanthellae remain in the coral host, they are only 2D variables and don't require advection.

P57 How is M computed?

This manuscript does not describe the dynamics of inorganic sediments, which determine $M$. I point the reader to section 5.1, which summarises some aspects of the sediment model, and then points the reader onto published papers of the sediment model.

P58L5 With the use of coding style, the manuscript turns towards the user manual. This is the first time that code is used in the document. I am not against it if properly explained, but makes the manuscript less coherent (see main point 2).

Agreed. Wrong place for this. Removed.

P58L7-8 Make reference to Table 34 where the variables are listed.

Yes. Thank you.

Sec 5.2.1 I am a bit confused here, because light is not an environmental variable that controls the dynamics in the sediment model. What is the difference with Sec. 4.2.2?

The derivation of light capture by macroalgae (and also seagrass) considers them to be a 2D leaf. In contrast, the microalgae are considered a layer of spheres lying in the top layer of the sediment. Thus the equation 112 (in the original manuscript) is fundamentally different to equation 206.

I understand that the optical model is a major part of this manuscript, but then I would separate the biogeochemistry from the optics in two different manuscripts and make reference when needed instead of mixing the descriptions (see main point 1).

It is my hope to have one succinct manuscript containing both the optical and biogeochemical components. Also, the biogeochemical model cannot exist without the optics model (although the reverse is not true)

Equally important, the key innovation of the model derivation is the geometric origin of the equations. Often this geometric origin results in similar forms in both the optical and biogeochemical models, such as the use of absorption cross-sections in Eq. 6, 12 and 35.

Sec 5.3.1 and 5.3.2 There is no reference to equation numbers in these short sections.

Yes. Thank you. Fixed by referencing to Eqs. in Table 37 by number.

Sec. 6 I understand that section 6 collects all processes that are common to the pelagic, epibenthic and benthic processes. Therefore, I would expect to find here cross-references to the other sections where they have been described, as well as the inverse (when the dynamics are first illustrated, inform the reader that a certain term is considered a common process and found in Sec. 6).

OK. Good point. The organisation is intended such that processes are fully contained within a section. BUT, as the pointed out by the reviewer, I fail this criteria with respect to temperature dependence of ecological rates and preferential ammonia uptake that are cross process. Section 6 has now been split into two sections, one that refers to common processes, and a second that is common parameterisations. These are now extensively cross-referenced.

P68L9-11 I agree with this sentence, but this would deserve some more discussion. According to the Introduction, the model is designed to be generic, and the combination of physical and biogeochemical processes may lead to stiffness (this is cursory mentioned somewhere in the text or in the captions). The authors can refer to Butenschön et al., 2012 for an illustration of the problem. I would also ask the authors to clarify what do they mean with the "time step of the splitting"? Different time steps can be used for the various steps, but I am not familiar with a splitting time step. (Please explain what GBR1-4 mean in the table)

The Hundsdorfer and Verwer reference is a 460 page textbook on advection-diffusion-reaction equations. But I wasn't aware of the Butenschön et al. (2012) which is more focus to marine models, and it is included in the references. I have removed reference to GBR1/GBR4 as these are particular configurations, and just retain the values used in the results presented (from GBR4). "Time step of the splitting" is removed, and the text amended to be more explicit as to what was meant.

P68L13 The choice of a 5th order ODE should be justified, especially in the case of empirically-derived parameterizations. Please also say here which scheme is used and that it includes adaptive stepping. This information is given further below somewhere.

Our approach is to split the description of the solvers which is described in the text, with the actual choice that is given in the Table. Thus, I have added more in the text about the choices that include $1^{st} - 8^{th}$ order, and then give further details in the table. The justification now appears in the text. I also moved the information from below to here.

P69L2-4 This justification raises some concerns. The method is explicit, which would actually lead to instabilities that would require a time step shortening. Is this what the authors mean?

Yes, the adaptive approach reduces the time-step to avoid instabilities.

P69L16 Please explain the term "between". Is this implicit or explicit?

Replaced "between the physical and ecological" with "after the physical and before the ecological"

According to Table 42,it uses the same timestep as the ecology, but not clear what light environment is used for the ecology.

Added: "The light climate used for ecological timestep is that calculated at the start time of the ecological integration."

P70L9-10 This sentence is not clear. I understand that the manuscript is not about the coupling between the physics and the ecology, but this sentence would require more context. Please refer to the table where the number of levels in the described applications are listed.

Yes, this sentence has now moved into the Section 7.1, and some more context is given. Added text: "The solution of the ecological equations are independent for each vertical column, and depend only on the layers above through which the light has propagated."

Sec 8 I would argue that this manuscript does not offer any model evaluation beyond what has been already published. Is the assessment done here a technical check that version vB3p0 produces the same results as vB2p0 described in Skerratt et al. (2019)?

Yes, it is about ensuring that there are results from the published version.

Sec 9 and 10. I am not sure these sections help the manuscript concept, and they confirm my impression of the lack of coherence in the original idea of the presentation (point 1 above). I would

suggest the authors to reconsider their structure and to move some sections to the appendix or to on line material.

I would be happy to move these to supplementary material, but I think they are a requirement of the journal?

Sec 11 I have some concerns about the content of this discussion. I would expect Sec 11.1 in the introduction. I cannot really see any discussion here as I have indicated in my main point 3 above.

I think the reviewer would be happier if the discussion of other biogeochemical models (that I have expanded and now is in the discussion) be in the introduction, and the explanation of the geometric descriptions (which is in the introduction) be in the discussion.

I really want the reader to appreciate that I set out to produce a biogeochemical model that was based on geometric constraints that place physical limits on ecological processes. This has been the motivation from the start. I chose this approach on the 18 May 1996, the first day of my PhD. My Master supervisor, Marlin Atkinson at the University of Hawaii, was a pioneering advocate of physical limits on nutrient uptake by coral communities, and I was pig-headedly determined to try this for a biogeochemical model, whether my new PhD supervisor (Jacqui McGlade, University of Warwick) liked it or not (she gave me a free rein to pursue it). CSIRO has since supported me in this endeavour. My point is, the model equations would not be like they are had I not, in the subsequent 24 years, always asked myself when parameterising a process was there a geometric property I could take advantage off. Thus, the geometric approach is for me not a post-implementation talking point but the fundamental motivation of the work.

But I am not so pig-headed now, so if you want these sections reversed (introducing other models in the introduction and discussing EMS's unique features in the discussion) I will oblige.

Thank you.

---

## Author Response (AR2)

Dear Dr. Hargraves,

Thank you again for editing this manuscript. I have amended the manuscript, this time highlighting the edits of this round in red, which are almost exclusively in the Model Evaluation section. The blue highlights from the last round of edits are still highlighted. Below I reply to your comment in dark green, and my reply in black.

Yours sincerely,

Mark Baird

Good progress is being made, but this paper doesn't yet meet the standard for a GMD paper.

Thank you.

Model description papers need to contain sufficient evaluation to show that the model behaves as expected, and the evidence must be comprehensible to those outside of the group of people developing the model.

I have significantly improved the Model Evaluation section.

The part of the model equations that will be least familiar to marine BGC modellers will the equations for microalgae growth and chlorophyll synthesis. For many in the community these will also be the most important component, since while the benthic component of BGC models vary between groups due to different habitats, all marine BGC models include the plankton.

As a result, I have included a new section "11.1 Analysis of microalgae growth and pigment synthesis dynamics". The analysis considers the common question challenge of marine BGC modellers of capturing the changing physiological response of microalgae with depth and cell size. The analysis adds a new figure and 35 lines of text, showing the model conforms to the classic understanding of the phenomena and assists the reader to understand the unique aspects of the model.

In this case the evaluation is performed with reference to previous evaluations of previous model versions or parts of the same model. This is an acceptable approach. However, the evaluation needs to be presented in a form that is useful to readers outside of the EMS model development community. This means that much more description and explanation of the results is required in the main manuscript (as suggested by the reviewer), with essential plots being brought from the supplement into the main manuscript.

In addition to highlighting the chlorophyll dynamics as found in the earlier submission, I have now included both nutrient (phosphorus, nitrate and ammonium) and carbon chemistry dynamics. I have also expanded the explanation of the model evaluation to give better understanding of the skill metrics and explanation of the differing performance of elements of the model.

Thank you.

[revised manuscript text omitted]
 C1), 10 dissolved (Table C2), 20 microalgal (Table C3), 2 zooplankton (Table C4), 7 non-living inorganic particulate (Table C5), 7 non-living organic particulate (Table C6) and 7 epibenthic plant (Table C7) and 5 coral polyp (Table C8) and 4 reaction centre (Table C9) state variables. All state variables that exist in the water column layers have an equivalent in the sediment layers (and are specified by `<variable name>_sed`). The dissolved tracers are given as a concentration in the porewater, while the particulate tracers are given as a concentration per unit volume (see Sec. 10.3.2).

| Name | Symbol | Units | Description |
|---|---|---|---|
| Temperature (temp) | $T$ | °C | Water temperature |
| Salinity (salt) | $S$ | PSU | Water salinity |
| Sea level elevation (eta) | $\eta$ | m | Sea level elevation relative to mean sea level |
| Porosity (porosity) | $\phi$ | - | Fraction of the volume made up of water |
| Bottom shear stress (us-trcw_skin) | $\tau$ | N m$^{-2}$ | Shear stress due to currents and waves |
| Sand ripple height (PHYS-RIPH) | - | m | Physical dimension used for estimating bottom roughness due to ripples according to Grant and Madsen (1982) |
| Solar zenith (Zenith) | $\theta_{air}$ | radians | Solar zenith angle is the angle between the zenith and the centre of the Sun's disc, taking a value of zero with the sun directly above, and $\pi/2$ when at, or below, the horizon. |

**Table C1.** Long name (and variable name) in model output files, symbol and units used in this document, and a description of ecologically-relevant physical state and diagnostic variables.

| Name | Symbol | Units | Description |
|---|---|---|---|
| Total alkalinity (alk) | $A_T$ | mol kg$^{-1}$ | Concentration of ions that can be converted to uncharged species by a strong acid. The model assumes $A_T = [HCO_3^-] + [CO_3^{2-}]$, often referred to as carbonate alkalinity. Alkalinity and DIC together quantify the equilibrium state of the seawater carbon chemistry. |
| Nitrate (NO3) | $[NO_3^-]$ | mg N m$^{-3}$ | Concentration of nitrate. In the absence of nitrite $[NO_2^-]$ in the model, nitrate represents $[NO_3^-]$ + $[NO_2^-]$. |
| Ammonium (NH4) | $[NH_4^+]$ | mg N m$^{-3}$ | Concentration of dissolved ammonium. |

[revised manuscript text omitted]

**Appendix E. Assessment of eReefs biogeochemical simulation against observations**

**[Supplementary Material for Geoscientific Model Development: CSIRO**

**Environmental Modelling Suite (EMS): Scientific description of the optical**

**and biogeochemical models (vB3p0)]**

**Model version: gbr4_H2p0_**B3p0**_Chyd_Dcrt**

**Model run period: 1 Dec 2010 to 1 Nov 2018**

- **Includes comparison with version B2p0 where applicable**

(Version Tuesday, 26 March 2019)

Wednesday, 3 June 2020

For more details of Methods see:

Skerratt J.H., M. Mongin, K. A. Wild-Allen, M. E. Baird, B. J. Robson, B. Schaffelke, M. Soja-Wozniak, N Margvelashvili, C. H. Davies, A. J. Richardson, A. D. L. Steven (2019) Simulated nutrient and plankton dynamics in the Great Barrier Reef (2011-2016). J. Mar. Sys. 192, 51-74.

**Document versions**

*Thursday, 3 January 2019 version*

- Includes observation updates to MMP Turbidity and MMP chlorophyll mooring obs to November 2018: p111 to 125
- Includes the new MMP sites which have decreased the metrics for both Turbidity and Fluorescence. The metrics are better if we leave summer of 2011 in.
- Simulated turbidity has zeros (night-time) removed in the model run. p 118 to 125.
- Simulated Fluorescence is not as good as simulated Chl *a* against MMP mooring obs however obs are modified fluorescence based on Chl *a*
- Turbidity is presented at full extent of NTU and again with NTU under 20 (p119 and 125)
- The QC of the new set of MMP data remains excellent but doesn't appear as stringently QC'd as in the past with blanks and some unrealistic data.

*Friday, 4 January 2019 version*

- Scatter plots of fluorescence against Chl *a* for all MMP moorings and combined scatterplot at end

*Tuesday, 19 February 2019 version*

- Added parameter file for H3 version

*Wednesday, 20 February 2019 version*

- Added satellite photos depth of MMP and LTR sites and glossary

*Tuesday, 26 March 2019 version*

- Added correct NRS nutrient metrics and graphs with extended observational time series and NRS alkalinity extension of observed dates and inclusion of North Stradbroke island (GBRNSI)

**Contents**

**1. Map: River and catchments in eReef model**

[Figure]

●Rivers and catchment model with hydro flow catchment loads B2p0 and B3p0
● Extra rivers in B3p0 where catchment in as point source loads
● Rivers in hydrodynamic model, some without flow, no catchment model data.

*Figure 1  Map of Queensland rivers included in eReef model versions B2p0 and B3p0.  Includes extra rivers for B3p0 in light blue*

[Figure]

*Figure 2 Map of observational sites in this report (black and pink), rivers (blue) and major towns (Green)*

**3. Map Wakmatha transect for Carbon Chemistry**

Figure shows Wakmatha transect and temp and salinity comparison with GBR1 (see page 180 Wakmatha transect line for Carbon chemistry assessment of Wakmatha transect line)

[Figure]

**4. eReefs Biogeochemical Model schematic**

[Figure]

*Figure 3.The eReefs modelling system, showing the linkages between hydrodynamic, wave, sediment and the optical and biogeochemical models, as well as the individual linkages within the biogeochemical model. The optically-active components are identified with orange font.*

**5. Model skill metrics description**

To evaluate model skill, we consider; bias, the root mean square (RMS) error, the mean absolute error (MAE). and the modified Willmott index or 'd2' (Willmott et al., 1985). The Willmott index uses the sum of absolute values.

Model bias assesses whether the simulated variables are under- or over-predicting observed values. The RMS error is a measure of the absolute magnitude of the "error"/square deviation averaged over the time-series. An RMS or MAE of 0 indicates a perfect fit.

The Willmott index of agreement is designed to quantify errors that are unevenly distributed in time or space and reduce the influence of errors during periods of large observed mean or variance. The Willmott index is the ratio of the mean absolute error and the mean absolute deviation about the observed mean and varies between 0 and 1. A value of 1 indicates a perfect match (x = y), and 0 indicates no agreement.

$$\text{Willmott} = 1 - [\, \textstyle\sum | x - y | \,) / [\, \textstyle\sum | x - \bar{y} |) + (| y - \bar{y} |)]$$

where x and y are vectors or arrays of time series data (x =observed, y = modelled).

A Willmott index above 0.7 is regularly obtained for high resolution models with high spatial and temporal observations for physical parameters such as salinity and temperature. In most cases for the eReefs model the salinity and temperature index was ≥ 0.8 when compared with observations (Appendix 1 of Herzfeld et al., 2016).

**6. Abbreviations**

| AIMS | Australian Institute of Marine Science |
|---|---|
| AODN | Australian Ocean Data Network |
| B2p0 | B2p0: biogeochemical model version 2.0 |
| B3p0 | B3p0: biogeochemical model version 3.0 |
| CDOM | colour dissolved organic matter |
| Chl a | chlorophyll a |
| CTD | Conductivity Temperature Depth profiler |
| d2 | Statistical metric, aka Willmott index ( see page 27) |
| DIN | dissolved inorganic nitrogen |
| DIN | Dissolved inorganic nitrogen (NH3 plus NOx) |
| DIP | dissolved inorganic phosphorus |
| DOC | dissolved organic carbon |
| DON | dissolved organic nitrogen |
| DOP | dissolved organic phosphorus |
| ENSO | El Niño-Southern Oscillation |
| GBR | Great Barrier Reef |
| gbr4_H2p0_B3p0_Cb | gbr4 : model grid with approximate 4 km grid resolution, H2p0: hydrodynamic model version 2.0, B3p0: biogeochemical model version 3.0, Cb: catchment model baseline version using empirical SOURCE Catchments |
| GBRMP | Great Barrier Reef Marine Park |
| GBRMPA | Great Barrier Reef Marine Park Authority |
| GBRWHA | Great Barrier Reef World Heritage Area |
| IMOS | Integrated Marine and Observing System |
| Kd(PAR) | light attenuation coefficient |
| LTM | AIMS long term monitoring site |
| mae | mean absolute error |
| mape | mean absolute percentage error |
| MMP | AIMS Marine Monitoring Program |
| MODIS | Moderate Resolution Imaging Spectroradiometer |
| NH3 | ammonia |
| NOx | nitrate plus nitrite |
| NRS | IMOS National reference station within the model grid these are Yongala (GBRYON) and North Stradbroke Island (GBRNSI) |
| NSI | North Stradbroke Island |
| NTU | Nephelometric Turbidity Unit |
| PON | particulate organic nitrogen |
| POP | particulate organic phosphorus |
| QA/QC | quality assurance/quality control |
| rms | root mean square |
| secchi | measurement of water transparency (depth in m) |
| TSS | total suspended solids |
| Willmott | statistical metric (see page 27) |

**7. Parameter tables for gbr4_H2p0_B3p0_Cb**

The following 4 pages give the parameters used in the model gbr4_H2p0_B3p0_Cb.

| Parameter description | Symbol | Units | Value | Reference |
|---|---|---|---|---|
| **Phytoplankton** | | | | |
| Chl-specific scattering coefficient. for microalgae | bphy | $m^{-1}$ (mg Chl a $m^{-3}$)$^{-1}$ | 0.2 | Typical microalgae value, Kirk (1994) |
| Natural (linear) mortality rate, large phytoplankton | PhyL_mL | $d^{-1}$ | 0.1 | Not attributed |
| Natural (linear) mortality rate in sediment, large phytoplankton | PhyL_mL_sed | $d^{-1}$ | 10 | Not attributed |
| Natural (linear) mortality rate, small phytoplankton | PhyS_mL | $d^{-1}$ | 0.1 | Not attributed |
| Natural (linear) mortality rate in sediment, small phytoplankton | PhyS_mL_sed | $d^{-1}$ | 1 | Not attributed |
| Respiration as a fraction of umax | Plank_resp | none | 0.025 | Not attributed |
| Radius of the large phytoplankton cells | PLrad | m | 0.000004 | Not attributed |
| Maximum growth rate of PL at Tref | PLumax | $d^{-1}$ | 1.4 | CSIRO Parameter Library |
| Ratio of xanthophyll to chl a of PL | PLxan2chl | mg $mg^{-1}$ | 0.81 | CSIRO Parameter Library |
| Radius of the small phytoplankton cells | PSrad | m | 0.000001 | Not attributed |
| Maximum growth rate of PS at Tref | PSumax | $d^{-1}$ | 1.6 | CSIRO Parameter Library |
| Ratio of xanthophyll to chl a of PS | PSxan2chl | mg $mg^{-1}$ | 0.51 | CSIRO Parameter Library |
| | | | | |
| ***Trichodesmium*** | | | | |
| DIN conc below which *Trichodesmium* N fixes | DINcrit | mg N $m^{-3}$ | 10 | Lower end of Robson et al., (2013) 4-20 mg N $m^{-3}$ |
| Maximum density of *Trichodesmium* | p_max | kg $m^{-3}$ | 1050 | Not attributed |
| Minimum density of *Trichodesmium* | p_min | kg $m^{-3}$ | 900 | Not attributed |
| Radius of *Trichodesmium* colonies | Tricho_colrad | m | 0.000005 | Not attributed |
| Critical *Trichodesmium* above which quadratic mortality applies | Tricho_crit | mg N $m^{-3}$ | 0.0002 | Not used in code |
| Linear mortality for *Trichodesmium* in sediment | Tricho_mL | $d^{-1}$ | 0.1 | Not attributed |
| Quadratic mortality for *Trichodesmium* due to phages in water column | Tricho_mQ | $d^{-1}$ (mg N $m^{-3}$)$^{-1}$ | 0.1 | At steady-state, indep. of temp, Tricho_N ~ Tricho_umax / Tricho_mQ = 0.27 / 0.405 = 0.7 mg N $m^{-3}$ ~ 0.1 mg Chl $m^{-3}$ |
| *Trichodesmium* grazing preference | Tricho_pref | none | 0 | Not attributed |
| Radius of *Trichodesmium* colonies | Tricho_rad | m | 0.000005 | Not attributed |
| Sherwood number for the *Trichodesmium* dimensionless | Tricho_Sh | none | 1 | Not attributed |
| Maximum growth rate of *Trichodesmium* at Tref | Tricho_umax | $d^{-1}$ | 0.2 | Robson et al., 2013 + Parameter library |
| Ratio of xanthophyll to chl a of *Trichodesmium* | Trichoxan2chl | mg $mg^{-1}$ | 0.5 | Subramaniam et al. 1999. LO 44:618-627 |
| | | | | |
| **Microphytobenthos** | | | | |
| Respiration as a fraction of umax | Benth_resp | none | 0.025 | Not attributed |
| Radius of the MPB cells | MBrad | m | 0.00001 | Not attributed |
| Maximum growth rate of MB at Tref | MBumax | $d^{-1}$ | 0.839 | CSIRO Parameter Library |
| Ratio of xanthophyll to chl a of MPB | MBxan2chl | mg $mg^{-1}$ | 0.81 | Not attributed |
| Natural (quadratic) mortality rate, microphytobenthos, applied in sediment | MPB_mQ | $d^{-1}$ (mg N $m^{-3}$)$^{-1}$ | 0.0001 | SS argument |
| **Parameter description** | **Symbol** | **Units** | **Value** | **Reference** |

**Zooplankton**

| Parameter description | Symbol | Units | Value | Reference |
|---|---|---|---|---|
| Growth efficiency, large zooplankton | ZL_E | none | 0.426 | CSIRO Parameter Library, [0.341 (0.017900) Baird and Suthers, 2007 from Hansen et al (1997) LO 42: 687-704] |
| Fraction of growth inefficiency lost to detritus, large zooplankton | ZL_FDG | none | 0.5 | Not attributed |
| Fraction of mortality lost to detritus, large zooplankton | ZL_FDM | none | 1 | Not attributed |
| Natural (quadratic) mortality rate, large zooplankton | ZL_mQ | $d^{-1}$ $(mg\ N\ m^{-3})^{-1}$ | 0.012 | Not attributed |
| Diel vertical migration rate of ZL | ZLdvmrate | $m\ d^{-1}$ | 0 | Not attributed |
| Grazing technique of large zooplankton | ZLmeth | none | rect | Not attributed |
| Light at which the | ZLpar | $mol\ photons\ m^{-2}\ s^{-1}$ | 1.00E-12 | Not attributed |
| Radius of the large zooplankton cells | ZLrad | m | 0.00032 | Not attributed |
| Swimming velocity for large zooplankton | ZLswim | $m\ s^{-1}$ | 0.003 | Not attributed |
| Maximum growth rate of ZL at Tref | ZLumax | $d^{-1}$ | 1.33 | Not attributed |
| Growth efficiency, small zooplankton | ZS_E | none | 0.462 | CSIRO Parameter Library [0.3080000 (0.026600) Baird and Suthers, 2007 from Hansen et al (1997) LO 42: 687-704] |
| Fraction of growth inefficiency lost to detritus, small zooplankton | ZS_FDG | none | 0.5 | Not attributed |
| Fraction of mortality lost to detritus, small zooplankton | ZS_FDM | none | 1 | Not attributed |
| Natural (quadratic) mortality rate, small zooplankton | ZS_mQ | $d^{-1}$ $(mg\ N\ m^{-3})^{-1}$ | 0.02 | Not attributed |
| Grazing technique of small zooplankton | ZSmeth | none | rect | Not attributed |
| Radius of the small zooplankton cells | ZSrad | m | 0.000005 | Not attributed |
| Swimming velocity for small zooplankton | ZSswim | $m\ s^{-1}$ | 0.0002 | Not attributed |
| Maximum growth rate of ZS at Tref | ZSumax | $d^{-1}$ | 4 | Not attributed |

**Coral**

| Parameter description | Symbol | Units | Value | Reference |
|---|---|---|---|---|
| Quadratic mortality rate of coral polyp | CHmort | $(g\ N\ m^{-3})^{-1}\ d^{-1}$ | 0.01 | Not attributed |
| Nitrogen-specific area of coral polyp density | CHpolypden | $m2\ g\ N^{-1}$ | 2 | Not attributed |
| Fraction of Host death translocated. | CHremin | - | 0.5 | Not attributed |
| Max. growth rate of Coral at Tref | CHumax | $d^{-1}$ | 0.05 | Not attributed |
| Linear mortality rate of Zooxanthellae | CSmort | $d^{-1}$ | 0.04 | Not attributed |
| Radius of the Zooxanthellae | CSrad | m | 0.000005 | Not attributed |
| Fraction of Zooxanthellae growth to Host. | CStoCHfrac | - | 0.9 | Gustafsson et al. (2013) Ecol. Mod. 250: 183-194 |
| Max. growth rate of Zooxanthellae at Tref | CSumax | $d^{-1}$ | 0.4 | Not attributed |
| Maximum daytime net coral calcification | k_day_coral | $mmol\ C\ m^{-2}\ s^{-1}$ | 0.0132 | Anthony et al. (2013), Biogeosciences 10:4897-4909, Fig 5A: 50, 50, 35 55 mmol $m^{-2}\ h^{-1}$ for Acropora aspera n=4 |
| Grid scale to reef scale ratio | CHarea | $m\ m^{-1}$ | 0.1 | Not attributed |
| Maximum night time net coral calcification | k_night_coral | $mmol\ C\ m^{-2}\ s^{-1}$ | 0.0069 | Anthony et al. (2013), Biogeosciences 10:4897-4909, Fig 5A: 20, 30, 20, 30 mmol $m^{-2}\ h^{-1}$ for Acropora aspera n=4 |
| Rate coefficient for plankton uptake by corals | Splank | $m\ d^{-1}$ | 3 | Ribes (2003), PARAMETER library analysis;Ribes and Atkinson (2007) Coral Reefs 26: 413-421 |

| Parameter description | Symbol | Units | Value | Reference |
|---|---|---|---|---|

**Seagrass and Macroalgae**

| Parameter description | Symbol | Units | Value | Reference |
|---|---|---|---|---|
| Half-saturation of SG N uptake in SED | SG_KN | $mg\ N\ m^{-3}$ | 420 | Lee and Dunton (1999) 1204-1215. Table 3 Zostera |

| Parameter description | Symbol | Units | Value | Reference |
|---|---|---|---|---|
| Half-saturation of SG P uptake in SED | SG_KP | mg P m$^{-3}$ | 96 | Gras et al. (2003) Aquatic Botany 76:299$^{-3}$15. Thalassia testudinum. |
| Natural (linear) mortality rate, seagrass | SG_mL | d$^{-1}$ | 0.03 | Fourquean et al.( 2003) Chem. Ecol. 19: 373$^{-3}$90.Thalassia leaves with one component decay |
| Critical shear stress for SG loss | SG_tau_critical | N m^{-2} | 1 | NESP project |
| Time-scale for critical shear stress for SG loss | SG_tau_efold | s | 43200 | NESP project |
| Half-saturation of SGD N uptake in SED | SGD_KN | mg N m$^{-3}$ | 420 | Not attributed |
| Half-saturation of SGD P uptake in SED | SGD_KP | mg P m$^{-3}$ | 96 | Not attributed |
| Natural (linear) mortality rate, aboveground SGD | SGD_mL | d$^{-1}$ | 0.06 | NESP project |
| Critical shear stress for SGD loss | SGD_tau_critical | N m$^{-2}$ | 1 | NESP project |
| Time-scale for critical shear stress for SGD loss | SGD_tau_efold | s | 43200 | NESP project |
| Fraction (target) of SGD biomass below-ground | SGDfrac | - | 0.25 | Duarte (1999) Aquatic Biol. 65: 159-174, Halophila ovalis. |
| Nitrogen-specific leaf area of SGD | SGDleafden | m$^2$ g N$^{-1}$ | 1.9 | Halophila ovalis: leaf dimensions from Vermaat et al. (1995) |
| Compensation irradiance for Halophila | SGDmlr | mol m$^{-2}$ | 1.5 | NESP project |
| Sine of nadir Deep Segrass canopy bending angle | SGDorient | - | 1 | No source |
| Natural (linear) mortality rate, belowground SGD | SGDROOT_mL | d$^{-1}$ | 0.004 | NESP project |
| Maximum depth for Halophila roots | SGDrootdepth | m | -0.05 | NESP project |
| Halophila seed biomass as fraction of 63 % cover | SGDseedfrac | - | 0.01 | Not attributed |
| Time scale for seagrass translocation | SGDtransrate | d$^{-1}$ | 0.0333 | Loosely based on Zostera marine Kaldy et al., 2013 MEPS 487:27-39 |
| Maximum growth rate of SGD at Tref | SGDumax | d$^{-1}$ | 0.4 | x2 nighttime, x2 for roots. |
| Fraction (target) of SG biomass below-ground | SGfrac | - | 0.75 | Babcock (2015) Zostera capricornii |
| Half-saturation of SGH N uptake in SED | SGH_KN | mg N m$^{-3}$ | 420 | Not attributed |
| Half-saturation of SGH P uptake in SED | SGH_KP | mg P m$^{-3}$ | 96 | Not attributed |
| Natural (linear) mortality rate, seagrassH | SGH_mL | d$^{-1}$ | 0.06 | Fourquean et al.(2003) Chem. Ecol. 19: 373$^{-3}$90.Thalassia leaves with one component decay |
| Critical shear stress for SGH loss | SGH_tau_critical | N m$^{-2}$ | 1 | NESP project |
| Time-scale for critical shear stress for SGH loss | SGH_tau_efold | s | 43200 | NESP project |
| Fraction (target) of SGH biomass below-ground | SGHfrac | - | 0.5 | Babcock 2015, Halophila ovalis |
| Nitrogen-specific area of seagrass leaf | SGHleafden | m2 g N$^{-1}$ | 1.9 | Halophila ovalis: leaf dimensions from Vermaat et al. (1995) |
| Compensation irradiance for SG | SGHmlr | mol m$^{-2}$ | 2 | Not attributed |
| Sine of nadir Halophila canopy bending angle | SGHorient | - | 1 | No source |
| Natural (linear) mortality rate, seagrassH | SGHROOT_mL | d$^{-1}$ | 0.004 | Fourquean et al. (2003) Chem. Ecol. 19: 373-390. Thalassia roots with one component decay |
| Maximum depth for Halophila roots | SGHrootdepth | m | -0.08 | Roberts (1993) Aust. J. Mar. Fresh. Res. 44:85-100. |
| Halophila seed biomass as fraction of 63 % cover | SGHseedfrac | - | 0.01 | Not attributed |
| Time scale for seagrass translocation | SGHtransrate | d$^{-1}$ | 0.0333 | Loosely based on Zostera marine Kaldy et al., 2013 MEPS 487:27-39 |
| Maximum growth rate of SGH at Tref | SGHumax | d$^{-1}$ | 0.4 | x2 night-time, x2 for roots. |
| Nitrogen-specific area of seagrass leaf | SGleafden | m2 g N$^{-1}$ | 1.5 | Zostera capricornia: leaf dimensions Kemp et al (1987) Mar Ecol. Prog. Ser. 41:79-86. |
| Compensation irradiance for SG | SGmlr | mol m$^{-2}$ | 4.5 | Not attributed |
| SGorient | SGorient | | 0.5 | Not attributed |
| Natural (linear) mortality rate, seagrass | SGROOT_mL | d$^{-1}$ | 0.004 | Fourquean et al. (2003) Chem. Ecol. 19: 373-390. Thalassia roots with one component decay |
| Maximum depth for Zostera roots | SGrootdepth | m | -0.15 | Roberts (1993) Aust. J. Mar. Fresh. Res. 44:85-100. |
| Seagrass seed biomass as fraction of 63 % cover | SGseedfrac | - | 0.01 | No source |
| Time scale for seagrass translocation | SGtransrate | d$^{-1}$ | 0.0333 | Loosely based on Zostera marine Kaldy et al., 2013 MEPS 487:27-39 |
| Maximum growth rate of SG at Tref | SGumax | d$^{-1}$ | 0.4 | x2 nighttime, x2 for roots. |
| Natural (linear) mortality rate, macroalgae | MA_mL | d$^{-1}$ | 0.01 | Not attributed |
| Nitrogen-specific area of macroalgae leaf | MAleafden | m$^2$ g N$^{-1}$ | 1 | Not attributed |
| Maximum growth rate of MA at Tref | MAumax | d$^{-1}$ | 1 | Not attributed |

| Parameter description | Symbol | Units | Value | Reference |
|---|---|---|---|---|

**Biogeochemistry**

| | | | | |
|---|---|---|---|---|
| Reference temperature | Tref | Deg C | 20 | CSIRO Parameter Library |
| Temperature coefficient for rate parameters | Q10 | none | 2 | CSIRO Parameter Library |
| Nominal rate of TKE dissipation in water column | TKEeps | $m^2 s^{-3}$ | 0.000001 | Not attributed |
| Atmospheric CO2 | xco2_in_air_dum | ppmv | 396.48 | Mean 2013 at Mauna Loa: htttrp://co2now.org/current-co2/co2-now/ |
| Wavelengths of light | Light_lambda | nm | Various* | Approx. 20 nm resolution with 10 nm about 440 nm. PAR (400-700) is integral of bands 2-22 (290 310 330 350 370 390 410 430 440 450 470 490 510 530 550 570 590 610 630 650 670 690 710 800)* |
| Nominal N:Chl a ratio in phytoplankton by weight | NtoCHL | g N (g Chl a)$^{-1}$ | 7 | Represents a C:Chl ratio of 39.25, Baird et al. (2013) Limnol. Oceanogr. 58: 1215-1226. |
| Concentration of dissolved N2 | N2 | mg N m$^{-3}$ | 2000 | Robson et al. (2013) |
| Fraction of labile detritus converted to refractory detritus | F_LD_RD | none | 0.19 | Not attributed |
| Fraction of labile detritus converted to dissolved organic matter | F_LD_DOM | none | 0.1 | Not attributed |
| fraction of refractory detritus that breaks down to DOM | F_RD_DOM | none | 0.05 | Not attributed |
| Breakdown rate of labile detritus at 106:16:1 | r_DetPL | d$^{-1}$ | 0.04 | Not attributed |
| Breakdown rate of labile detritus at 550:30:1 | r_DetBL | d$^{-1}$ | 0.001 | Not attributed |
| Breakdown rate of refractory detritus | r_RD | d$^{-1}$ | 0.001 | Not attributed |
| Breakdown rate of dissolved organic matter | r_DOM | d$^{-1}$ | 0.0001 | Achieves approx. SS of global ocean at 20 C. |
| Oxygen half-saturation for aerobic respiration | KO_aer | mg O m$^{-3}$ | 256 | Not attributed |
| Maximal nitrification rate in water column | r_nit_wc | d$^{-1}$ | 0.1 | Not attributed |
| Maximal nitrification rate in water sediment | r_nit_sed | d$^{-1}$ | 20 | Not attributed |
| Oxygen half-saturation for nitrification | KO_nit | mg O m$^{-3}$ | 500 | Not attributed |
| Rate at which P reaches adsorbed/desorbed equilibrium | Pads_r | d$^{-1}$ | 0.04 | Not attributed |
| Freundlich Isothermic Const P adsorption to TSS in water column | Pads_Kwc | mg P kg TSS$^{-1}$ | 30 | Not attributed |
| Freundlich Isothermic Const P adsorption to TSS in sediment | Pads_Ksed | mg P kg TSS$^{-1}$ | 74 | Not attributed |
| Oxygen half-saturation for P adsorption | Pads_KO | mg O m$^{-3}$ | 2000 | Not attributed |
| Exponent for Freundlich Isotherm | Pads_exp | none | 1 | Not attributed |
| Maximum denitrification rate | r_den | d$^{-1}$ | 0.8 | Not attributed |
| Oxygen half-inhibition of denitrification rate | KO_den | mg O m$^{-3}$ | 10000 | Not attributed |
| Rate of conversion of PIP to immobilised PIP | r_immob_PIP | d$^{-1}$ | 0.0012 | Not attributed |
| Sediment-water diffusion coefficient | EpiDiffCoeff | $m^2 s^{-1}$ | 3.00E-07 | Not attributed |
| Thickness of diffusive layer | EpiDiffDz | m | 0.0065 | Not attributed |
| age tracer growth rate per day | ageing_decay | d$^{-1}$ | 1 | Not attributed |
| age tracer decay rate per day outside source | anti_ageing_decay | d$^{-1}$ | 0.1 | Not attributed |
| net dissolution rate of sediment without coral | dissCaCO3_sed | mmol C m$^{-2}$ s$^{-1}$ | 0.001 | Anthony et al. (2013), Biogeosciences 10:4897-4909, Fig 5E: -1 2 3 6 mmol m$^{-2}$ h$^{-1}$ |
| DOC-specific absorption of CDOM at 443 nm | acdom443star | $m^2$ mg C$^{-1}$ | 0.00013 | Not attributed |
| Minimum carbon to chlorophyll ratio | C2Chlmin | wt/wt | 20 | Not attributed |
| swr scaling factor | SWRscale | none | 1 | Not attributed |
| Bleaching ROS threshold | ROSthreshold | - | 5.00E-04 | Not attributed |
| increased breakdown fraction DetrP to DOP | r_RD_NtoP | - | 2 | Not attributed |
| increased breakdown fraction DOMP to DIP | r_DOM_NtoP | - | 1.5 | Not attributed |

**8. Site and model grid depth of the MMP and NRS sites**

| MMP and NRS Sites | GBR4 grid depth (m) | Site depth (m) |
|---|---|---|
| Barren Island | 24 | 15 - 19 |
| Daydream Island | 17 | 23 - 25 |
| Double Cone Island | 17 | 23 - 31 |
| Dunk Island | 9 | 9 - 10 |
| Fitzroy Island | 27 | 15 - 17 |
| Geoffrey Bay | 10 | 9 - 10 |
| High Island | 18 | 22 - 25 |
| Humpy Island | 13 | 12 - 19 |
| North Stradbroke Island (NSI) | 66 | 65 - 67 |
| Pandora Island | 17 | 13 - 14 |
| Pelican Island | 4 | 9 - 10 |
| Pelorus Island | 25 | 25 - 31 |
| Pine Island | 18 | 20 - 25 |
| Russell Island | 20 | 22 - 24 |
| Snapper Island | 22 | 8 - 11 |
| Yongala | 29 | 26 - 27 |

**9. Site and depths for additional triannual sites or depths**

| AIMS additional Triannual Water Quality sites | Sampling Depths (m) | | |
|---|---|---|---|
| Cape Tribulation | 10 | | |
| Snapper Island | 10 | | |
| Port Douglas | 0 | 15 | |
| Double Island | 0 | 18 | |
| Green Island | 0 | 18 | 36 |
| Yorkeys Knob | 0 | 8 | |
| Fairlead Buoy | 0 | | |
| Fitzroy Reef | 0 | 15 | |
| High Island | 0 | 10 | 20 |
| Russell Island | 0 | 10 | 20 |
| Dunk Island | 5 | | |
| Pelorus Island | 0 | 14 | 28 |
| Double Cone Island | 10 | 23 | |
| Daydream Island | 10 | 23 | |
| Pine Island | 0 | 20 | |
| Barren Island | 10 | | |
| Humpy Island | 0 | 10 | |

**10. Simulated Chl *a* assessment against AIMS Long Term Monitoring**

[Figure]

*Figure 4 Metrics for Long Term Monitoring sites Chlorophyll assessment against observations for model version 3p0 and 2p0 d2 = Willmott index see Statistical metric page 27.mae:mean absolute error, rms root mean square*

[Figure]

Pelican_5m  3.0 d2:0.49, mape:57.4, rms:0.4306
bias:−0.2007, r:0.2329, obsmean:0.6437
Pelican_5m 2.0 d2:0.54, mape:56.6, rms:0.4907
bias:−0.3625, r:0.4393, obsmean:0.6437

[Figure]

Humpy873_10m  3.0 d2:0.63, mape:86.7, rms:0.3797
bias:0.1801, r:0.5769, obsmean:0.3725
Humpy873_10m 2.0 d2:0.70, mape:55.6, rms:0.3449
bias:0.0125, r:0.6545, obsmean:0.3725

[Figure]

Humpy873_0m  3.0 d2:0.70, mape:58.6, rms:0.2604
bias:0.1022, r:0.6019, obsmean:0.3552
Humpy873_0m 2.0 d2:0.77, mape:49.0, rms:0.2437
bias:−0.0469, r:0.7150, obsmean:0.3552

[Figure]

[Figure]

[Figure]

[Figure]

**Daydream_23m 3.0 d2:0.37, mape:40.5, rms:0.4067
bias:−0.0757, r:0.2319, obsmean:0.6542
Daydream_23m 2.0 d2:0.42, mape:39.6, rms:0.3114
bias:−0.1758, r:0.1886, obsmean:0.6542**

[Figure]

**Daydream330_10m 3.0 d2:0.55, mape:35.5, rms:0.3463
bias:−0.0645, r:0.4642, obsmean:0.5693
Daydream330_10m 2.0 d2:0.56, mape:31.6, rms:0.2735
bias:−0.1611, r:0.3283, obsmean:0.5693**

[Figure]

**DoubleCone_23m 3.0 d2:0.40, mape:73.9, rms:0.6570
bias:−0.1586, r:0.1348, obsmean:0.6454
DoubleCone_23m 2.0 d2:0.66, mape:58.4, rms:0.3045
bias:−0.0987, r:0.4722, obsmean:0.5267**

[Figure]

[Figure]

[Figure]

[Figure]

[Figure]

[Figure]

[Figure]

[Figure]

[Figure]

[Figure]

[Figure]

[Figure]

[Figure]

[Figure]

[Figure]

[Figure]

[Figure]

[Figure]

[Figure]

Green830_36m 3.0 d2:0.77, mape:42.6, rms:0.1302
bias:0.0132, r:0.5727, obsmean:0.3083
Green830_36m 2.0 d2:0.68, mape:59.7, rms:0.1901
bias:0.0289, r:0.4754, obsmean:0.3009

[Figure]

Green830_18m 3.0 d2:0.86, mape:42.7, rms:0.1654
bias:0.0400, r:0.7804, obsmean:0.3134
Green830_18m 2.0 d2:0.75, mape:47.9, rms:0.2277
bias:-0.0015, r:0.6213, obsmean:0.3134

[Figure]

Green830_0m 3.0 d2:0.56, mape:34.2, rms:0.2756
bias:-0.0867, r:0.5624, obsmean:0.3272
Green830_0m 2.0 d2:0.49, mape:39.2, rms:0.3629
bias:-0.1369, r:0.3158, obsmean:0.3434

[Figure]

[Figure]

[Figure]

[Figure]

[Figure]

[Figure]

**11. Simulated Secchi depth assessment against AIMS Long Term Monitoring**

[Figure]

*Figure 5  Metrics for Long Term Monitoring sites Secchi depth assessment against observations for model version 3p0, d2 = Willmott index see Statistical metric page 27.mae:mean absolute error, rms root mean square*

[Figure]

[Figure]

[Figure]

[Figure]

[Figure]

[Figure]

[Figure]

[Figure]

[Figure]

[Figure]

**Pine 3.0 d2:0.52, mape:125.2, rms:5.8227**
**bias:4.8346, r:0.7486, obsmean:4.3583**

[Figure]

**Daydream 3.0 d2:0.41, mape:177.0, rms:6.6422**
**bias:6.0742, r:0.6805, obsmean:4.3462**

[Figure]

**DoubleCone 3.0 d2:0.31, mape:150.6, rms:7.3992**
**bias:6.8722, r:0.3042, obsmean:5.6250**

[Figure]

[Figure]

[Figure]

[Figure]

[Figure]

[Figure]

[Figure]

[Figure]

[Figure]

*Figure 6 Scatter plot of observed Secchi for long Term Monitoring sites and NRS sites (Yongala and North Stradbroke) assessment against simulated Secchi for model version 3p0*

**12. Simulated DIP assessment against AIMS Long Term Monitoring**

[Figure]

*Figure 7 Metrics for Long Term Monitoring sites DIP assessment against observations for model version 3p0 and 2p0 d2 = Willmott index see Statistical metric page 27.mae:mean absolute error, rms root mean square*

[Figure]

[Figure]

[Figure]

**Barren411_10m  3.0 d2:0.58, mape:67.9, rms:2.3439**
**bias:0.5380, r:0.6883, obsmean:2.1001**
**Barren411_10m 2.0 d2:0.61, mape:67.6, rms:1.4913**
**bias:−1.2843, r:0.6697, obsmean:2.1001**

[Figure]

**Pine329_20m  3.0 d2:0.66, mape:71.0, rms:2.6551**
**bias:−1.9380, r:0.6373, obsmean:3.5062**
**Pine329_20m 2.0 d2:0.51, mape:81.7, rms:3.2758**
**bias:−2.9881, r:0.6426, obsmean:3.7783**

[Figure]

**Pine329_0m  3.0 d2:0.63, mape:71.6, rms:2.6900**
**bias:−1.9750, r:0.6078, obsmean:3.4935**
**Pine329_0m 2.0 d2:0.48, mape:80.9, rms:3.2962**
**bias:−3.0075, r:0.5815, obsmean:3.7724**

[Figure]

Daydream_23m 3.0 d2:0.71, mape:49.9, rms:2.1620
bias:−1.2492, r:0.6306, obsmean:3.9418
Daydream_23m 2.0 d2:0.49, mape:71.8, rms:3.0820
bias:−2.8184, r:0.6410, obsmean:3.9418

[Figure]

Daydream330_10m 3.0 d2:0.56, mape:52.1, rms:2.5075
bias:−1.5843, r:0.4367, obsmean:4.0124
Daydream330_10m 2.0 d2:0.41, mape:78.0, rms:3.3453
bias:−3.1183, r:0.4114, obsmean:4.0124

[Figure]

DoubleCone334_10m 3.0 d2:0.57, mape:59.0, rms:2.0945
bias:−1.2781, r:0.6372, obsmean:3.1990
DoubleCone334_10m 2.0 d2:0.38, mape:79.5, rms:2.6785
bias:−2.4991, r:0.4809, obsmean:3.1990

**DoubleCone334_10m  3.0 d2:0.57, mape:59.0, rms:2.0945**
**bias:−1.2781, r:0.6372, obsmean:3.1990**
**DoubleCone334_10m 2.0 d2:0.38, mape:79.5, rms:2.6785**
**bias:−2.4991, r:0.4809, obsmean:3.1990**

[Figure]

**GeoffreyBay336_5m  3.0 d2:0.71, mape:51.1, rms:2.0757**
**bias:−1.2867, r:0.6263, obsmean:3.2645**
**GeoffreyBay336_5m 2.0 d2:0.48, mape:81.8, rms:2.9963**
**bias:−2.7370, r:0.6518, obsmean:3.4186**

[Figure]

**Pandora_5m  3.0 d2:0.74, mape:47.5, rms:1.6856**
**bias:−1.1941, r:0.7463, obsmean:3.3284**
**Pandora_5m 2.0 d2:0.49, mape:73.8, rms:2.6102**
**bias:−2.4281, r:0.8166, obsmean:3.3284**

[Figure]

[Figure]

Pelorus686_28m  3.0 d2:0.79, mape:56.4, rms:1.4723
bias:−0.9280, r:0.7673, obsmean:2.1467
Pelorus686_28m 2.0 d2:0.69, mape:66.9, rms:1.7906
bias:−1.6133, r:0.8244, obsmean:2.5784

[Figure]

Pelorus686_14m  3.0 d2:0.75, mape:54.8, rms:1.4622
bias:−0.9590, r:0.7311, obsmean:2.5622
Pelorus686_14m 2.0 d2:0.51, mape:72.0, rms:1.9891
bias:−1.8062, r:0.8184, obsmean:2.5622

[Figure]

Pelorus686_0m  3.0 d2:0.71, mape:72.1, rms:1.4860
bias:−1.1856, r:0.7430, obsmean:1.9342
Pelorus686_0m 2.0 d2:0.50, mape:81.2, rms:1.9110
bias:−1.7108, r:0.8447, obsmean:2.1720

[Figure]

[Figure]

[Figure]

[Figure]

[Figure]

[Figure]

[Figure]

[Figure]

[Figure]

[Figure]

[Figure]

[Figure]

[Figure]

[Figure]

[Figure]

[Figure]

Doublel520_18m  3.0 d2:0.49, mape:83.0, rms:1.7888
bias:−1.2322, r:0.2824, obsmean:2.1176
Doublel520_18m 2.0 d2:0.41, mape:72.2, rms:1.7771
bias:−1.4020, r:0.2258, obsmean:2.1854

[Figure]

Doublel520_0m  3.0 d2:0.38, mape:96.1, rms:1.6505
bias:−0.9586, r:0.0324, obsmean:1.7727
Doublel520_0m 2.0 d2:0.32, mape:79.7, rms:1.5164
bias:−1.1006, r:−0.0882, obsmean:1.8115

[Figure]

PortD_15m  3.0 d2:0.48, mape:108.5, rms:1.7973
bias:−1.1596, r:0.1849, obsmean:2.0684
PortD_15m 2.0 d2:0.39, mape:88.4, rms:1.7539
bias:−1.2932, r:0.0399, obsmean:2.0950

[Figure]

[Figure]

**13. Simulated NOx assessment against AIMS Long Term Monitoring**

[Figure]

*Figure 8  Metrics for Long Term Monitoring sites NO3 assessment against observations for model version 3p0 and 2p0 d2 = Willmott index see Statistical metric page 27.mae:mean absolute error, rms root mean square*

[Figure]

[Figure]

[Figure]

[Figure]

[Figure]

[Figure]

[Figure]

Daydream_23m  3.0 d2:0.38, mape:70.0, rms:5.9701
bias:−3.1180, r:−0.0331, obsmean:3.7987
Daydream_23m 2.0 d2:0.38, mape:56.5, rms:5.7215
bias:−3.0172, r:0.5135, obsmean:3.7987

[Figure]

Daydream330_10m  3.0 d2:0.41, mape:88.8, rms:5.3704
bias:−3.3853, r:0.0920, obsmean:3.7434
Daydream330_10m 2.0 d2:0.41, mape:83.4, rms:5.2331
bias:−3.3416, r:0.7824, obsmean:3.7434

[Figure]

DoubleCone_23m  3.0 d2:0.49, mape:78.3, rms:1.6977
bias:−1.1957, r:0.2328, obsmean:1.5676
DoubleCone_23m 2.0 d2:0.41, mape:76.9, rms:1.6603
bias:−1.2709, r:0.0638, obsmean:1.5635

**DoubleCone334_10m 3.0 d2:0.48, mape:86.2, rms:2.3105**
**bias:−1.7163, r:0.4151, obsmean:2.0549**
**DoubleCone334_10m 2.0 d2:0.44, mape:83.3, rms:2.4648**
**bias:−1.8359, r:0.4988, obsmean:2.0549**

[Figure]

**GeoffreyBay336_5m 3.0 d2:0.30, mape:119.3, rms:7.9536**
**bias:−2.8808, r:−0.0450, obsmean:4.2282**
**GeoffreyBay336_5m 2.0 d2:0.40, mape:92.5, rms:6.7874**
**bias:−4.2492, r:−0.0701, obsmean:4.5487**

[Figure]

**Pandora_5m 3.0 d2:0.30, mape:342.3, rms:4.9838**
**bias:−2.1844, r:−0.3308, obsmean:3.1039**
**Pandora_5m 2.0 d2:0.42, mape:93.3, rms:4.2381**
**bias:−2.9310, r:−0.4393, obsmean:3.1039**

**Pelorus686_28m 3.0 d2:0.45, mape:199.3, rms:2.7353**
**bias:−0.1555, r:0.3491, obsmean:1.3843**
**Pelorus686_28m 2.0 d2:0.54, mape:72.8, rms:2.1121**
**bias:−0.4446, r:0.6411, obsmean:1.6470**

[Figure]

**Pelorus686_14m 3.0 d2:0.12, mape:384.5, rms:2.1355**
**bias:−0.2636, r:−0.5484, obsmean:1.2138**
**Pelorus686_14m 2.0 d2:0.41, mape:84.2, rms:1.1510**
**bias:−0.8703, r:−0.2183, obsmean:1.2138**

[Figure]

**Pelorus686_0m 3.0 d2:0.28, mape:136.0, rms:2.0423**
**bias:−0.3503, r:−0.0325, obsmean:1.1362**
**Pelorus686_0m 2.0 d2:0.38, mape:80.0, rms:1.5936**
**bias:−1.1607, r:−0.2143, obsmean:1.3218**

[Figure]

**Dunk859_5m  3.0 d2:0.09, mape:513.9, rms:9.0890**
**bias:2.9842, r:-0.1698, obsmean:2.3160**
**Dunk859_5m 2.0 d2:0.30, mape:198.9, rms:3.9263**
**bias:-0.0223, r:-0.0643, obsmean:2.3160**

[Figure]

**Russell695_20m  3.0 d2:0.80, mape:90.9, rms:1.0418**
**bias:-0.5843, r:0.7304, obsmean:1.0395**
**Russell695_20m 2.0 d2:0.45, mape:118.0, rms:1.5485**
**bias:-0.5405, r:0.2552, obsmean:1.1335**

[Figure]

**Russell695_10m  3.0 d2:0.32, mape:134.6, rms:1.1518**
**bias:-0.6778, r:-0.1952, obsmean:1.0413**
**Russell695_10m 2.0 d2:0.33, mape:97.1, rms:1.3502**
**bias:-0.4942, r:0.0698, obsmean:1.0413**

[Figure]

[Figure]

[Figure]

[Figure]

[Figure]

[Figure]

[Figure]

[Figure]

[Figure]

[Figure]

[Figure]

[Figure]

[Figure]

[Figure]

[Figure]

[Figure]

[Figure]

[Figure]

**14. Simulated NH4 assessment against AIMS Long Term Monitoring**

[Figure]

*Figure 9 Metrics for Long Term Monitoring sites NH4 assessment against observations for model version 3p0 and 2p0 d2 = Willmott index see Statistical metric page 27.mae:mean absolute error, rms root mean square*

[Figure]

[Figure]

[Figure]

[Figure]

[Figure]

[Figure]

[Figure]

[Figure]

[Figure]

[Figure]

[Figure]

[Figure]

[Figure]

[Figure]

[Figure]

[Figure]

[Figure]

[Figure]

[Figure]

[Figure]

[Figure]

[Figure]

[Figure]

[Figure]

[Figure]

[Figure]

[Figure]

[Figure]

Green830_36m  3.0 d2:0.44, mape:109.7, rms:1.5635
bias:−0.7596, r:0.0206, obsmean:1.0298
Green830_36m 2.0 d2:0.39, mape:211.8, rms:1.8060
bias:−0.5515, r:−0.0371, obsmean:1.1478

[Figure]

Green830_18m  3.0 d2:0.41, mape:85.3, rms:1.4584
bias:−0.7597, r:0.0662, obsmean:0.8007
Green830_18m 2.0 d2:0.40, mape:75.8, rms:1.4221
bias:−0.7089, r:0.1792, obsmean:0.8007

[Figure]

Green830_0m  3.0 d2:0.46, mape:376.2, rms:0.3936
bias:−0.2920, r:−0.0007, obsmean:0.3296
Green830_0m 2.0 d2:0.46, mape:533.1, rms:0.4101
bias:−0.3135, r:0.1547, obsmean:0.3712

[Figure]

[Figure]

[Figure]

[Figure]

[Figure]

[Figure]

**15. Simulated DON assessment against Long Term Monitoring**

[Figure]

*Figure 10  Metrics for Long Term Monitoring sites DON assessment against observations for model version 3p0 and 2p0 d2 = Willmott index see Statistical metric page 27.mae:mean absolute error, rms root mean square*

[Figure]

[Figure]

[Figure]

[Figure]

[Figure]

[Figure]

[Figure]

[Figure]

[Figure]

**DoubleCone334_10m  3.0 d2:0.36, mape:43.1, rms:38.1129
bias:13.2573, r:0.0655, obsmean:80.0153
DoubleCone334_10m 2.0 d2:0.34, mape:46.1, rms:38.4647
bias:16.4833, r:0.0542, obsmean:80.0153**

[Figure]

**GeoffreyBay336_5m  3.0 d2:0.60, mape:26.5, rms:25.0887
bias:−1.7775, r:0.3018, obsmean:87.2797
GeoffreyBay336_5m 2.0 d2:0.59, mape:28.7, rms:25.7535
bias:1.1268, r:0.2954, obsmean:88.3884**

[Figure]

**Pandora_5m  3.0 d2:0.63, mape:17.0, rms:19.3077
bias:−0.9496, r:0.4027, obsmean:90.6250
Pandora_5m 2.0 d2:0.60, mape:17.3, rms:19.5210
bias:1.5785, r:0.3654, obsmean:90.6250**

[Figure]

[Figure]

[Figure]

[Figure]

[Figure]

[Figure]

[Figure]

[Figure]

[Figure]

[Figure]

[Figure]

[Figure]

[Figure]

[Figure]

[Figure]

[Figure]

[Figure]

[Figure]

[Figure]

Doublel520_18m 3.0 d2:0.39, mape:32.7, rms:38.0161
bias:-24.1627, r:-0.0160, obsmean:86.0209
Doublel520_18m 2.0 d2:0.39, mape:31.5, rms:37.3854
bias:-20.1241, r:0.0034, obsmean:85.7153

[Figure]

Doublel520_0m 3.0 d2:0.45, mape:30.3, rms:34.1023
bias:-24.4886, r:0.1262, obsmean:86.5055
Doublel520_0m 2.0 d2:0.44, mape:30.0, rms:34.8091
bias:-22.9979, r:0.0667, obsmean:88.7066

[Figure]

PortD_15m 3.0 d2:0.44, mape:26.5, rms:39.4654
bias:-20.2788, r:0.0194, obsmean:83.8735
PortD_15m 2.0 d2:0.43, mape:26.3, rms:40.1281
bias:-16.8833, r:0.0269, obsmean:84.7222

[Figure]

[Figure]

[Figure]

**16. Simulated DOP assessment against Long Term Monitoring**

[Figure]

*Figure 11  Metrics for Long Term Monitoring sites DOP assessment against observations for model version 3p0 and 2p0 d2 = Willmott index see Statistical metric page 27.mae:mean absolute error, rms root mean square*

[Figure]

Pelican_5m  3.0 d2:0.34, mape:213.0, rms:9.0130
bias:8.1407, r:−0.0260, obsmean:6.0475
Pelican_5m 2.0 d2:0.34, mape:214.8, rms:9.0716
bias:8.3057, r:−0.0211, obsmean:6.0475

[Figure]

Humpy873_10m  3.0 d2:0.30, mape:170.8, rms:8.2185
bias:7.6571, r:0.1190, obsmean:5.5803
Humpy873_10m 2.0 d2:0.31, mape:174.7, rms:8.2907
bias:7.8138, r:0.1271, obsmean:5.5803

[Figure]

Humpy873_0m  3.0 d2:0.22, mape:198.8, rms:9.0211
bias:8.4801, r:−0.0761, obsmean:5.0333
Humpy873_0m 2.0 d2:0.21, mape:204.4, rms:9.1983
bias:8.6909, r:−0.1363, obsmean:5.0333

[Figure]

[Figure]

[Figure]

[Figure]

[Figure]

[Figure]

**DoubleCone334_10m  3.0 d2:0.24, mape:209.7, rms:8.8998
bias:8.0449, r:0.0996, obsmean:4.6710
DoubleCone334_10m 2.0 d2:0.22, mape:229.3, rms:9.5831
bias:8.8789, r:0.0968, obsmean:4.6710**

[Figure]

**GeoffreyBay336_5m  3.0 d2:0.29, mape:135.0, rms:6.7514
bias:6.0107, r:0.0362, obsmean:5.5183
GeoffreyBay336_5m 2.0 d2:0.28, mape:147.1, rms:7.3647
bias:6.7776, r:0.1103, obsmean:5.6437**

[Figure]

**Pandora_5m  3.0 d2:0.24, mape:272.9, rms:8.5729
bias:7.4558, r:−0.5241, obsmean:4.8058
Pandora_5m 2.0 d2:0.25, mape:284.1, rms:8.9209
bias:7.9681, r:−0.4350, obsmean:4.8058**

[Figure]

[Figure]

[Figure]

[Figure]

[Figure]

[Figure]

[Figure]

[Figure]

[Figure]

[Figure]

[Figure]

[Figure]

[Figure]

**FairleadBuoy518_0m 3.0 d2:0.31, mape:126.3, rms:5.4486**
**bias:4.2860, r:-0.1066, obsmean:4.8941**
**FairleadBuoy518_0m 2.0 d2:0.31, mape:145.2, rms:6.1590**
**bias:5.0506, r:-0.1021, obsmean:5.0971**

[Figure]

**Yorkeys519_8m 3.0 d2:0.32, mape:117.7, rms:5.2190**
**bias:4.2313, r:-0.0351, obsmean:4.8639**
**Yorkeys519_8m 2.0 d2:0.30, mape:144.1, rms:6.1209**
**bias:5.1846, r:-0.0482, obsmean:4.8802**

[Figure]

**Yorkeys519_0m 3.0 d2:0.32, mape:150.2, rms:5.5450**
**bias:4.3322, r:-0.1136, obsmean:4.6951**
**Yorkeys519_0m 2.0 d2:0.32, mape:177.3, rms:6.2968**
**bias:5.1391, r:-0.0955, obsmean:4.8112**

[Figure]

[Figure]

[Figure]

[Figure]

[Figure]

[Figure]

[Figure]

[Figure]

[Figure]

**17.  Simulated EFI assessment against Long Term Monitoring TSS**

[Figure]

*Figure 12  Metrics for Long Term Monitoring sites EFI model assessment against TSS observations for model version 3p0 and 2p0 d2 = Willmott index see Statistical metric page 27.mae:mean absolute error, rms root mean square*

[Figure]

Pelican_5m 3.0 d2:0.41, mape:79.4, rms:0.0032
bias:−0.0019, r:0.0770, obsmean:0.0027
Pelican_5m 2.0 d2:0.41, mape:81.4, rms:0.0035
bias:−0.0024, r:−0.1694, obsmean:0.0027

[Figure]

Humpy873_10m 3.0 d2:0.44, mape:85.4, rms:0.0011
bias:−0.0002, r:0.1055, obsmean:0.0009
Humpy873_10m 2.0 d2:0.50, mape:72.5, rms:0.0011
bias:−0.0005, r:0.1395, obsmean:0.0009

[Figure]

Humpy873_0m 3.0 d2:0.41, mape:91.6, rms:0.0009
bias:−0.0001, r:0.1776, obsmean:0.0006
Humpy873_0m 2.0 d2:0.28, mape:112.5, rms:0.0010
bias:−0.0003, r:−0.0965, obsmean:0.0006

[Figure]

[Figure]

[Figure]

[Figure]

Daydream_23m  3.0 d2:0.41, mape:61.2, rms:0.0033
bias:−0.0012, r:0.1259, obsmean:0.0030
Daydream_23m 2.0 d2:0.36, mape:68.0, rms:0.0035
bias:−0.0016, r:−0.0337, obsmean:0.0030

[Figure]

Daydream330_10m  3.0 d2:0.55, mape:47.1, rms:0.0026
bias:−0.0016, r:0.4055, obsmean:0.0028
Daydream330_10m 2.0 d2:0.46, mape:54.5, rms:0.0029
bias:−0.0018, r:0.2468, obsmean:0.0028

[Figure]

DoubleCone_23m  3.0 d2:0.49, mape:61.0, rms:0.0023
bias:−0.0014, r:0.3674, obsmean:0.0025
DoubleCone_23m 2.0 d2:0.42, mape:68.6, rms:0.0022
bias:−0.0010, r:0.0436, obsmean:0.0021

[Figure]

[Figure]

[Figure]

[Figure]

[Figure]

[Figure]

[Figure]

Dunk859_5m  3.0 d2:0.44, mape:44.5, rms:0.0013
bias:-0.0008, r:-0.0062, obsmean:0.0019
Dunk859_5m 2.0 d2:0.45, mape:57.5, rms:0.0016
bias:-0.0012, r:0.1025, obsmean:0.0019

[Figure]

Russell695_20m  3.0 d2:0.36, mape:71.5, rms:0.0007
bias:-0.0003, r:0.0144, obsmean:0.0009
Russell695_20m 2.0 d2:0.28, mape:58.1, rms:0.0007
bias:-0.0001, r:0.0532, obsmean:0.0010

[Figure]

Russell695_10m  3.0 d2:0.36, mape:277.0, rms:0.0006
bias:-0.0002, r:-0.1517, obsmean:0.0007
Russell695_10m 2.0 d2:0.65, mape:145.8, rms:0.0004
bias:-0.0002, r:0.5109, obsmean:0.0007

[Figure]

[Figure]

[Figure]

[Figure]

[Figure]

[Figure]

**Yorkeys519_8m 3.0 d2:0.38, mape:55.0, rms:0.0045**
**bias:−0.0026, r:−0.1355, obsmean:0.0043**
**Yorkeys519_8m 2.0 d2:0.40, mape:68.8, rms:0.0037**
**bias:−0.0027, r:0.3853, obsmean:0.0035**

[Figure]

**Yorkeys519_8m 3.0 d2:0.38, mape:55.0, rms:0.0045**
**bias:−0.0026, r:−0.1355, obsmean:0.0043**
**Yorkeys519_8m 2.0 d2:0.40, mape:68.8, rms:0.0037**
**bias:−0.0027, r:0.3853, obsmean:0.0035**

[Figure]

**Yorkeys519_0m 3.0 d2:0.45, mape:70.4, rms:0.0032**
**bias:−0.0023, r:0.3135, obsmean:0.0029**
**Yorkeys519_0m 2.0 d2:0.44, mape:83.9, rms:0.0025**
**bias:−0.0020, r:0.4884, obsmean:0.0023**

[Figure]

[Figure]

[Figure]

[Figure]

[Figure]

[Figure]

[Figure]

[Figure]

**18.  Simulated Chl *a* assessment against IMOS NRS HPLC Chl *a**

[Figure]

*Figure 13  Metrics for IMOS NRS sites Chlorophyll assessment against observations for model version 3p0 and 2p0 d2 = Willmott index see Statistical metric page 27.mae:mean absolute error, rms root mean square*

[Figure]

[Figure]

[Figure]

[Figure]

[Figure]

**19. Simulated Chl *a* and Fluorescence assessment against AIMS MMP fluorescence (includes scatter plots)**

[Figure]

*Figure 14  Metrics for AIMS MMP fluorescence against Chl a and fluorescence for model version 3p0 and 2p0 d2 = Willmott index see Statistical metric page 27.mae:mean absolute error, rms root mean square*

**BUR13  B3.0 Chla Willmott:0.42, mape:70.1, rms:0.79**
**bias:−0.69, r:0.26, obsmean:0.95**
**BUR13 B3.0 Fluor Willmott:0.37, mape:89.8, rms:0.94**
**bias:−0.86, r:0.25, obsmean:0.95**

[Figure]

**TUL10  B3.0 Chla Willmott:0.39, mape:58.1, rms:0.94**
**bias:−0.66, r:0.09, obsmean:1.05**
**TUL10 B3.0 Fluor Willmott:0.37, mape:84.6, rms:1.14**
**bias:−0.92, r:0.08, obsmean:1.05**

[Figure]

**RM10  B3.0 Chla Willmott:0.45, mape:66.6, rms:0.54**
**bias:−0.36, r:0.14, obsmean:0.76**
**RM10 B3.0 Fluor Willmott:0.42, mape:79.3, rms:0.74**
**bias:−0.63, r:0.11, obsmean:0.76**

[Figure]

[Figure]

[Figure]

[Figure]

[Figure]

[Figure]

[Figure]

[Figure]

[Figure]

[Figure]

[Figure]

**Pelorus_5m  B3.0 Chla Willmott:0.37, mape:53.1, rms:0.46
bias:−0.25, r:0.11, obsmean:0.49
Pelorus_5m B3.0 Fluor Willmott:0.35, mape:80.9, rms:0.56
bias:−0.41, r:0.10, obsmean:0.49**

[Figure]

**Dunk859_5m  B3.0 Chla Willmott:0.27, mape:93.6, rms:0.72
bias:−0.25, r:0.09, obsmean:0.57
Dunk859_5m B3.0 Fluor Willmott:0.31, mape:83.3, rms:0.81
bias:−0.46, r:0.08, obsmean:0.57**

[Figure]

**Russell_5m  B3.0 Chla Willmott:0.46, mape:58.4, rms:0.27
bias:−0.14, r:0.14, obsmean:0.41
Russell_5m B3.0 Fluor Willmott:0.40, mape:77.0, rms:0.38
bias:−0.32, r:0.14, obsmean:0.41**

[Figure]

**High_5m  B3.0 Chla Willmott:0.45, mape:72.4, rms:0.27**
**bias:−0.08, r:0.14, obsmean:0.40**
**High_5m B3.0 Fluor Willmott:0.41, mape:74.6, rms:0.37**
**bias:−0.30, r:0.13, obsmean:0.40**

[Figure]

**Fitz_5m  B3.0 Chla Willmott:0.47, mape:50.5, rms:0.32**
**bias:−0.15, r:0.24, obsmean:0.42**
**Fitz_5m B3.0 Fluor Willmott:0.40, mape:76.3, rms:0.43**
**bias:−0.33, r:0.23, obsmean:0.42**

[Figure]

*Figure 15 Scatter plot of observed Fluorescence for AIMS MMP assessment against simulated Chl a for model version 3p0*

[Figure]

**20. Simulated Turbidity assessment against AIMS MMP Turbidity**

[Figure]

*Figure 16 Metrics for AIMS MMP turbidity against simulated turbidity Dec 2010 to November 2018 for model version 3p0 d2 = Willmott index see Statistical metric page 27.mae:mean absolute error, rms root mean square*

**Simulated and observed turbidity at MMP sites (y axis to max extent)**

[Figure]

[Figure]

[Figure]

[Figure]

[Figure]

[Figure]

[Figure]

[Figure]

[Figure]

[Figure]

[Figure]

[Figure]

[Figure]

[Figure]

[Figure]

**Simulated and observed turbidity at MMP sites (y axis fixed at 20 NTU)**

[Figure]

[Figure]

[Figure]

[Figure]

[Figure]

[Figure]

[Figure]

[Figure]

[Figure]

[Figure]

[Figure]

[Figure]

[Figure]

[Figure]

[Figure]

[Figure]

[Figure]

[Figure]

[Figure]

[Figure]

**21.    Simulated Chl *a* assessment against IMOS/NRS fluorescence**

[Figure]

*Figure 17 Metrics for IMOS and NRS fluorescence against Chl a for model version 3p0 and 2p0 d2 = Willmott index see Statistical metric page 27.mae:mean absolute error, rms root mean square*

**North_Stradbroke_20  3.0 d2:0.45, mape:129.4, rms:0.5768**
**bias:0.0702, r:0.2402, obsmean:0.4530**
**North_Stradbroke_20 2.0 d2:0.44, mape:98.8, rms:0.5501**
**bias:−0.0500, r:0.2529, obsmean:0.4530**

[Figure]

**North_Stradbroke_50  3.0 d2:0.42, mape:301.7, rms:0.7567**
**bias:0.4732, r:0.1829, obsmean:0.4872**
**North_Stradbroke_50 2.0 d2:0.42, mape:266.1, rms:0.6750**
**bias:0.3632, r:0.1515, obsmean:0.4872**

[Figure]

**GBROTE_45m  3.0 d2:0.30, mape:125.5, rms:0.6452**
**bias:0.4133, r:−0.0315, obsmean:0.6461**
**GBROTE_45m 2.0 d2:0.33, mape:94.0, rms:0.5041**
**bias:0.2175, r:−0.0193, obsmean:0.6461**

[Figure]

[Figure]

[Figure]

[Figure]

[Figure]

[Figure]

[Figure]

[Figure]

[Figure]

[Figure]

[Figure]

[Figure]

**GBRMYR_17m 3.0 d2:0.49, mape:47.3, rms:0.1464**
**bias:−0.0480, r:0.2613, obsmean:0.1679**
**GBRMYR_17m 2.0 d2:0.45, mape:58.0, rms:0.1573**
**bias:−0.0877, r:0.2652, obsmean:0.1679**

[Figure]

**GBRLSH_30m 3.0 d2:0.30, mape:66.0, rms:0.5020**
**bias:−0.2140, r:−0.1121, obsmean:0.5537**
**GBRLSH_30m 2.0 d2:0.33, mape:64.9, rms:0.5123**
**bias:−0.2955, r:−0.0946, obsmean:0.5537**

**22. Simulated NOx assessment against NRS: Yongala and NSI**

[Figure]

*Figure 18  Metrics for NRS NOx against model version 3p0 and 2p0 until 2014 for model version 3p0 and 2p0 d2 = Willmott index see Statistical metric page 27.mae:mean absolute error, rms root mean square*

[Figure]

[Figure]

[Figure]

[Figure]

[Figure]

**23.    Simulated NH4 assessment against NRS: Yongala and NSI**

[Figure]

*Figure 19  Metrics for NRS NH4 for model version 3p0 and 2p0 d2 = Willmott index see Statistical metric page 27.mae:mean absolute error, rms root mean square*

**North_Stradbroke_50  3.0 d2:0.31, mape:168.9, rms:3.9650**
**bias:0.1315, r:-0.0367, obsmean:3.0019**
**North_Stradbroke_50 2.0 d2:0.34, mape:164.3, rms:3.9159**
**bias:0.1140, r:0.0275, obsmean:2.8723**

[Figure]

**North_Stradbroke_20  3.0 d2:0.39, mape:94.1, rms:4.4470**
**bias:-2.7033, r:-0.0254, obsmean:2.7669**
**North_Stradbroke_20 2.0 d2:0.38, mape:93.5, rms:3.9994**
**bias:-2.4815, r:-0.0191, obsmean:2.5500**

[Figure]

**North_Stradbroke_0  3.0 d2:0.39, mape:93.1, rms:3.1746**
**bias:-1.9830, r:-0.2707, obsmean:2.0403**
**North_Stradbroke_0 2.0 d2:0.39, mape:94.4, rms:3.0627**
**bias:-1.9039, r:-0.2316, obsmean:1.9543**

[Figure]

[Figure]

**24. Simulated DIP assessment against NRS: Yongala and NSI**

[Figure]

Metrics for IMOS NRS DIP for model version 3p0 and 2p0 d2 = Willmott index see Statistical metric page 27.mae:mean absolute error, rms root mean square

[Figure]

[Figure]

[Figure]

[Figure]

[Figure]

**25. Simulated DIC assessment against NRS Yongala**

**Yongala_26  3.0 d2:0.58, mape:0.9, rms:251.8943
bias:−75.8681, r:0.4184, obsmean:23057.0386
Yongala_26 2.0 d2:0.56, mape:1.1, rms:299.8606
bias:−157.4924, r:0.3608, obsmean:23057.0386**

[Figure]

**Yongala_10  3.0 d2:0.76, mape:0.9, rms:258.2364
bias:−44.3495, r:0.7547, obsmean:22989.6377
Yongala_10 2.0 d2:0.69, mape:1.1, rms:310.3398
bias:−126.1681, r:0.6250, obsmean:22989.6377**

[Figure]

**Yongala_0  3.0 d2:0.64, mape:1.2, rms:409.6874
bias:−2.2563, r:0.7140, obsmean:22933.1885
Yongala_0 2.0 d2:0.57, mape:1.4, rms:452.0324
bias:−82.6526, r:0.5539, obsmean:22933.1885**

[Figure]

**26. Simulated alkalinity assessment against NRS Yongala North Stradbroke**

[Figure]

[Figure]

[Figure]

[Figure]

[Figure]

[Figure]

[Figure]

**27. Simulated aragonite assessment against Yongala**

[Figure]

[Figure]

**28. Wakmatha transect line for Carbon chemistry assessment**

[Figure]

**29. Satellite images of MMP NRS and LTM sites**

[Figure]

[Figure]

[Figure]

[Figure]

[Figure]

[Figure]

Capricorn Channel (IMOS)

Heron Island North (no WQ)(IMOS)

One Tree East (IMOS)

Heron Island South (IMOS)

[Figure]

Double Cone Island

Daydream Island

Pine Island

---

## Author Response (AR3)

Dear Dr. Hargraves,

Thank you again for editing this manuscript. We have now completed all your requested revisions.

**Major Revision**

**Topical Editor Decision: Publish subject to minor revisions (review by editor)** (22 Jun 2020) by Julia Hargreaves
Comments to the Author:
The manuscript is getting quite close to being acceptable now.

In the main text, please move the Code and data availability section to the correct place in the manuscript (after Conclusions). I noticed that there should be an "s" on the last word of section 14.2 so please correct that and go through and look for other grammatical errors. The manuscript will be copyedited but the more mistakes we catch now, the fewer there should be in the final version.

At GMD, Appendices are included in the main manuscript not in the Supplement. You therefore need to change the labels on the sections of the supplement. Maybe S1, S2 etc rather than Appendix A,B....

Please explain the provenance of the presently labelled section Appendix E in the Supplement. There is a page within this Appendix that refers to previous versions of the document and the style is that of an internal report. This makes me think it is actually a CSIRO internal report. If this is the case, the authors of the report need to be listed and their permission gained to the report to be included in the supplement and I would prefer this to be uploaded as a separate standalone file in the Supplement, in its original format.

Uploaded Files validated (18 Jun 2020) by Anna Wenzel

File Upload (03 Jun 2020) by M. Baird ▸ Manuscript ▸ Supplement ▸ Author's Response ▸ Abstract

Changes to the manuscript:

1. Moved code and data availability to after the conclusions.
2. Thorough read to hunt out grammatical errors.
3. References to Appendix changed to Supplement. There are now no Appendices in the main document.
4. CSIRO does have reports which are held in a publication repository, but Section 5 of the supplementary material has never been submitted to this repository. I too would prefer this to be a separate document and so now have a zip file with two file: the first containing E1-4, and the second E5, the skill assessment report.

Yours sincerely,

Mark Baird

---

## Author Response (AR4)

Dear Dr. Hargreaves,

Thank you again for editing this manuscript. We have now completed all your requested revisions.

Mark (first author) stated in that email that the supplement with the CSIRO logo was written by Jennifer Skerratt who is a co-author on the manuscript. This authorship needs to be stated on the supplement itself and also acknowledged in the author contributions paragraph of the manuscript.

Authorship is now stated on the supplement and acknowledged in the author contributions of the manuscript.

With authorship of the supplement now clear, I would prefer that you remove the CSIRO logo from the supplement, as it does make it look like an internal report and this could cause confusion.

CSIRO logo removed.

You could add further acknowledgements to other CSIRO staff in the supplement if appropriate. As the supplement is not a pre-existing CSIRO report, any individuals that have a large role in the supplement and are not presently named in the authorship should be included as authors of the manuscript. Their role as contributing only to the supplement can be clearly specified in the author contribution section of the main manuscript.

There was no need for further acknowledgement as the major writers of the supplement were Jennifer Skerratt and me. This role is now stated in the author contributions.

Yours sincerely,

Mark Baird